# Depletion-dependent activity-based protein profiling using SWATH/DIA-MS detects serine hydrolase lipid remodeling in lung adenocarcinoma progression

Systematic inference of enzyme activity in human tumors is key to understanding cancer progression and resistance to therapy. However, standard protein or transcript abundances are blind to the activity status of the measured enzymes, regulated, for example, by active-site amino acid mutations or post-translational protein modifications. Current methods for activity-based proteome profiling (ABPP), which combine mass spectrometry (MS) with chemical probes, quantify the fraction of enzymes that are catalytically active. Here, we describe depletion-dependent ABPP (dd-ABPP) combined with automated SWATH/DIA-MS, which simultaneously determines three molecular layers of studied enzymes: i) catalytically active enzyme fractions, ii) enzyme and background protein abundances, and iii) context-dependent enzyme-protein interactions. We demonstrate the utility of the method in advanced lung adenocarcinoma (LUAD) by monitoring nearly 4000 protein groups and 200 serine hydrolases (SHs) in tumor and adjacent tissue sections routinely collected for patient histopathology. The activity profiles of 23 SHs and the abundance of 59 proteins associated with these enzymes retrospectively classified aggressive LUAD. The molecular signature revealed accelerated lipoprotein depalmitoylation via palmitoyl(protein)hydrolase activities, further confirmed by excess palmitate and its metabolites. The approach is universal and applicable to other enzyme families with available chemical probes, providing clinicians with a biochemical rationale for tumor sample classification.

Regulation of enzyme catalytic activity can contribute to drug resistance and cancer progression[1-4], and can be mediated by protein posttranslational modifications (PTMs)[5], nonsense somatic mutations[2], compartmentalization[6], or intra-molecular interactions[3,7]. Such biological regulations often occur without observable alterations in enzyme concentration or transcript expression levels.

SHs are one of the largest enzyme families[8], accounting for 1–1.5% of all human proteins[9], with more than 250 members annotated in the human genome. They catalyze a wide range of vital biochemical reactions such as the hydrolytic cleavages of peptide bonds in proteins[10], the ester bonds in lipids[11], or the thioester linkages of lipid moieties in proteins[12], representing potent pharmacological targets[8,13,14]. This is exemplified by the serine proteases of the complement cascade in inflammatory diseases[15] or the serine proteases that play a central role in molecular pathways underlying the hallmarks of cancer[16], and are typically activated by selective proteolytic

e-mail: tatjana.sajic@chuv.ch; sven.hillinger@usz.ch; aebersold@imsb.biol.ethz.ch

processing of inactive zymogens[10]. Further, serine lipases catalyze fatty de-acylation of proteins[17], and indirectly regulate compartmentalization of numerous cancer drivers via the cleavage of lipid moieties[6,12,18] or modulate lipid composition in primary tumors to promote cancer pathogenesis[19].

The level of catalytically active enzymes can be determined by activity-based chemical probes, reagents that selectively react with groups of mechanistically related enzymes to form stable intermediates[20,21]. Tagged proteins are then analyzed by MS[22]. Using the ABPP method, several de-regulated groups of enzymes related to cancer progression have been discovered[12,23,24] and the altered activity of SHs has been linked to poor clinical outcomes in cancer[25].

However, the molecular processes associated with the altered catalytic regulation of concerned enzymes in cancerous tumors remain largely unexplored in biomarker and drug-target discovery[26]. Typical ABPP quantifies solely the fraction of enzymes that are catalytically active but does not indicate the relative abundance of the enzymes, thus compromising the interpretation of the overall enzyme regulation.

Several other technical issues have limited broader clinical application of ABPP. The first limitation is poor accessibility of tumor biopsies for reliable enzyme profiling. In oncology or any tissue bank collection, the surgically resected specimens are commonly prepared using tissue embedding materials[27,28], complicating enzyme analyses based on fresh or frozen tissue. Chemical removal of embedding material using detergents or organic solvents may result in the enzyme losing catalytic activity. Second, standard peptide samples derived from on-bead digestion of tagged and captured enzymes in standard ABPP are frequently contaminated with high levels of streptavidin peptides, which additionally interfere with MS signals of the target substrates, particularly those of low intensity[25]. Third, recent advances in peptide quantification based on data-independent MS acquisition (DIA)[29–34] have not been sufficiently exploited in activity based profiling[35]. The DIA-MS mode ensures consistency of sample measurement[32,33] by generating complete data records of all ionized peptides from biological samples without instrument-driven criteria. Bioinformatics tools have been developed to improve the analysis of complex DIA data records, including spectral deconvolution, reproducible signal mining, and quantification accuracy based on MS2 intensity trace analysis[29–31,34,36]. Previous studies have used fluorescence-based in-gel protein quantification or on-bead peptide quantification based on data-dependent MS acquisition (DDA-MS)[23], which leads to sparse matrices when sample cohorts are analyzed, and inconsistent MS1 peak-area quantification across samples[37].

To overcome these limitations, we established dd-ABPP-SWATH/DIA-MS method compatible with sections of optimal-cutting-temperature (OCT)-embedded tissue and combines experimental and computational techniques to concurrently determine the activity and quantity of hundreds of enzymes. This is achieved by depleting sample proteomes for enzymes tagged by class-specific activity probes, followed by measurements of the depleted and naïve (i.e., non-depleted) samples by automated sequential window acquisition of all theoretical spectra (SWATH/DIA-MS)[29,36,38].

LUAD is one of the deadliest human cancers, with limited treatment options and high mutational diversity[39–41]. The latter further challenges the selection of optimal therapy and prognosis of disease development[40,42] and previous studies showed that the routine testing of patient samples for mutational status of known LUAD oncogenes (e.g., *KRAS*, *EGFR*, *ST11*) does not improve patient prognosis[42–45]. Therefore, patient stratification with aggressive LUAD phenotypes is a major challenge and unmet clinical need[41,43,46,47].

In this work, we monitor nearly 200 members of the SH family in collected tissue pairs of primary LUAD and adjacent noncancerous tissues in advanced IIIA disease stage. A molecular signature involving the active fraction of 23 SHs and 59 tissue proteins co-captured with their active enzymes retrospectively discriminated high-risk-aggressive– from less-aggressive disease in both tumor and surrounding nontumor lung tissue. The results uncover increased lipase activities, which lead to the accelerated de-palmitoylation cycle of lipoproteins and can serve as a marker for aggressive LUAD tumor. Successive analyses of FA reveal an increase in palmitic acid and its metabolites, palmitoleic and oleic FA, likely due to palmitate (16:0) desaturation in the aggressive versus less aggressive tumors. We validate our major hits by using orthogonal analytical techniques. Our approach outperforms standard method with respect to the number of quantified SH enzymes in LUAD (e.g., 131 vs. 78, respectively) and is able to detect only minor changes in protein levels of SHs with dysregulated catalytic fractions. Collectively, the ensemble of our proteomic data integrated via computational modeling tools[48,49] has the ability to provide new insights into enzyme functional regulation in tumors, offering the user a more holistic view of the disease biology.

## Results
### Outline of the dd-ABPP method
Here, we reasoned that a divergence between two enzyme forms (i.e., active enzyme and total enzyme) monitored in parallel across distinct tissue conditions could be linked to the modulators of enzyme regulations in cancer progression. Consequently, we carried out two parallel MS measurements per sample, specifically the intensity of peptides in total tissue protein extract and an extract depleted for active enzyme through their binding with activity probe (Fig. 1a). The proteome profiles from both extracts were subsequently combined via bespoke software tools integrated in SWATH/DIA-MS workflow to generate mechanistic insights into the regulation of enzyme activity (Fig. 1a, b).

First, to maintain the enzyme activity status, the dd-ABPP extraction protocol for OCT-embedded tissue sections was optimized. The OCT medium freezes rapidly for optimal preservation of biological specimens and dissolves in water to facilitate removal from tissue. Proteins were extracted from each biospecimen by a two-step cleaning procedure (Methods). Extracts from the same amount of OCT-embedded and fresh-frozen (FF) tissue were then compared for protein and peptide yield and the number of enzymes detected on avidin-conjugated beads using standard ABPP. The workflow was as effective for OCT-embedded as for FF samples, as demonstrated by a large overlap of identified proteins (-90%) of the total sample digests and comparable numbers of captured enzymes (Supplementary Fig. 1a, b).

Then, one aliquot of the resulting native protein extract was incubated with reaction solvent as control and served to determine the total amount of each enzyme (total extract, Fig. 1a) and generate in-depth proteome map. A second, equal sample aliquot was incubated with the well characterized chemical probe biotinylated fluorophosphonate (6-N-biotinylaminohexylisopropylphosphorofluoridate that we named FP-probe)[25], and the active hydrolases were depleted from the sample by binding to avidin-conjugated beads (depleted extract, Fig. 1a). The two samples per tissue were recorded and the data were combined via automated OpenSWATH workflow[29] after applying the feature alignment algorithm TRIC[36] to link the complementary MS records per peptide, total and depleted (Fig. 1a).

To increase the method specificity and enzyme coverage, we assembled a project specific spectral library of high-quality SH peptides (Supplementary Fig. 1c) and combined the large MS spectral library of human tissue derived peptides[50] with the results from the deep shotgun sequencing of LUAD samples. We manually assembled a query list consisting of 241 bona fide SHs, 53 enzymes by predicted activity, 26 non-annotated cases and 15 human inactive SHs, of which we retained spectral information for 215 enzymes corresponding to 3'593 proteotypic peptides (Supplementary Fig. 1d and Supplementary Data 1). The degree of enzyme activity, or relative activity-dependent depletion index (RADDi), was calculated from the difference in the MS

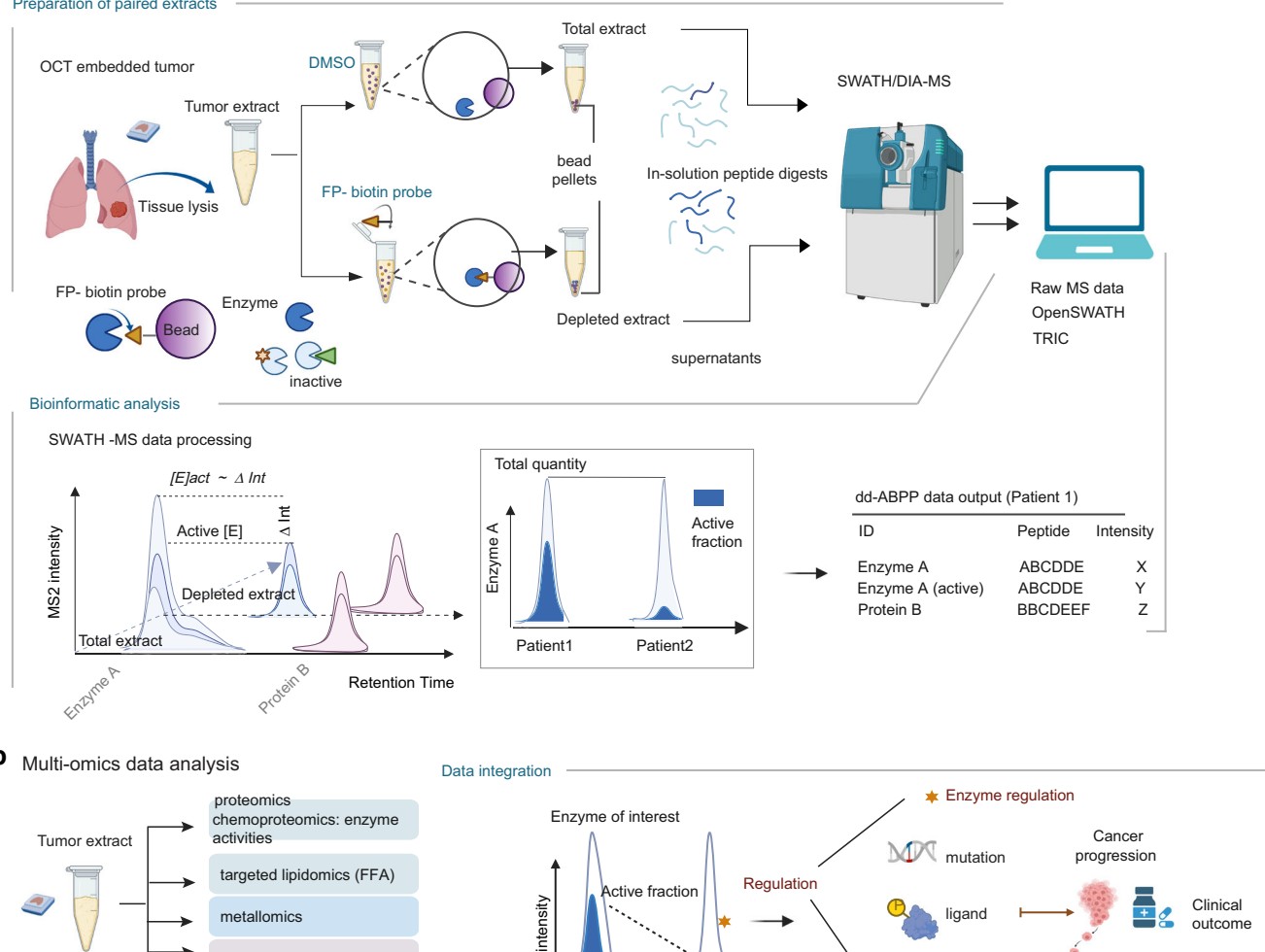

**Fig. 1 | Overview of the dd-ABPP-SWATH/DIA-MS workflow. a** Schematic of clinical sample analysis in dd-ABPP-SWATH/DIA-MS. Preparation of the paired extracts (upper panel): Protocol starting with mechanical OCT removal from OCT-embedded tissue, PBS washes, and tissue homogenization. Individual tissue extracts were incubated with reaction solvent DMSO (total extract) or biotinylated chemical probe FP-biotin (depleted extract), and active enzymes removed on streptavidin beads. The soluble proteomes were recovered in the remaining supernatants of the paired extracts and analyzed using an automated SWATH/DIA-MS platform that included software tools for targeted spectral library search, MS raw data alignment and quantification. Bioinformatic analysis (lower panel): Paired sample records were normalized by consistently detected fragment ion chromatograms of a set of endogenous ISPs selected from housekeeping and cytoskeletal proteins. The enzyme activity status was assessed from differences in complementary total and depleted sample records based on SWATH/DIA-MS peptide intensities (ΔInt) in a logarithmic scale (i.e., log2ΔInt). **b** Multi-omics data analysis. Integration of multiple levels of omics data, background proteome, depleted proteome and enzyme activity status recorded in the same sample set with available clinical post-surgery follow-up allows interpretation of enzyme regulation in cancer progression. Examples of cancer-related regulation of enzyme activity status detectable by dd-ABPP-MS pattern. [E]act, [E]inact, and [E]tot correspond to active, inactive, and total enzyme forms. FP-biotin ABP corresponds to biotinylated fluorophosphonate. Of note, streptavidin bead digests were optionally analyzed as negative or positive control beads by shotgun DDA-MS and data searched using the Trans-Proteomic Pipeline (TPP). Figure 1. created with BioRender.com. Source data are provided as a source data file.

fragment ion signal spectra of proteotypic SH peptides assessed from the depleted and total sample records for each tissue as a depletion ratio (i.e., log2ΔInt, Fig. 1a). Confident depletion ratios were selected based on their significant distribution shift compared with a reference set of internal standard peptides (ISPs) estimated with Kolmogorov–Smirnov (KS) distribution test (Supplementary Fig. 2a, b, c). ISPs were selected from housekeeping and cytoskeletal proteins to span the range of technical noise between paired samples (Supplementary Data 2). The combined results indicate: (i) the total amount of each detected enzyme, (ii) the fractions of catalytically active form of each detected enzyme and (iii) a quantitative map of the naïve and depleted tissue proteomes at the time of SH activity assessment. The quantitative data thus generated, i.e., SH molecular forms, reference tissue proteomes, and depleted proteomes were then computationally combined with other experimental (i.e. lipidomics, metallomics, genomics) and clinical data record to indicate the mechanisms of enzyme regulation that become altered depending on biological context (Fig. 1b).

## Evaluation of the dd-ABPP method in locally-advanced LUAD
To benchmark our method, we analyzed a cohort of tissues collected during thoracic surgical resection of untreated LUAD patients, mainly

**Table 1 | Clinical characteristics of study participants**

| Characteristics | N = 16 |
|---|---|
| Sex – no. (%) | |
| Male | 10 (62.5) |
| Female | 6 (37.5) |
| Age – year | |
| Mean | 67.7 +/– 7.6 |
| Range | 51–79 |
| Type of surgery – no. (%) | |
| pneumonectomy | 1 (6.25) |
| lobectomy | 13 (81.25) |
| wedge resection | 2 (12.5) |
| TNM adenocarcinoma type IIIA (follow up) – no. | |
| T Types 1/2 | 5/11 |
| N2 Types | 16/16 |
| M Local | 3/16 |
| M CNS | 2/16 |
| M Bone | 1/16 |
| M Lymph. node | 1/16 |
| M Visceral | 3/16 |
| M Diffuse | 2/16 |
| M UNKN | 4/16 |
| Survival – months | |
| Long OS (range) | 108 (55–162) |
| Short OS (range) | 24.5 (13–31) |
| Long RFS (range) | 50.5 (12–111) |
| Short RFS (range) | 13 (11–19) |
| Sample types – no. | |
| OCT /FF | 14/2 |
| Tumor/Non tumor | 16/16 |
| Percent tumor cells – (%) | |
| Mean | 70 |
| Range | 60–100 |
| Smoking status – no, | |
| Smokers | 13 |
| (pack-year - mean (range)) | 39 (5–70) |
| Non-smokers | 3 |
| Biospecimens available per cohort– no | |
| Discovery | |
| Adjacent tissue (Short; Long) | 12 (6; 6)/16 |
| Tumor tissue (Short; Long) | 12 (6; 6)/16 |
| Validation | |
| Adjacent tissue (Short; Long) | 14 (7; 7)/16 |
| Tumor tissue (Short; Long) | 14 (7; 7)/16 |

*TNM* stands for tumor (T), nodes (N), and metastases (M), *OS* Overall Survival, *RFS* Recidive Free Survival, *OCT* Optimal Cutting Temperature compound, *FF* Fresh Frozen. Disaggregation of data in Supplementary Data 3.

self-reported smokers with long-term clinical follow-up. (e.g., RFS = Recidive Free Survival in months, smoking status, type of surgery; Table 1, Supplementary Data 3). Standard morphological analysis, and pathology staging of hematoxylin-eosin (HE)- stained tumor (T)– and surrounding tissue (i.e., nontumor (N), Fig. 2a) were used by clinical pathologists to predict the patient outcomes in accordance with the 6[th] TNM classification guidelines (tumor size, lymph node status, and metastases)[51]. Yet, the survival outcome of patients classified with the same type IIIA adenocarcinoma stage dramatically differs based on clinical data record (Fig. 2a).

Aiming to retrospectively segregate aggressive– from less-aggressive tumor and adjacent-tissues in locally-advanced LUAD, we applied our method to 24 of 32 available tumor and nontumor biospecimens of 16 LUAD patients (Fig. 2b, Supplementary Fig. 3) in discovery experiment. We processed, respectively, 203 and 221 sections from 6 tumors and 6 nontumors of patients with 5-year overall survival, called long-term survivals; and 177 and 136 sections from 6 tumors and 6 nontumors of patients with 1-year overall survival, called short-term survivals (Fig. 2a, b). The percentage of tumor cells per each tumor tissue specimen was available from morphological analysis (Table 1). To generate standard protein abundance and SH activity profiles, we analyzed 48 total and depleted lung extracts from 24 biospecimens as described above. The proteomic data obtained this way were related to the patient clinical outcomes to derive signatures capable of retrospectively separating aggressive– from less-aggressive forms of LUAD.

## Classifying LUAD conditions via tissue proteome maps from dd-ABPP

Reference tissue proteomes are a by-product of the dd-ABPP method. We quantified and compared 4000 protein groups across tissue extracts to classify LUAD conditions (Supplementary Fig. 4a). To reduce data dimensionality[52] we applied supervised multivariate partial least squares discriminant analysis[48] (PLS-DA, Fig. 2c) that classified tumor and nontumor samples (left), although significant overlap of nontumor and tumor tissues was manifested between survival subtypes (right). Among the top selected proteins in supervised analysis, we detected 471 with statistically significant changes when we controlled patient age and smoking status (Fig. 2d). The differential proteins mostly encompassed changes between tumor and adjacent-tissue (i.e., 438 out of 471, Fig. 2d), for both survival types, and classified LUAD tumors but not survival subtypes (Supplementary Fig. 4b). For the two survival types, the overlap of differential changes between tumor and the adjacent nontumor tissue consisted of 146 common proteins revealing enrichment of glycolytic enzymes (e.g., GAPDH, ALDOC, ALDO), proteins involved in oxygen transport (e.g. HBD) and enzymes involved in heme biosynthesis (e.g. ALAD, QDPR, UROD, CPOX) (Fig. 2e, f and Supplementary Fig. 4c). Consistent with a previous report we observed a several fold increase of broncheotrachial protein MUC5B in tumors[53], a major contributor of lung mucus (Supplementary Fig. 4d). We detected smaller differences (i.e. 130 vs. 438, Fig. 2d) between long– and short-survival patients when comparing protein expressions data for tumor and for surrounding tissue. In aggressive tumors compared to less-aggressive, we detected a higher expression of delta-aminolevulinic acid dehydratase (ALAD) an enzyme of heme biosynthesis (Fig. 2f). The aggressive tumors expressed high levels of the mitochondrial enzyme ornithine amino-transferase (OAT, Fig. 2g). Tumor tissues of two survival subtypes further express differences in peroxiredoxins involved in mitochondrial redox homeostasis, proteins involved in oxygen transport, haptoglobin binding, the calcium-binding cytosolic proteins S100, and lipase inhibitors (Fig. 2h, Supplementary Fig. 4e, f). Interestingly, only five SHs had altered protein expression between any of four LUAD conditions of which three are involved in FA metabolism (ACOT7, ABHD10 and MGLL, Fig. 2i) including mitochondrial palmitoyl-protein thioesterase ABHD10. Overall, the quantitative comparison of standard proteomes in the four sample types succeeded in separating tumor and surrounding tissue in both long– and short-term surviving patients, but lower differences were found between the two LUAD survival subtypes.

## The altered SH activity profiles distinguish short- and long-term survivals

We then analyzed the SH activity profiles across LUAD conditions. The data quality inspection of paired tissue extracts indeed revealed

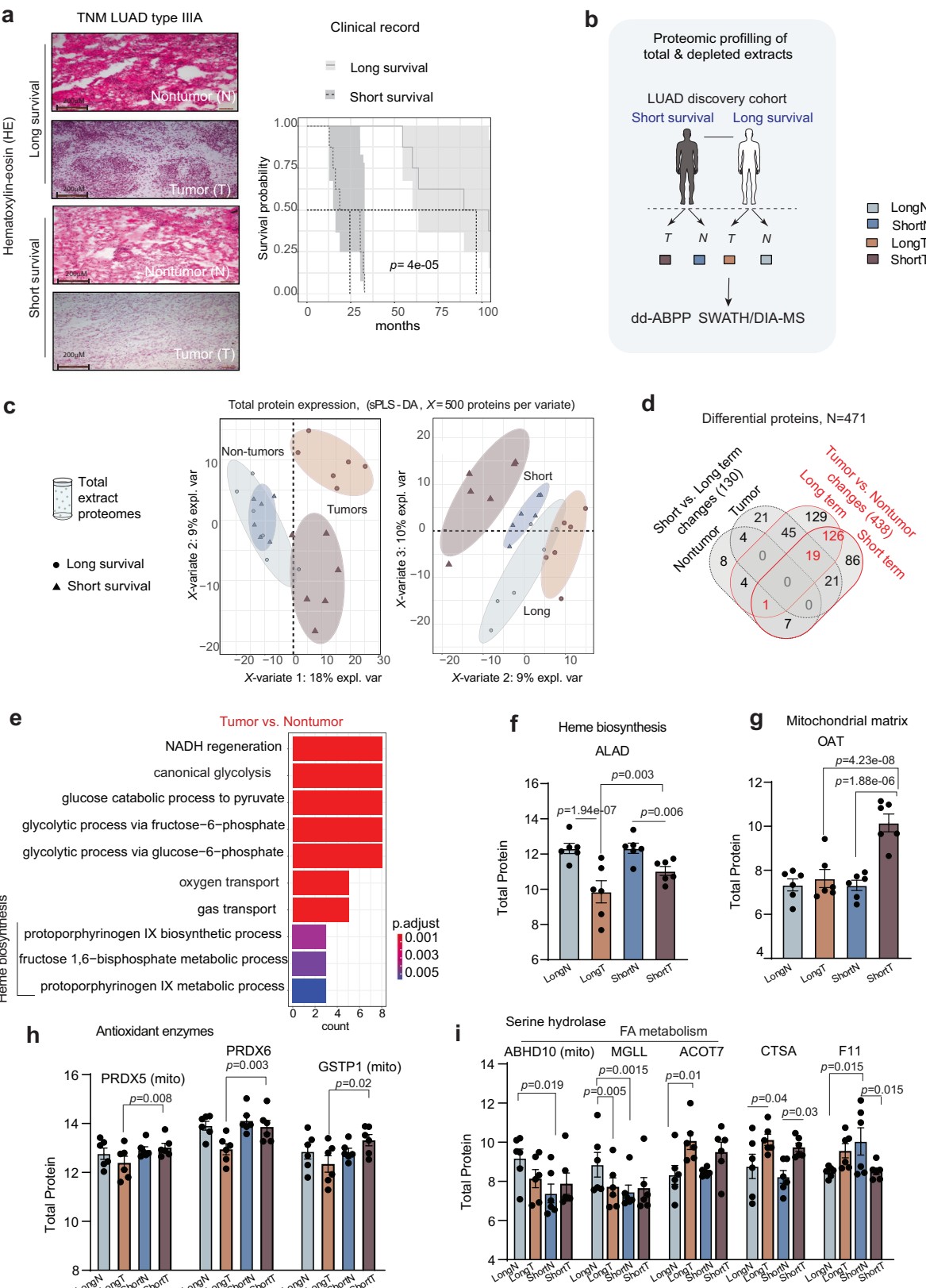

substantial depletion of SHs in extracts incubated with chemical probe compared with the matched total extracts (e.g. LYPLA1, PREP, APEH, CES1; Fig. 3a), most of which were confirmed in the peptide digests of collected beads (i.e., 77% or 27/35, Supplementary Fig. 5a).

By using a SH targeted spectral library, we monitored almost 200 SHs and assessed activity status of 131 SHs based on 793 proteotypic peptides (peptide FDR < $10^{-4}$, Supplementary Data 4). The presence of a confident active fraction in the clinical cohort was revealed for 72 out

**Fig. 2 | Detection of LUAD tissue proteomes via dd-ABPP. a** Left: HE staining of lung tissue sections from primary tumor (T) and surrounding nontumor (N) lung parenchyma. Scale bars, 200 μm. Right: Kaplan–Meier curve of survival time of patients since LUAD diagnosis. Dark grey and light grey color depict short- or long-survival patients. Two-sided *p*-value by Log-rank test. **b** Primary resected tumor and corresponding adjacent tissues (cohort 1) analyzed as depleted and total extracts by dd-ABPP-SWATH/DIA-MS proteomic workflow. Each biological sample was a pool of 30 (12–75) consecutive tissue microsections (30 μm) on average collected from the single patient specimens used in TNM classification. **c** Supervised PLS-DA analysis of protein abundance data for predefined LUAD conditions. Class separation based on variates 1 and 2 (left panel), and variates 2 and 3 (mid panel). The number of selected protein variables (*X* = 500) per PLS-DA variate. **d** Venn diagram represents overlap of differential proteins in four LUAD study comparisons, tumor versus nontumor and short- versus long- survival subtype, respectively. **e** GO enriched biological processes of proteins changed in both tumor

subtypes (i.e., 146 common proteins). Benjamini- Hochberg (BH) adjusted *p*-value < 0.005 corresponds to two-sided Fisher's exact test. All genes in *Homo sapiens* database are used as reference list. **f** LUAD group expression of ALAD. **g** Differential expression of OAT, mitochondrial protein. **h** Differential expressions of antioxidant proteins. **i** Differential protein expressions of SHs among LUAD groups. Two-sided *p*-value corresponds to a generalized linear model (GLM) with Gaussian distribution used in (**d**), (**f–i**) and accounted for patient smoking status, age and gender. Post hoc analysis (FDR) used in (**d**), (**f–i**) to address a multiple testing correction and two-sided *p*-value ≤ 0.05 considered significant. Barplots with data points shown in (**f–i**), are mean values +/− standard error of mean (S.E.M) (*n* = 6 biologically independent samples per group). Color codes correspond to sample classes annotated in figure legends. Total protein on y-axis is a log2 protein abundance. Source data and individual morphological tissue images are provided as a Source Data file.

of 131 SHs that fulfilled statistically significant criteria estimation from a KS-distance test (Fig. 3b). Remarkably, we detected twice as many catalytically active enzymes in both categories of LUAD tumors compared with adjacent tissue (i.e. 46 /47 vs. 18/22, for tumors vs. controls, respectively), triggering up to 70% of the total enzyme forms as catalytically active (e.g., PREP, LYPLA1, SIAE). To assess our procedure for intra-tissue section variability for active enzyme fraction, we collected consecutive OCT-embedded sections of a single surgically resected LUAD tumor and prepared replicated extracts. Reassuringly, among replicated extracts, the S.E.M. of measured SH activities was below <10% (Fig. 3c). For example, sialic acid acetylesterase (SIAE, Supplementary Fig. 5b) shows high individual variation between replicate extracts (e.g., S.E.M. of 9%). To determine whether the degree of SH activity detected in the LUAD samples was capable of classifying tumor vs. adjacent control tissue from two survival subtypes, we performed supervised multiclass PLS-DA analysis. Despite some overlaps, the catalytically active profiles of 64, out of 72 SHs (Supplementary Fig. 5c, d), supported a moderate separation between tumor and nontumor samples (Variates 1 and 2) and between survival phenotypes (Variates 2 and 3). Supervised analysis entirely discriminated tumors of two survival subtypes (Variate 2). Of note, the most important discriminative variables were lipases involved in protein fatty acylation status, as depicted by Lysophosphatidylserine (LPS) lipase (ABHD12)[17] (Supplementary Fig. 5d).

Subsequent differential analysis revealed that one third of active enzymes (i.e., 24 out of 72 SHs) had altered the percentage of catalytic fractions across four LUAD conditions. Hereunder, we detail the results for the different class comparisons after adjusting for potential confounders (Supplementary Table 1).

The comparison of tumor vs. adjacent nontumor-tissue for both long– and short-survival subtypes identified 11 annotated SHs with elevated activities in long-term– and 9 with elevated activities in short-term survivals (Fig. 3d–f) with an overlap of SHs elevated in tumors (e.g., SIAE, Fig. 3e and Supplementary Fig. 6a). In the tumor samples of long-term survivors, for example, SIAE, Prolyl endopeptidase (PREP), Lactotransferrin (LTF), Retinoid-inducible serine carboxypeptidase (SCPEP1) and Dipeptidyl peptidase 4 (DPP4) showed a two-to-threefold increase in the percentage of active form when compared with adjacent-tissue (Fig. 3d).

Conversely, Neutrophil Elastase (ELANE), Tripeptidyl-peptidase 1 (TPP1) and Liver Carboxylesterase 1 (CES1) showed threefold enhanced activity in more aggressive tumors of short-term survivors compared with their adjacent-tissue. Consistent with previous reports on lipid status in cancer progression[19], we found an increase in the activity of lipogenic enzyme fatty acid synthase (FASN) in long-term survivors' tumors compared with adjacent-tissue (Fig. 3f).

While some elevated SHs in LUAD tumors to adjacent-tissue displayed specific patterns of catalytic fractions for survival-subtype (e.g., FASN, DPP4, SCPEP1 for long-term survivals), we observed that most

enzymes that separated LUAD tumors from matched controls of both subtypes (e.g., Variate 1), showed an overall increased hydrolytic activity in both tumor subtypes compared to adjacent-tissues. As an example, a group of 13 SHs, including the abovementioned FASN, PREP, DPP4, ELANE, SIAE, and TPP1, separated, albeit not perfectly, tumors from surrounding lung tissue in both survival subtypes based on their consistent activity increase in tumor tissue (Fig. 3g, left). Conversely, protein levels of these enzymes (e.g. FASN, DPP4, PREP, ELANE, Fig. 3d–f) showed only weak or no changes between tumors and nontumor adjacent-tissues, resulting in substantially less accurate tumor classification (Fig. 3g, right).

Remarkably, two survival subtypes displayed different lipolytic enzyme patterns, mainly serine thioesterases that control palmitoylation of oncogenes and tumor suppressors[17,18] at cysteine residues (e.g., ABHD10, LYPLA2, ABHD12, Fig. 3h). Five lipases, Isoamyl acetate-hydrolyzing esterase 1 (IAH1), Acyl-protein thioesterase 2 (LYPLA2), Lysophospholipase-like protein 1 (LYPLAL1), LPS lipase ABHD12, palmitoyl- thioesterase ABHD10 (Fig. 3h), and the mitochondrial ATP-dependent Clp protease (CLPP, Supplementary Fig. 6b), showed significantly attenuated activity in long-term survival tumors compared with their controls or compared with the tumors of short-term survivals. We optionally performed the analysis adjusted for variability in tumor cell percentage across the tumor cuts (range: 60–100%) and noted lipases, notably, Acyl-protein thioesterase 1 (LYPLA1) and Fatty-acid amide hydrolase 2 (FAAH2), that manifested significant dysregulations across LUAD subtypes (Supplementary Fig. 6c). The lipases that catalyze the removal of a palmitic acid from cysteine residues in proteins (LYPLA2, ABHD12, LYPLA1)[18] were dominant among the SHs with a potential for separating tumors of LUAD survival subtypes.

Collectively, our dd-ABPP proteomic dataset experimentally demonstrated that classical protein expression measurements in LUAD tissues do not distinguish altered metabolic activities from dozens of serine proteases and lipases. By contrast, the attenuated lipases activities involved in protein de-lipidation status clearly indicated LUAD tumors with better prognosis.

## Validation of SH activity profiles by orthogonal techniques

Next, we collected an additional set of 1080 cryostat sections (i.e., 573 and 507 for tumor and nontumor sections, respectively) spanning over 28 tumor and adjacent-tissues originating from 14 LUAD patients of which 10 were analyzed within the discovery experiment (Fig. 2b) and four were independent of the original discovery cohort (Supplementary Fig. 3).

We first compared our results with results obtained by gel electrophoretic analyses of proteins labeled with chemical probe that integrates a TAMRA fluorescent tag at the active enzyme site[54]. Protein patterns extracted from tumor and surrounding tissue displayed higher fluorescence intensities for tumor samples in respect to their

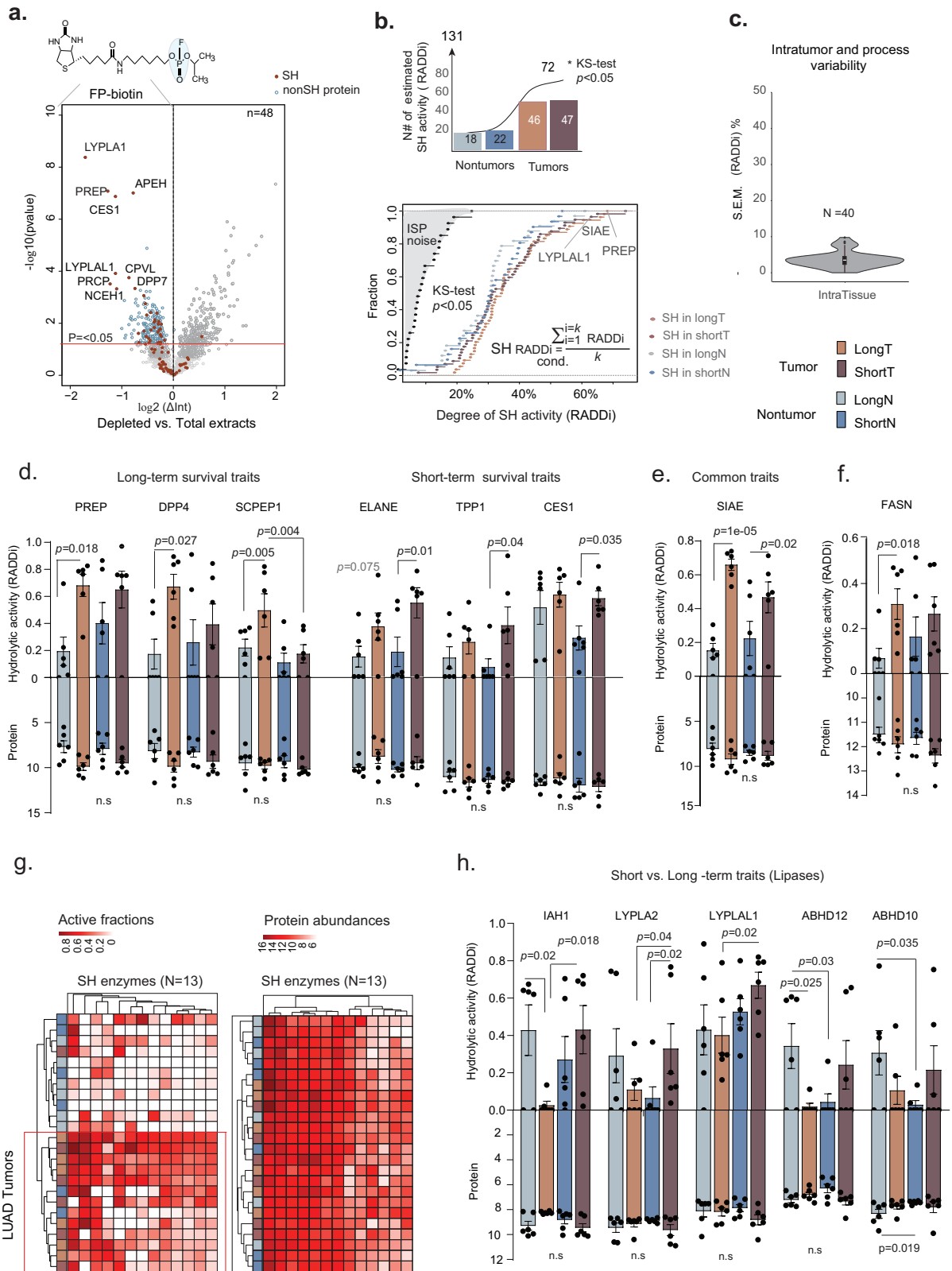

corresponding adjacent tissues (Fig. 4a) as in agreement with dd-ABPP (Fig. 3b, d–g). We performed standard in- gel digestion of the gel region with high fluorescent contrast between control and tumors, and analyzed the peptide mixture (Supplementary Note 1). The gel results were inconsistent as the band resolution was insufficient to relate the detected enzyme elevated in tumors (e.g., PREP, DPP4, ELANE) to a

particular band (gel area: 25–200 kDa, Supplementary Fig. 7a, left). Nevertheless, for enzymes for which conclusive results were obtainable with the gel based ABPP method (i.e., FASN), the results concur (Supplementary Fig. 7a, right).

We then performed standard ABPP measurements on the beads collected in the first experiment. We aimed to compare the amount of

**Fig. 3 | Differential profiling of SH activity via dd-ABPP. a** Quality control of dd-ABPP experiment: volcano plots showing comparison of 24 paired depleted and total extracts recovered from patients' tissue blocks. Red circles indicate SH protein. Blue circles indicate depleted proteins. The one-sided *p*-value from paired *t*-test. **b** Upper plot: Number of enzymes with estimated activity per condition (*N* = 131, right). Solid line is cumulative number of unique enzymes with significantly estimated active fraction. KS-distance test (one-sided *p* ≤ 0.05) per each LUAD condition used to select SH depletion ratios (log2Δlnt) which distribution significantly differ from technical noise (ISP distribution). Lower plot: Cumulative distribution function of active enzyme fractions across conditions. Each dot in the respective color code is the average SH activity (RADDi) within that condition. The *k* is the number of samples. **c** Violin plot showing distribution of S.E.M values (%) calculated between the SHs activity values (*N* = 40 enzymes) obtained from independent intratumor extracts (*n* = 3 technical replicates). Boxplot is median S.E.M. values (middle lines), with first and third quartiles as box edges and whiskers min to

max. **d** The enhanced SHs' activities in comparison tumor vs. nontumor for long- or short- survival subtype, respectively. **e** The SIAE activity in both tumor subtypes. **f** The enhanced FASN activity in tumors of long-term survivals. **g** Unsupervised sample clustering based on 13 tumor hydrolases. Heatmap of SH activity profiles (right) or expression data (left). Manhattan distances used for hierarchical clustering. **h** Degree of activity and total protein levels of serine lipases. Pairwise comparisons of LUAD conditions in (**d**–**f**), (**h**), each plot point representing patient (*n* = 6 per tissue group). GLM (quasibinomial model) statistical analysis of RADDi values in (**d**–**f**), (**h**) accounted for patient age, sex and smoking status and two-sided p-values ≤ 0.05 were considered significant, with no adjustment for multiple comparisons. Barplots in (**d**–**f**), (**h**) are mean values +/− S.E.M representing active fractions (upper panel) and total protein (lower panel) of dysregulated SHs. Confounder nonadjusted analysis in Supplementary Table 1. Legends depict the color of respective conditions. Source data are provided as a Source Data file.

catalytically active enzymes detected on beads with the elevated tumor catalytic fractions of SHs detected with dd-ABPP. After on-bead digestion, 53 of 78 captured enzymes were confidently quantified by standard DDA-MS. We detected six of 13 SHs that classified LUAD tumors with sufficient peptide coverage for spectral counting (i.e. SIAE, PREP, FASN, DPP4, PRCP, ELANE; Fig. 3g). Indeed, the data confirmed increased activity of five key tumor enzymes (SIAE, PREP, FASN, DPP4 and PRCP) in LUAD tumors compared to surrounding tissue, with four showing statistically significant changes (Fig. 4b and Supplementary Fig. 7b)

We further used functional assays with well-designed fluorescent substrates for more specific validation of enzyme of interest. By using the synthetic substrate MAK088 that is efficiently cleaved by prolyl endopeptidases like PREP and DPP4, we found significantly higher measurements of total prolyl endopeptidase activity in tumors when compared to controls (EndoP, Fig. 4c, Supplementary Fig. 7c). We then measured DPP4 activity by including sitagliptin, its specific inhibitor in the assays, allowing us to calculate the DPP4 activity (DPP4, Fig. 4c). Consistent with the above results, a higher increase of DPP4 activity was detected in long-survival LUAD tumors vs. controls, while differences in short-survival vs. control subtypes were smaller (Fig. 4c, Supplementary Fig. 7b, d). The residual activity after DPP4 inhibition could be attributable to other prolyl endopeptidases such as PREP (PREP, Fig. 4c, Supplementary Fig. 7e), which also exhibited high activity in tumor samples, especially in long-term survivors converging with dd-ABPP. Notably, all signals could be specifically attributed to serine protease activities, as demonstrated by PMSF inhibition[55] (Supplementary Fig. 7f).

We further selected ELANE as an important candidate for dd-ABPP method validation as its increased proteolytic activity in LUAD tumors was neither validated by the conventional method (Supplementary Fig. 7g) nor did it correlate with its enzyme relative abundance (rho = −0.04, Fig. 4d). We used the commercial substrate assay MAK246 that is used to detect the activity of ELANE protease. In accordance with dd-ABPP results, the data showed a significant elevation of ELANE activity in tumor samples of both survival types (Fig. 4d, Supplementary Fig. 7h), but with differences related to slightly higher activity in long-term– compared with short-term subtypes. Some divergences between the different chemo-proteomic approaches are however expected, due also to different experimental design of methods and intratumor variability of active enzyme fractions across novel collected tissue cuts.

Finally, we investigated changes in the degree of activity of serine lipases as these were the most prominent biological differences between LUAD tumors of the two survival subtypes. In conventional ABPP on-bead MS quantifications detected three key lipases IAH1, LYPLA2 and LYPLAL1 which successively confirmed enhanced activities via spectral counting in aggressive tumors compared to less-aggressive (Fig. 4e). The extensive validation

experiments corroborated dd-ABPP-SWATH/DIA-MS as a valid proteomic strategy for high-throughput screening of the active fraction of whole enzyme families and targeted strategy that considerably extended the number of SH's for which activity profiles could be calculated.

## dd-ABPP proteomic dataset confirms accelerated protein de-palmitoylation in aggressive LUAD

We further hypothesized that the capture of active enzymes under native conditions would also co-deplete proteins associated to SHs such as enzyme regulatory proteins or protein complex assemblies[3,10], based on consistent co-depletion patterns of a subset of cellular proteins with SHs (Fig. 3a, blue dots). We observed similar co-enrichment of accessory proteins with activity-captured enzymes in conventional ABPP (Supplementary Fig. 5a, right).

To test this hypothesis, we used as a reference the resource of protein-protein interactions (PPIs) in human from an Integrated Interactions Database[56]. We extracted proteins annotated with experimental evidence as 1st degree interactors for 72 SHs for which we had estimated the activity status in LUAD (Fig. 3b), resulting in a set of 2078 proteins (Supplementary Data 5). We compared the 978 proteins co-depleted in dd-ABPP data (*p* < 0.05, two-sided KS-test, Fig. 5a, Supplementary Data 6) with randomly selected protein networks of the same size. The depleted proteome contained a substantially higher number of 1st degree SH interactors compared with the random networks (*p* = 3.0096e[−12], Fig. 5a, right). Remarkably, 95,3% of depleted proteins were annotated as 1st- or 2nd- degree SH interactors (i.e., 258 and 675 interactors, respectively, Fig. 5a). Among depleted 1st degree interactors we observed chaperone binding proteins, regulators of endopeptidase activity, ligases, peptidase inhibitors, and others (Supplementary Fig. 8a).

To corroborate the inferred interactions with physical evidence, we investigated patterns of recurring co-depletion (i.e. quantitative depletion ratios (log2Δint)) of protein interactors with the SH activity profiles across specific conditions. We extracted enzyme-protein pairs based on canonical correlation value (CCA, Methods)[49] between two data sets: i) the combined list of 388 proteins depleted with more stringent criteria in any of the LUAD groups tested (i.e., *p* < 0.01, two-sided KS-test) and ii) the degree of activity of 64 class-descriptive SHs selected by PLS-DA (Supplementary Fig. 5d). We observed 3798 mutual correlations above the specified cutoff (i.e., |cor>0.4 | , *p* < 0.05, *N* = 24, Supplementary Data 7) between the degree of activity of 37 annotated SHs and recurring co-depletion of 146 proteins across LUAD samples. Remarkably, in tumors we observed a strong co-depletion between the active fractions of Plasminogen (PLG) and Fatty Acid Synthase (FASN) and their known interactors related to lung cancer Matrix metallopeptidase-9 (MMP9)[57] and Histone-arginine methyltransferase (CARM1)[58], respectively (Fig. 5b, left). Among selected enzyme-protein pairs, we confirmed 55 interactions (of 21 SHs) with strong

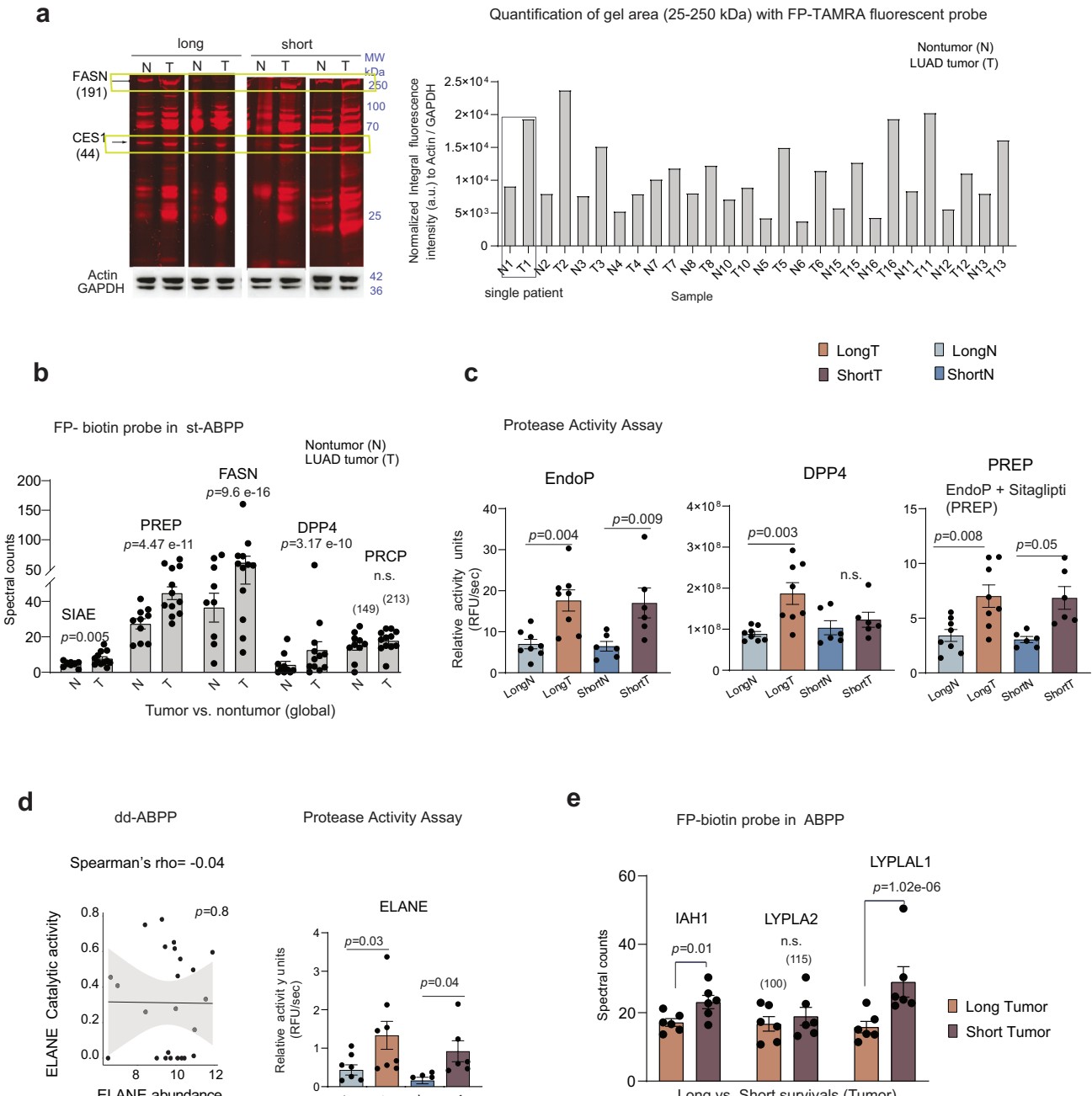

**Fig. 4 | Validation of dd-ABPP-SWATH/DIA MS by orthogonal techniques. a** Left: Labeling of active SHs with TAMRA-FP fluorescent probe. Fluorescent image shows labeled gel-separated proteome from two gels processed in parallel; below the gel image are respective loading controls of β-actin and GAPDH. Corresponding SHs detected are annotated at side of image. Right: Barplot of quantification of fluorescence intensity of gel image normalized to loading controls. **b** Standard ABPP workflow via DDA-MS. The spectral counting (SC) from bead digest of 5 tumor elevated SHs collected in all tumor (T) and nontumor (N) samples. Each point in the barplot represents a single patient, ($n = 10$ for N and $n = 12$ for T). **c** Activity assay for total prolyl endopeptidase activity (EndoP, left). Sitagliptin-dependent portion of total prolyl endopeptidase activity corresponding to DPP4 (DPP4, mid). Residual

activity corresponding to PREP (PREP, right). **d** Left: Spearman rho correlations of degree of ELANE catalytic activity with total ELANE relative abundance level. *p*-value from two-tailed Spearman rho correlations test. Right: Activity assay for ELANE protease. Two-sided *p*-value from one-way ANOVA Sidak's post hoc tests for multiple comparisons in (**c, d**) (patient tissue: $n = 8$ for LongN, $n = 8$ for LongT, $n = 6$ for ShortN and $n = 6$ for ShortT). **e** The SC data for IAH1, LYPLA2, and LYPLAL1 across survival subtypes in tumor tissue ($n = 6$ patient per LUAD group). Two-sided *p*-values from GLM (Poisson regression) analysis of count data in (**b**) and (**e**). Total number of SC in brackets for statistically non-significant cases. Data in (**b–e**) are presented as mean values +/− S.E.M. Source data are provided as a Source Data file.

experimental evidence, significantly exceeding the expected random number (i.e., $p = 1.048e-07$, 11.4 interactions/average; Fig. 5b, right). These results demonstrate that a significant fraction of proteins co-depleted with SHs from the contextual proteome engages into molecular complexes, thus presenting potentially valuable data resources

to conceive functional links between SHs and associated proteins in LUAD.

Following this rationale, we integrated 388 co-depleted proteins and 64 class-relevant SHs to select the consensus model with maximum ability to separate between LUAD groups for clinical utility.

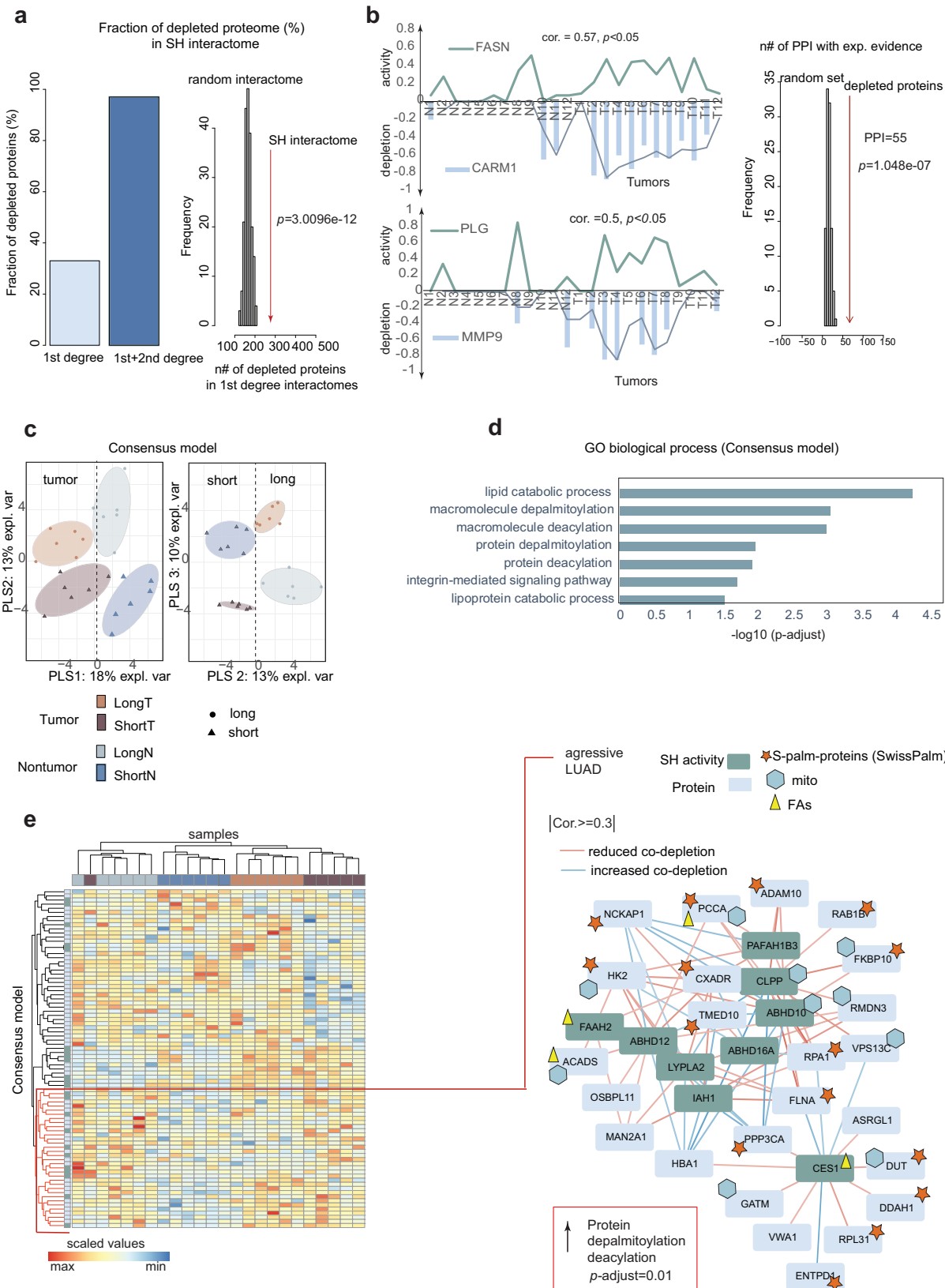

Astonishingly, the model of 82 classifiers that incorporate the activity data of 23 catalytic SHs with the depletion ratios of 59 proteins succeeded in retrospectively separating the LUAD groups with a prediction error rate below 10% estimated by 5-fold cross-validation schema (Fig. 5c, Supplementary Fig. 8b). The Gene Ontology (GO) enrichment analysis integrating enzymes and co-depleted proteins in the discriminant model suggested significantly accelerated lipid catabolic process, macromolecule de-palmitoylation, macromolecule de-acylation and protein de-palmitoylation (FDR ≤ 0.01) via enrichment of several palmitoyl hydrolases (i.e. ABHD12, LYPLA2, PAFAH1B3, ABHD10, ABHD16) and the mitochondrial proteins (i.e. PCCA, ACADS) involved in fatty acid (FA) processing (Fig. 5d, e, sub-network,

**Fig. 5 | Functional links between catalytic enzymes and significantly often co-depleted proteins reveal differences in protein palmitoylation status in aggressive LUAD subtype. a** Left: Fraction of depleted proteins annotated as either 1st (sky-blue) or 2nd degree (blue) SH interactors. Right: Frequency of depleted proteins found in 200 random interactomes. Red arrow indicates the number of depleted proteins in the real SH interactome ($N = 258$), which significantly exceeds the number of binary SH interactors compared to random sets. One-tailed $p$-value calculated by pnorm-test. **b** Left: Line charts show activity and depletion of respective "enzyme-protein" pairs that correlated between samples and demonstrated physical PPI evidence. Two-sided $p < 0.05$ from Correlation coefficient significance test ($n = 12$ paired samples). Green and blue lines depict either average SH activity or average protein depletion across samples, respectively. The x-axis correspond to adjacent-tissue (N) or tumor (T) samples. Right: Histogram depicts the number of physical interactions for 100 random subsets generated by sampling of quantified proteins. Red arrow is the number of confirmed physical interactions found within subsets of co-depleted proteins and SHs displaying significant

pairwise correlations ($N = 55$). One-tailed $p$-value from pnorm-test in R. **c** Left panel: Sample classification based on consensus model with 75% confidence ellipses and PLS-DA variates 1, 2 and 3. Consensus model obtained by block PLS-DA data integration and successive tuning steps (i.e., 82 classifiers), integrating 23 respective SHs and 59 significantly often co-depleted proteins (Supplementary data Fig. 8c). **d** Functional enrichment analysis of classifiers from consensus model in PANTHER Overrepresentation Fisher Exact Test. FDR adjusted $p$-value for multiple testing. *Homo sapiens* database is used as reference. **e** Left panel: Unsupervised sample clustering based on final consensus model. Manhattan distances used for hierarchical clustering. Right panel: Network depicts 9 hydrolases (i.e., SH in green) and subset of co-depleted proteins (i.e., DP in blue) including those with observed S-palmitoylation motif (asterisk in orange). The correlation between data levels extracted after performing block PLS-DA (Sparse Generalized Canonical Correlation Analysis). Blue hexagon and green triangle depict co-depleted proteins either with mitochondrial localization or involved in FA metabolism, respectively. Source data are provided as a Source Data file.

Supplementary Fig. 8c). Strikingly, 29 out of 59 depleted proteins (i.e. 49,2%) in the consensus model were reported as endogenous substrates of palmitoyl hydrolases - S-palmitoylated proteins (e.g., ENTPD1, ADAM10, CXADR, RPL31, HK2, NCKAP1, RAB1B, Fig. 5e, network), substantially exceeding the 7% palmitoylation rate observed in the entire human proteome[59]. Thus, dd-ABPP has demonstrated the capacity to concurrently measure the activity status of LUAD specific SHs and select their putative substrates from the contextual tissue proteome, offering the user a more holistic view of the recorded data compared to the standard method (Supplementary Note 2).

## FA analysis shows an excess of palmitate and its mono-unsaturated metabolites, palmitoleic and oleic acids, in the aggressive phenotypes

Recent studies have demonstrated that lipid remodeling via changes in FA composition[60,61], status of protein palmitoylation[18,62], and dietary uptake of palmitate[63] contribute to general cancer metastatic capacity, but not cancer incidence[64,65]. To experimentally test our hypothesis that aggressive tumors in LUAD exhibit lipid remodeling via an increased rate of lipolysis and protein de-palmitoylation (Fig. 5d, e), we analyzed the sections of the ten tumors (i.e., 5 per each LUAD subtype, Supplementary Fig. 3) for a panel of 25 saturated, monoenoic, and unsaturated (C8-C22) FA levels with a primary focus on palmitic (hexadecenoic, C16:0) acid metabolism. We performed an initial profiling of FA species using liquid chromatography high-resolution mass spectrometry (LC-HRMS) and detected peak areas corresponding to 14 annotated FA species according to LC-HRMS coordinates matched to an in-house database generated from pure authentic standard (AS) solutions of FA analyzed under the same analytical conditions (Supplementary Data 8, Supplementary Note 3). While no changes were detected in relative comparisons of peak areas corresponding to palmitic acid, we found a statistically significant increase in three species corresponding to eicosapentaenoic acid (C20:5), palmitoleic acid (C16:1), and oleic acid (C18:1) in the aggressive tumors (Fig. 6a). Interestingly, the endogenous levels of palmitoleic and oleic acid are primarily produced via palmitate desaturation[66] by the rate-limiting enzyme Stearoyl-CoA Desaturase (SCD1)[61] and represent the major endogenous metabolites of palmitic acid[67]. We confirmed a significant increase in the abundance of SCD1 enzyme by western blot indeed confirming increased saturation of palmitic acid in tumor tissue of patients with poor survival ($P = 0.006$, Fig. 6b, Supplementary Note 1). Finally, we quantified palmitic acid and its metabolites, palmitoleic (C16:1) and oleic (C18:1) acids (Fig. 6c–e) using calibration curves AS solutions and the stable isotope-labelled internal standards (IS) to obtain the absolute concentrations of these FA in tumor subtypes (Supplementary Data 8, Supplementary Note 4). Our analysis revealed a statistically significant increase in palmitic acid concentration in aggressive compared to less aggressive LUAD tumors, confirming our

hypothesis that excess palmitate contributes to tumor aggressiveness (Fig. 6c). Moreover, increased concentrations of major endogenous metabolites of palmitate desaturation, palmitoleic and oleic acids, also indirectly confirmed palmitate excess, although palmitoleic acid levels did not reach statistical significance ($p = 0.09$, Fig. 6c, d). We also found a significant increase in polyunsaturated eicosapentaenoic acid (C20:5) in aggressive tumors by targeted analysis, consistent with the initial FA screen (Fig. 6a, f). Our results were to some extent consistent with previous studies demonstrating the process of rapid desaturation of FAs in LUAD expansion[68]. Collectively, the dd-ABPP analysis of LUAD tumor and adjacent tissue pairs in conjunction with targeted lipidomics of the same specimens indicated a link between lipase activity levels, protein palmitoylation status, and the increased presence and desaturation of palmitate in tumors at an advanced stage of disease, a link that may in turn promote LUAD cancer progression (Fig. 6g).

## The degree of SH catalytic activity associates with LUAD molecular composition

Finally, we sought to demonstrate how our multimodal dataset in combination with other quantitative omics readouts can guide molecular hypotheses relating to enzyme regulation in the specific cancers (Fig. 1b). As LUAD conditions display no substantial change in SH protein levels, we reasoned that dozens of their dysregulated active fractions were linked to tumor heterogeneity or molecular composition affected by external, environmental factors. Tobacco smoke is the most common LUAD etiology[69] as the lung epithelia is directly exposed to cigarette- or environmental tobacco smoke. As most of our patients were self-reported smokers (13/16) and their tumors revealed the enhancement of the protein sensors of toxic metal exposure (i.e. ALAD, Fig. 2f)[70], we further investigated patient tissues for abundance of trace elements concentrated in tobacco smoke. We applied Inductively coupled plasma (ICP)-MS[71,72] and quantified the levels of the nine toxic metals identified in tobacco smoke by the Food and Drug Administration (FDA) in patient tumor and adjacent tissues. Specifically, we measured the levels of Cadmium (Cd), Chromium (Cr), Lead (Pb), Nickle (Ni), Selenium (Se), Mercury (Hg), Arsenic (As), Cobalt (Co), and Aluminum (Al), (www.fda.gov/tobacco-products/; Supplementary Data 9). We correlated trace element profiles with the active fractions of 72 catalytically active SHs detected in LUAD samples (Fig. 3b). Of nearly 700 computed correlations, we found 12 positive and 7negative significant correlations (Fig. 7a), while also considering the confounders of age, gender, and smoking status. We found that the enhanced proteolytic activity of ELANE, in tumors, correlated with lung concentrations of Cr (rho = 0.73, $p = 0.01$) and Al (rho = 0.65, $p = 0.03$) (Supplementary Fig. 9a), while tissue levels of known ELANE inhibitors, measured simultaneously within contextual LUAD proteomes, correlated negatively with these respective metal ions (Fig. 7b). Indeed, the protease inhibitors have been previously

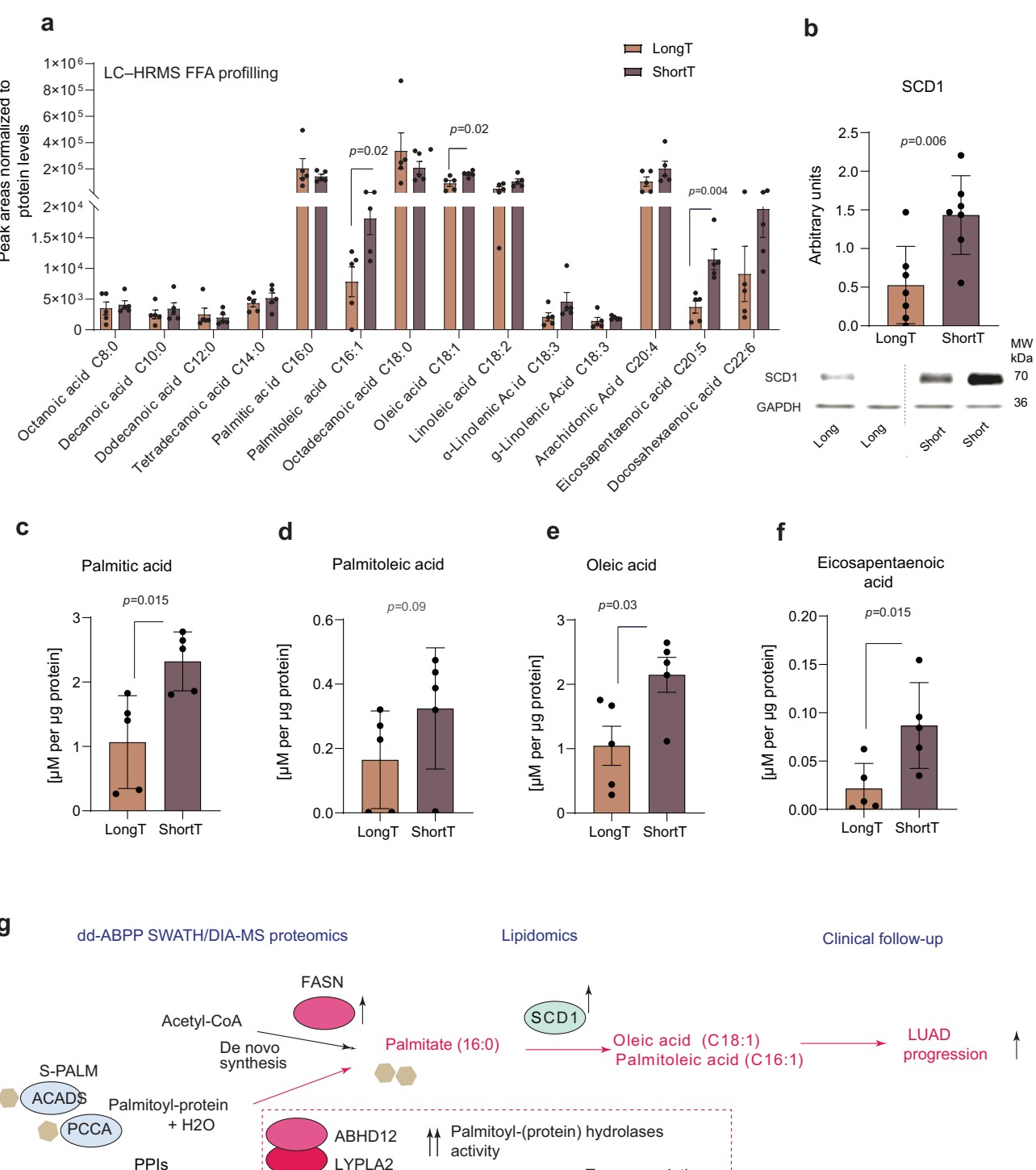

**Fig. 6 | FA analysis reveals the excess of palmitate and its monounsaturated metabolites in the tumors of the aggressive LUAD phenotype. a** Relative comparison of selected free fatty acid levels obtained through a full scan untargeted LC-HRMS profiling. FA annotation was done by the accurate mass and retention time (AMRT) matching against standard database. No MS/MS spectra are available due to well-known low fragmentation efficiency of FAs in negative ionization mode. Barplots with data points are levels of annotated FA in LUAD tumors of long– and short-survival patients and represent mean values +/− S.E.M. Two-sided *p*-value < 0.05 considered significant and corresponds to a Student's *t*-test (*n* = 5 samples per LUAD T subtype). **b** Protein expressions of Stearoyl-CoA Desaturase (SCD1), (*n* = 7 biological replicates per group). GAPDH is used as loading control. Boxplot represents mean ± S.E.M expressed as arbitrary units normalized to the mean of long-term survival subtype. The *p*-value from two-sided Student's *t*-test. **c**–**f**, the absolute tumor concentration of the FAs of interest, hexadecenoic/palmitic, palmitoleic, oleic and eicosapentaenoic acids, respectively, determined through the internal standard spike using isotopically labeled standards and multipoint calibration curves. The *p*-value corresponds to two-sided Wilcoxon signed rank test (*n* = 5 biological samples per LUAD subtype). **g** Schematic of proposed lipid remodeling in aggressive LUAD. dd-ABPP analysis revealed increased catalytic fractions of serine lipases interacting with palmitoylated lipoproteins in aggressive tumors. Increased lipase activities degrade substrate lipoproteins into naked proteins and palmitoyl residues, triggering lipid composition remodeling in aggressive tumors. PPIs correspond to protein-protein interactions. Data in (**a**), (**c**–**f**) are presented as mean values +/− SEM. Source data are provided as a Source Data file.

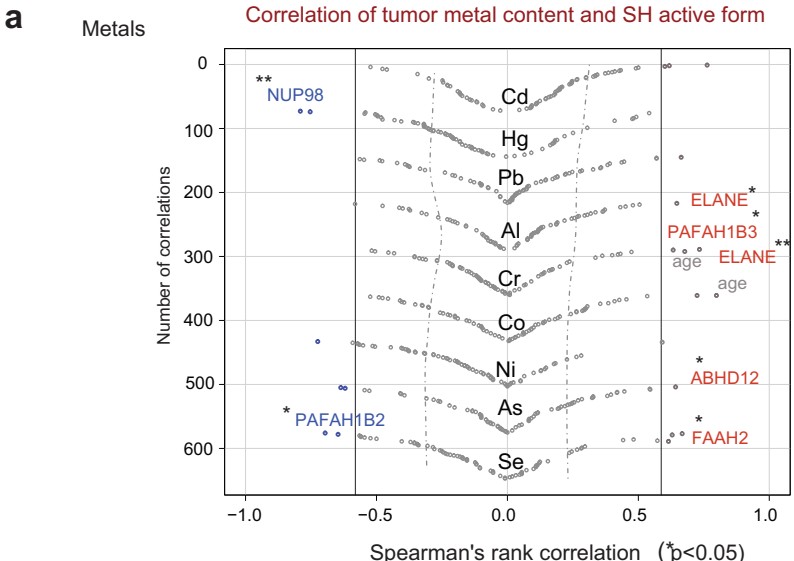

**a** Metals

Correlation of tumor metal content and SH active form

**b** Endogenous protease inhibitors of ELANE

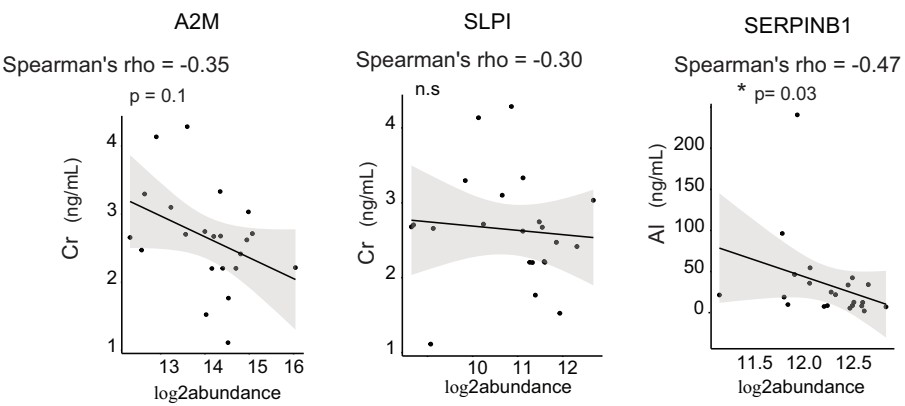

**c** Gene   The co-occurrence of *KRAS* oncogene with SH's gene mutations in TCGA

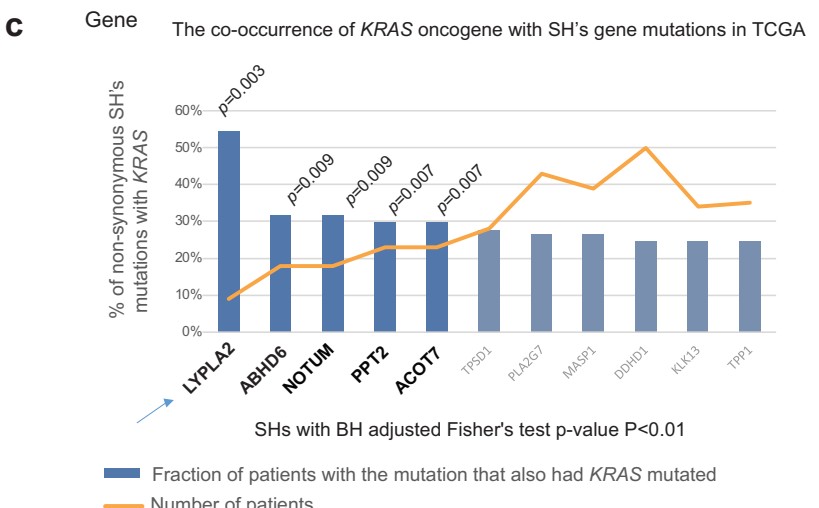

reported as key regulators of ELANE activity in acute lung exposure to smoke[73,74]. In addition, we found that the small nuclear protease of the Nup98 · Nup96 (NUP98) complex shows a significant negative correlation of its active fraction with the Hg ion concentration (rho = −0.75, $p = 0.007$), consistent with a significant loss of its proteolytic activity in half of the LUAD tumors tested (i.e., 6 out of 12, Supplementary

Fig. 9b, c). Most interestingly, even three tumor lipases, ABHD12, FAAH2, and PAFAH1B3, which are associated with patient low life expectancy within discriminative signature, showed statistically significant correlations of their catalytic fractions with the tumor concentration of the divalent metal ions As, Se, and Cr, respectively (ABHD12 and As: rho=0.64, $p = 0.03$; FAAH2 and Se: rho = 0.67,

**Fig. 7 | Dysregulated active enzyme fractions correlate with FDA-identified toxic metals in tobacco smoke and display genetic polymorphism relevant to KRAS-LUAD oncogene. a** Scatter plot represents Spearman rho correlations (rho) of degree of SH catalytic activity with tumor content of nine respective metals. Correlation significance depicts two-sided correlation test *p-value* < 0.5. (rho> |0.6 | ). **b** Representative scatter plots with 95% confidence intervals of measured content of endogenous ELANE inhibitors annotated in Merops, with lung tissue concentration of Cr and Al (ng/ml). These metals highly correlated with the proteolytic fraction of ELANE. **c** Co-occurrence of SHs and *KRAS*

mutations in TCGA *KRAS* mutated cancers. Barplots show the fraction of patients (%) with the SH mutation that also had mutated *KRAS*. For each SH gene, we assessed the number of patients with at least one non-synonymous mutation within a gene. Arrow depicts *LYPLA2* that displays high co-occurrence of gene mutations with *KRAS* oncogene and was detected with increased catalytic fractions in aggressive compared with less-aggressive LUAD disease in our study. BH-adjusted Fisher's test *p*-value for the co-occurrence of two mutations. SHs detected in the current study and with *p*-value < 0.01 are shown on the graph. Source data are provided as a Source Data file.

$p$ = 0.024; PAFAH1B3 and Cr: rho = 0.063, $p$ = 0.04; Fig. 7a). Notably, no significant correlation was observed between metal ions and the protein abundance of the respective lipases in lung tumors (Supplementary Fig. 9d, e).

Finally, we investigated the genetic polymorphisms of SHs enzymes within The Cancer Genome Atlas (TCGA) datasets (http://cancergenome.nih.gov/). At least 25% of cancer drivers are inherently palmitoylated[18] including multiple isoforms of the RAS GTPases protein (KRAS, HRAS and NRAS) that are the most frequently mutated oncogenes in human cancers, including in LUAD, (i.e., 30% of LUADs contain somatic *KRAS* mutation)[1]. They attain oncogenic properties only in a de-palmitoylated molecular state that supports a protein subcellular translocation[12,75,76]. To check for the co-occurrence of the *KRAS* oncogene with non-synonymous mutations within an SH gene, we used the cohort of all *KRAS* mutated cancers founded in the TCGA dataset (i.e., 466 *KRAS* mutated out of 6953 cancers) including a smaller cohort of 72 *KRAS* mutated LUAD samples (i.e., out of 230 LUAD cancers). Among more than 250 verified SHs genes within all *KRAS* mutated TCGA cancers, we found the highest co-occurrence (≥30%) of genomic alterations with *KRAS* for four palmitoyl and palmitoleoyl hydrolases (i.e., *LYPLA2, NOTUM, PPT2,* and *ACOT7,* BH adjusted Fisher's *P* < 0.01, Fig. 7c), including the here dysregulated LYPLA2, the major indicator of LUAD aggressiveness. These metabolic vulnerabilities might be further explored since, for instance, the inhibitors of serine lipases are already promising targets in other *RAS*-mutant carcinomas[1,76].

We also noted that *NUP98* was selected as the top mutated SH genes in *KRAS* mutated LUAD in TCGA (Supplementary Fig. 9f) and thus we suppose that the lung accumulation of mercury from environmental exposure, together with genetic variants of its protein sequence, most likely could contribute individually or jointly to the loss of NUP98's proteolytic ability and its tumor suppressor function[77,78]. Collectively, the enhanced catalytic fractions of several SH enzymes (i.e., ELANE, LYPLA2, ABHD12, NUP98) across all LUAD conditions were not compensated by their biological abundances, but most likely make a genetic and environmental contribution to aberrant hydrolytic activities observed in aggressive LUAD tumors.

## Discussion

Here, we introduce a method of activity-based proteomics that, builds on the previous approaches[23,25], and applies experimental design through selective chemical tagging of active enzymes and their subsequent depletion from the reference tissue proteomes. The approach takes into consideration that a chemical-probe serves as an affinity handle for protein-enzyme interactions in the contextual proteomes[3,10]. Thus, this method generates multiple levels of complementary data in the same operation: the total enzyme quantities, the active enzyme fractions, the levels of functionally proximal endogenous proteins (e.g., inhibitors, substrates, regulatory proteins), and context-dependent protein interactors with enzymes.

We apply this methodology to the cohort of specimens that had been collected for histopathological diagnosis of patients with the same advanced LUAD stage and who have over the years displayed different clinical outcomes. Retrospective analysis revealed the unique

molecular signatures of the activity profiles of 23 SHs and 59 contextual tissue proteins that discriminated aggressive LUAD tumors (Fig. 5c−e). The most prominent characteristic of an aggressive tumor phenotype was the enhanced activity of palmitoyl protein hydrolases (ABHD12, LYPLA2 and ABHD10)[17] associated with a large number of S-palmitoylated proteins within discriminative signature (Fig. 5e). Subsequent FA lipidomic analysis of the respective tumors detected the excess of palmitic acid and its mono-unsaturated metabolites and thus confirmed increased palmitate levels in aggressive LUAD. We experimentally validated our findings with three different orthogonal analytical methods. Our results further suggest that dysregulated hydrolase activities are often not compensated by biological amounts of enzymes. Most likely, the altered fractions of the active SHs are related to peculiarities of the enzyme protein sequence, to disease-associated somatic mutations of the SH gene (*LYPLA2, NUP98*), a distorted enzyme-inhibitor equilibrium (ELANE), or a distorted tissue molecular composition (Fig. 7). We believe that workflows based on chemical affinity enrichment of protein forms of interest (e.g., catalytic enzymes, PTM-modified proteins) would further benefit from standardizing their measurements to their temporal and spatial context to obtain a more complete biological picture of condition of interest. Although this study focused on the SH activity profiles in LUAD tissues, we established a workflow that could be readily adopted for large-scale screening of other enzyme families for which chemical probes are available[21], including matrix metalloproteases[79] and cysteine proteases[80], prominent targets in cancer diagnosis, staging, and therapeutic intervention, or inflammation[81] and host-pathogen interactions in infectious diseases[82,83]. Additionally, we envisage that the catalytic activity level control of the enzyme measured by our method in combination with quantitative omics readouts, for instance, post-translational protein modifications, could guide further discovery of mechanisms of enzyme regulation (Fig. 1b).

Notwithstanding the high level of data comprehensiveness of the enzyme family of interest generated here, limitations of this concept in chemical proteomics should be acknowledged. First, as expected, the method shows decreased sensitivity toward the small variances of enzymatic activities (<10%) across tested samples and a certain variability of intra-tissue individual measurements. However, whole tissue sections that were used for histochemical staining were also used for dd-ABPP analysis, so user can optionally perform cell-count adjustment for percentage of cell of interest for the enzymes that show individual variability (e.g., SIAE, FAAH2). Second, specificity of method is compromised by the strength of enzymes binding with likely interactors from surrounding proteomes, which is limitation inherent to high-throughput proteomic workflows based on chemical affinity binding[84]. To increase method specificity, that is; to specifically query the activity profiles of expected members of the enzyme family of interest, we employed here the targeted DIA spectral libraries[50] based on prior information of the peptides of interest.

Taken together, these results represent an alternative concept in activity proteomics, aiming at a streamlined experimental setup for the study of enzyme catalytic activity and its regulation mode, and placing this information in the context of the sample proteome. We believe this approach will provide essential information to map the molecular

landscape of tumors, support the development of diagnostic tools, and advance personalized cancer therapy.

## Methods

### Study design

Clinical cohort comprised patients who had undergone surgical lung resection at the University Hospital Zurich (UHZ) between 2005 and 2013. Informed consent was obtained from each patient. The Cantonal Ethics Committee Zurich (KEKZH) has approved all procedures involving human material and each patient has signed an informed consent form (KEK-ZH-No. 2020-02566). The study, design, and conduct complied with all relevant regulations governing the use of human subjects and were conducted in accordance with the criteria of the Declaration of Helsinki. The tissues were collected routinely on the date of the patient's surgery (OP date) as either tumor or surrounding, non-neoplastic lung parenchyma (Supplementary Data 3). For the complete study, we included 32 lung tissues, 16 tumors and 16 matched non-tumor tissues collected from patients before therapy administration. The cohort included an equal number of long– and short-term survival patients, with survival superior to 5 years or 1 year, respectively. 28 collected tissue samples were embedded in an OCT compound, while 4 samples were available in FF form. Samples were handled in the same manner, frozen, and stored at −80 °C until used (Table 1). For all samples in the clinical cohort, analysis of independent 8 μm sections was performed according to the 6th TNM classification of malignant tumors, hematoxylin and eosin (H&E) stained and morphologically reviewed at the time of LUAD diagnosis. In summer 2017, 24 of 32 samples, representing 424 long-term and 313 short-term tissue cryosections from the discovery cohort, were selected for dd-ABPP-SWATH/DIA-MS analysis. All cryosections from each sample were prepared on the same day by the same person using a HM 560 cryostat (Microm) at a temperature of −20 °C. The 30 μm sections, corresponding to approximately 60 mm3 of tissue volume each, were placed in the Eppendorf tubes for further OCT cleaning. The same cryosectioning procedure was used for experimental validation of the targets discovered by dd-ABPP proteomics in 2020 and for subsequent lipidomic and metallomic analysis in 2022. We collected a new set of cryostat sections of the primary cohort of 24 samples, as well as additional sections of 8 new tissues from 4 carcinoma patients selected either as long– or short survivals (i.e., 1080 collected cryo-sections, validation cohort, Supplementary Fig. 3).

### Processing of OCT-embedded and FF tissues for dd-ABPP-SWATH/DIA

To remove the water-soluble OCT medium(www.cellpath.com/oct-embedding-matrix) and obtain high quality peptide digests for LC-MS analysis, while simultaneously preserving physiological enzyme activity for probe labelling, we used a two-step clean-up procedure. The first step clean-up was performed to wash away OCT medium prior to probe labelling. Three consecutive washes of cryostat sections were performed in Eppendorf tubes with 500 μL of ice-cold phosphate-buffered saline (PBS, pH 7.4) (#P3813 Sigma Aldrich), and the tubes containing tissue sections in PBS were placed in a thermomixer at 200 x g at 25 °C for 2 min each time. Thereafter, the samples were centrifuged at 800 x g at 4 °C for 45 s to remove the water-soluble OCT from the tissue surface before homogenization with a tissue grinder (#D8938, Sigma-Aldrich) in 500 μL PBS. The sample extracts were sonicated with 1 min intervals (60% output, 80% duty cycle, 20 times, Heat Systems Ultrasonics, Inc.) and centrifuged for 5 min at 4 °C at 800 x g to remove any non-soluble material. The protein concentration was measured by Pierce BSA Protein-assay kit (#23225, Thermo-Scientific).

Probe labelling. A final volume of 2 mL of the diluted extract with total protein concentration between 0.5–1 μg/μL was divided into two equal aliquots and used for probe labeling (positive control) or blank dimethyl sulfoxide (DMSO) incubation (negative control). The labeling was performed with FP-probe – 6-N-biotinylaminohexylisopropylphosphorofluoridate- purchased from Toronto Research Chemicals (CAS number 353754-93-5, TRC) and reconstituted at 5 mM in DMSO. The FP-probe selectivity for SHs was previously reported[84] and the effective probe concentration of 5 μM tested for labeling and used[25]. After 2 h of incubation with the FP-probe at room temperature (RT), Triton X-100 (#T8787, Sigma Aldrich) was added to 1% (v/v) followed by 1 h of rotation at 4 °C. Samples were desalted with PD Midi-Trap G-25 (GE Healthcare), then Sodium dodecyl sulfate (SDS) solution (Sigma) added at 0.5% (w/v) and the samples incubated for 8 min at 90 °C. Next, the samples were combined with 67.5 μL of pre-washed streptavidin coated agarose beads (#20347, Thermo Pierce), diluted with PBS to a final volume of 2.4 mL and rotated for 1 h at RT to allow avidin-bead binding of FP-biotin-labeled enzymes. The beads were pulled down at 200 x g for 3 min at 4 °C and stored for further "on-beads" digestion by a conventional ABPP approach.

The second step clean-up and protein digestion. After tagged-enzymes depletion, the 600 μL of collected supernatants were further processed to remove any residual impurities that could interfere with subsequent LC-MS analysis of peptides. The proteins were precipitated by adding 200 μL of trichloroacetic acid (TCA) to a final 25% (v:v) concentration. After 1 h, the protein pellets were collected at 21,000 x g for 15 min and washed three times with ice-cold acetone. Prior to digestion, the protein pellets were dissolved in 200 μL of 8 M urea with 50 mM Ammonium Bicarbonate (AMBIC, pH 7.8) by vigorous vortex, reduced at room temperature for 30 min with 5 mM tris(2-carboxyethyl)phosphine hydrochloride (TCEP, #C4706, Sigma Aldrich) and alkylated in dark for 30 min with 10 mM of iodoacetamide (IAA, #I1149, Sigma Aldrich). Prior to digestion, all the samples were diluted to the 1 M urea concentration with 50 mM AMBIC and overnight digestion performed with a 1:20 ratio of trypsin (#V5113, Promega). The following day, the reaction was stopped by adding formic acid to a final 5% (v/v) and the peptide digests were stored at −80 °C until SWATH/DIA-MS analysis.

### Conventional on-bead ABPP sample processing

The beads collected from tissue extracts after labeling with the probe (positive control beads) or with blank DMSO incubation (negative control beads) were washed three times with 1% SDS (w/v), three times with 6 M urea and three times with PBS (each wash rotating for 5 min at RT at a 2.4 mL final volume followed by centrifugation for 3 min at 1400 rpm at 4 °C). The enriched proteomes from positive and negative control beads were reduced for 1 h with 5 mM TCEP (#C4706, Sigma Aldrich) and alkylated for 1 h with 10 mM IAA (#I1149, Sigma Aldrich). Next, the beads were washed once with 1 mL of 50 mM AMBIC, reconstituted in 400 μL of 1 M urea in 50 mM AMBIC and an on-beads digestion performed for 12 h at 37 °C by adding 6 μL of trypsin (V5113, Promega). Thereafter, samples were centrifuged for 10 min at 17,000 x g at 4 °C and FA added to the peptide samples to a final 5% (v/v). Samples were stored at −80 °C until analysis.

### MS analysis and data processing of LUAD proteomes

Analysis of the samples with LC-MS. Digested peptides were cleaned up prior to LC-MS analysis by MicroSpin Columns C18 (#SEM SS18V, Nest Group Inc., Southborough, MA) at low-speed centrifugation (i.e., 400 g for 2 min) and by five successive washing steps in 0.1% aqueous FA with 2% acetonitrile (ACN) to remove undigested proteome and sample impurities. The purified peptides were solubilized in 25 μl (on-bead peptides digest) or 100 μl (in-solution peptide digest) of 0.1% aqueous formic acid with 2% ACN. Retention time (iRT) peptides were added (RT-kit WR, Biognosys) in equal 1 pmol/μL amount to each sample prior to MS injection. All MS data were acquired on a TripleTOF 5600 mass spectrometer equipped with a NanoSpray III source with a heated interface (AB Sciex, Concord, Ontario, Canada). The peptide

samples were injected onto a C18 nanocolumn with PicoTip emitter (New Objective, Woburn, MA, USA) packed in-house with 3 μm 200 Å Magic C18 AQ resin (Michrom BioResources, Auburn, CA, USA). The separation was performed on a NanoLC-Ultra 2D Plus system (Eksigent–AB Sciex, Dublin, CA, USA) with a constant flow rate of 300 nL/min and an oven temperature of 70 °C with a linear gradient from 2 to 35% B (0.1% FA in ACN) either over 60 or 120 min depending on the acquisition method. The nano-LC-MS was operated by Analyst TF 1.5.1 software (AB Sciex). Electrospray ionization was performed in positive polarity at 2.6 kV and assisted pneumatically by nitrogen at 20 psi.

**DDA and DIA sample acquisition.** A part of the samples (51 MS injections from full tissue extracts or on-bead tissue digests) was acquired via bottom-up proteomic acquisition in DDA mode to generate a library of peptide transitions from LUAD tissue samples (LUAD library). Tandem mass spectra (MS/MS) were recorded in "high-sensitivity" mode over a 50–2000 *m/z* range with a resolving power of 30,000. The MS/MS spectra acquisition was triggered by DDA mode consisting of a full scan of 250 ms followed by 20 MS/MS-dependent acquisitions of 50 ms each. Collision-induced dissociation (CID) was used to induce fragmentation with dynamic collision energy (i.e., rolling collision energy). DDA selection of the precursor ions was as follows: the 20 most intense ions above a threshold of 50 counts, charge state from +2 to +5, isotope exclusion of 4 Da and dynamic precursor exclusion of 8 s leading to a maximum total MS duty cycle of 1.15 s. External mass calibration was performed by 100 fmol solution of β-galactosidase tryptic digest standard.

The intratumor tissue samples (three pairs of total– and depleted tissue extracts) were acquired in standard SWATH/DIA-MS mode by using 32 fragment ion spectra (32 SWATH windows)[38] with an accumulation time of 100 ms for each and for the precursor scans, resulting in a total cycle time of 3.3 s. The SWATH windows of 26 Da were overlapped by 1 Da covering the range from 400 to 1200 *m/z*. A reverse phase peptide separation was performed with linear nano-LC gradient of 120 min as described above.

For the clinical cohort, accounting for 48 sample injections, we employed SWATH/DIA-MS mode with short LC gradient time (i.e., 60 min per sample injection) suitable for large sample cohorts. The quadrupole settings were optimized to cover the range from 400 to 1200 *m/z*, with the selection of 64 variable width windows overlapping by 1 Da as described earlier[85]. Reverse phase peptide separation was performed at 300 nL/min flow rate and a linear nano-LC gradient over 60 min. An accumulation time of 50 ms was used for 64 fragment-ion scans operating in high-sensitivity mode. At the beginning of each SWATH-MS cycle, a 250 ms TOF MS scan (precursor scan) was acquired at high-resolution mode, resulting in a total cycle time of 3.45 s. The collision energy for each window of both SWATH sliding-window scheme was determined according to the calculation for a 2+ window-cantered ion with a spread of 15. The complete set of MS raw data files is deposited on the PRIDE archive with Project accession: PXD019357.

**Analysis of DDA data from LUAD samples.** DDA raw data files (.wiff) were centroided and converted into.mzXML by openMS. The converted data files were searched using X! TANDEM Jackhammer TPP (2013.06.15.1 - LabKey, Insilicos, ISB) and Comet version "2016.01 rev. 3" against the ex_sp_9606_decoy.fasta database (the reviewed canonical Swiss-Prot complete proteome database for *human*, released 2014-01-24) appended with common contaminants, reversed sequence decoys[86] and iRT peptides. The search parameters included trypsin digestion and allowed for 2 missed cleavages. The modification list included 'Carbamidomethyl (C)' as static and 'Oxidation (M)' as variable modification. The mass tolerances were set to 50 ppm for precursor-ions and 0.1 Da for fragment-ions. The identified peptides

were processed and analyzed with the Trans-Proteomic Pipeline (TPP v4.7 POLAR VORTEX rev 0, Build 201403121010) using PeptideProphet[87], iProphet[88] and ProteinProphet scoring. Spectral counts and peptides for ProteinProphet were filtered at FDR of 0.010045 mayu-protFDR (=0.992284 iprob).

**Generation of phylogenetic tree of SH family members.** We generated a list of metabolic serine hydrolases (mSHs) and serine proteases (SPs) by extensive literature and database search. All SHs that hydrolyze peptide bonds with trypsin/chymotrypsin/subtilisin enzyme activity or that participates in cleaving the terminal peptide bond (e.g., including prolyl endopeptidase), here, are classified as SPs. SHs that cleave ester, amide, or thioester bonds in proteins or other biological molecules were classified as 'metabolic' SHs. We used Merops[89], Uni-Prot (https://www.uniprot.org, Release 2020_01), PANTHER[90], InterPro[91], Superfamily databases and previously annotated lists[8]. We also included 26 proteins that were not classified (non-annotated cases, Source data for Supplementary Fig. 1) and that showed frequent enrichment with FP-probe and had structural domains of the SH family according to the Superfamily database[92]. For example, 26 proteins included three proteasome subunits (e.g. PSMA6), that were not classified as SH but were included in the analysis based on their reported enrichment by SH-specific probe, also previously reported[23]. These cases, indicated as "others", account for 7.5% of all cases. We catalogued a total of 335 proteins (including 15 inactive enzymes): 175 serine proteases, 134 metabolic serine hydrolases and 26 non-annotated cases. Specifically, 294 proteins were documented as SHs, including SHs with putative activity or SHs for which catalytic activity was not directly demonstrated. (e.g., human Haptoglobin (HP) classified in Merops, Serotransferrin (TF), Merops S60.972). We also included PLD3 that, according to a previous study, contains a Serine nucleophile (corresponding to Ser 109 in the endonuclease)[93] and CD163 annotated as serine protease by PANTHER Class ID PC00203. The table of 335 enzymes and the source of its associations to the SH superfamily is available (Source data Supplementary Fig. 1c, d). The corresponding phylogenetic tree depicting 327 of the 335 SHs (Supplementary Fig. 1c, d) was generated by using protein SH sequences as FASTA format with Clustal Omega[94] to generate STOCKHOLM 1.0 alignment format. The web tool available at https://itol.embl.de/ was used for visualization of phylogenetic tree.

**Development of SH semi-specific assay library.** In order to increase the sensitivity of our method we combined the MS assays for the SHs available from the LUAD specific library (in-house library of shotgun LUAD tissue sequencing) and the large library of human peptides, the Pan Human Library (PHL)[50]. The LUAD assays were generated as previously described[95] from valid peptide spectrum matches (PSM) corresponding to tryptic peptides of 51 DDA-MS injections (i.e., 24 "full proteome" and 27 "on-bead enriched" LUAD digests). We excluded 12 DDA injections of negative control beads from the library creation process. We separately filtered all the SH assays available in the LUAD library and PHL, excluding any overlap between the assays, while the peptide query parameters were limited to 6 unique transitions per peptide. To estimate the error rate of our SH semi-specific library (that we named SHL) generated by the filtering of specific targets from the two spectral libraries, we created in silico two virtual library matrices of similar size to those of the PHL and LUAD, both calibrated to 1% of false positive targets. In the next step, a permutation test with 100 random data samplings was performed to estimate the error rate distribution. The FDR of our new SHL was estimated to be less than 2% with statistically significant probability ($p < 0.05$). We generated a.tsv file of SHL specific spectral library containing 3593 proteotypic peptides constructed from the top six most intense transitions with Q1 range from 350 to 2000 *m/z* with 64 window scheme excluding the precursor SWATH window (i.e. 12,653 peptide transition groups) and converging

to 215 SHs (Supplementary Data 1). We converted the library.tsv file to TraML format using the OpenMS tool ConvertTSVToTraML (version 1.10.0) and the decoy transition groups were generated based on shuffled sequences by the OpenMS tool OpenSwathDecoyGenerator (version 1.10.0). The generated SH library also involved peptide transition groups corresponding to internal standard peptides and experimentally detected peptides of known endogenous inhibitors and protein targets of SHs annotated in Merops (https://www.ebi.ac.uk/merops/). We used the SHL in the clinical cohort analysis, which increased the number of identified enzymes. For the analysis of total LUAD proteomes we used the large PHL.

**Analysis of SWATH/DIA-MS data from LUAD samples.** Raw SWATH sample files were converted into the mzXML format using ProteoWizard (version 3.0.331655)[96] and the data analysis performed using the OpenSWATH[29] integrated in the Euler portal workflow[97]. The input files consisted of: the mzXML files from the SWATH acquisitions, the TraML library file and the TraML file for iRT peptides. For the total LUAD proteome analysis and the initial analysis of intratumor tissue, we used the input TraML assay library file of previously published PHL containing 10,316 distinct proteins (139,449 proteotypic peptide sequences). For clinical cohort data analysis, we used the SHL containing peptide MS assays for 3593 proteotypic SH peptides in order to better fit the new analysis. After performing iRT data alignment, SWATH data records were extracted with 50 ppm extraction window and ±300 s around the expected retention time. The runs were aligned with a target FDR of 0.01 and a maximal FDR of 0.1[36]. In the absence of a confidently identified feature, the peptide and protein intensities were annotated as missing values. Next, the recorded feature intensities obtained from automatic OpenSWATH data processing were filtered with functions from Mayu FDR analysis to keep the identified features below 1% of FDR estimated on the protein level. The functions of R/Bioconductor package SWATH2stats (version 1.8.1) were used to reduce the size of the output data[98] and prepare the input files for R/Bioconductor package MSstats (version MSstats.daily 2.3.5)[99].

**Selection of "stable internal proteome" and internal standard peptides (ISPs).** We isolated a subset of stable endogenous transitions of housekeeping (HK) and cytoskeletal (CK) detected proteins that we used as internal standard peptides (ISPs). List of ISP was generated based on five-fold selection criteria: i) HK/CK proteins detected across the samples in the experiment i) peptides of HK/CK proteins <5% missing values for large clinical sample cohort; ii) low peptide variability iii) high Pearson correlation of peptides between paired digests and iv) minimal peptide sequences more-than >7AA. The standard HK proteins were selected manually based on information in the Human Protein Atlas (https://www.proteinatlas.org/). Correlations and variance tests were performed on standard quantile normalized data. We selected peptides of proteins that display a non-significant ratio of variability (F-test, $p > 0.2$) between paired depleted and total digests and with high Pearson correlation (Pearson cc> 0.6, $p < 0.05$). We selected −96 high-quality endogenous peptides (i.e. ISPs) that were used in intratumor or clinical study (Supplementary Data 2). The selected endogenous peptide transitions were used for two steps: i) tailored normalization strategy and 2) creation of reference distribution that defines the range of standard experimental noise.

**Quantitative data matrices of relative protein intensities.** Our analysis was based on the fragment ion MS2 intensities that are first filtered by R/Bioconductor package SWATH2stats and then summarized to the relative protein abundances based on a flexible family of linear mixed models in MSstats[99]. For tailored normalization, we introduced a selected list of ISPs in MSstats function "dataProcess" as Global standard peptides. MSstats-package was used to normalize data (i.e., Global standard peptides or quantile normalization) and convert

filtered fragment intensities to protein abundances. Missing values were imputed by an in-house script sampling from random distribution of values whose mean value corresponded to 90% of empirically detected minimal precursor peptide value ± 0.5 of standard deviation (SD) obtained from respective protein measurements.

**Evaluation of high confidence depletion ratio for SHs based on MS2 peak area.** We tested the hypothesis that the active enzyme displayed a significant distribution distance ($p < 0.05$) of its depletion ratio (log2Δint or -Δint for simplicity) with respect to the reference ISP distribution. Reference ISP distribution covers the range of the standard chemical noise in the MS experiment. To increase statistical power, we assessed the significance with a one-sided (i.e., alternative) hypothesis of the KS distance trend of two distributions. We expected enzyme depletion due to chemical probe affinity for the SHs' active site with serine nucleophile. The estimation of enzyme depletion range is performed by in-house written R script and based on the formula:

$$\overline{RADDi}(enzyme) = 1 - 2^{\wedge}(\log 2\Delta \text{ Int}), \quad (1)$$

$$\log 2\Delta \text{ Int} = \log 2\left(\sum_{i=1}^{i=k} dep.intensity[E]\right) - \log 2\left(\sum_{i=1}^{i=k} tot.intensity[E]\right) \quad (2)$$

$k$ = total number of confident peptides per enzyme[E] *
(* identical peptide transitions were used between depleted and total extracts to summarize enzyme levels in two respective extracts)

We used computed enzyme depletion ratios across samples (i.e., ΔInt) as input parameters to test the statistical significance of the enzyme activity per condition. We performed the goodness of fit test, i.e., two sample KS tests in R with alternative hypothesis (i.e., alternative = "greater"). ISP proteome is used as reference distribution against empirical distribution generated from observed depletion ratios of each respective enzyme per sample group. A similar goodness of fit test was applied previously in order to detect dispersion of data upon protein degradation[100]. In-house written R script was used to generate RADDi sample values for each enzyme detected in the clinical cohort. Only SH enzymes with confident RADDi value were selected for generation of activity matrices. Where the depletion signal of an enzyme fell within the defined noise range, its enzyme activity or RADDi value was set to 0. Analyses were performed in Rstudio Version 1.2.1335 and the R base package.

## Bioinformatics analyses of generated omics data

**Partial least squares discriminant analysis (PLS-DA).** We applied the Supervised Partial Least Squares Discriminant Analysis (PLS-DA) to reduce omics data dimensionality[52], and select the most descriptive features from large data matrices. The PLS-DA method was used for the selection of class-related features and is suitable when the number of features exceeds the number of samples. The number of selected latent variables that we imputed in the analysis is defined on the principle N-1, where N-corresponds to the number of classes in multiclass comparison. We thus used 3 components of the model and a limited number of variables for each PLS-DA component in order to select the most class-relevant markers. The subsequent variable importance or the VIP values (order of magnitude) are used to identify the X most important proteins per each PLS-DA variate. For activity data matrices wherein the number of imputed features was below <100, we limited analysis on 30 variables per component (in total: 64 unique SHs from combined PLS-DA variates 1-2-3). For total protein expression analysis with 3000 imputed features, we the limited analysis to the top 500 variables for each PLS-DA component (in total :1368 unique proteins from combined PLS-DA variates 1-2-3).

To combine different data levels (e.g., SHs and co-depleted proteins), we used the multi-block PLS-DA within the DIABLO[48]. The

algorithm searches explicit causal relationships between the latent variables of different data levels by Sparse Generalized Canonical Correlation Analysis (sGCCA), and whose values we extracted after performing block PLS-DA analysis[101]. In order to maximize the number of pair-wise associations between data levels, we did not limit the number of variables per component in multi-block PLS-DA. However, imputed data were often statistically pre-processed to alleviate computational time (e.g., KS-test $p < 0.001$, for imputation of depleted proteins). Consensus models were generated by tuning steps of integrated data (i.e., SH activities and co-depleted proteome) via MixOmics package function "tune.block.splsda". The input parameters included five-fold cross-validation and "centroids.dist" to estimate the classification error rate. The tuning function outputs is the optimal number of components achieving best performance based on overall classification error rate estimated by five cross-validation folds. The best consensus model was inspected for: i) the optimal number of components, ii) performance based on overall error rate estimated by five cross-validation folds and iii) visualization of sample clustering based on final classifiers within the consensus model (e.g., heat-maps). PLS-DA and multi-block PLS-DA were performed in R package 'mixOmics' (version 6.10.9)[48,49,101].

**GO analysis, network visualization and prediction of enzyme-protein interactions.** GO analysis was performed using R package Disease Ontology (DO) Semantic and Enrichment analysis[102]. To predict putative enzyme-protein interactions from dd-ABPP dataset we first used the Integrated Interactions Database (IIP)[56] as a reference and selected the list of the proteins that show recurring co-depletion with the active enzymes (i.e. depleted proteome). From the IIP database we extracted the list of 1st and 2nd degree interactors of SHs measured in study (SH-interactome, Supplementary Data 5). We first compared the overlap of reference SH-interactome and list of depleted proteome (Supplementary Data 6). In order to assess if this overlap was higher than expected, we generated 200 random interactomes of the same size as the reference SH-interactome. The interactors were selected for the same number of target proteins as the number of SHs in SH-interactome. Target proteins were randomly sampled from our total list of proteins identified in the study. This was repeated 200 times and each time we noted the total number of interaction partners that were also known as SH-protein pairs. Distribution of the values observed for the random interactors was compared to the corresponding value for the original number of SH interactors found in depleted proteome using the pnorm test in R.

Next, we used canonical correlation analysis (CCA) that tests for correlating structures between sets of discriminative variables (i.e., 64 SHs and 388 depleted proteins) within the multi-omics data integration tool DIABLO. We selected highly correlating enzyme and depleted-protein features (i.e., cor or sGCCA>0.4, $p < 0.05$) from PLS-DA latent components as inputs. P-value was estimated based on correlation significance test, wherein correlation between any two tested variables in the general population is estimated zero (cor=0), and if the sample size is N. We next compared overlap of these significantly "correlating features" with those enzyme-protein interactions with annotated strong experimental evidence. We assess if this overlap of discovered binary interactions based on sGCCA correlation was higher than expected by chance with pnorm test in R. Relevance networks were generated with Diabolo MixOmics package[101]. Then, networks were exported as "gml" files, using the igraph library and visualized with Cytoscape3.6.0 (https://www.cytoscape.org/)[103].

**Measurements of FA levels**
**Sample extraction.** A new set of cryostat sections from OCT-embedded tumors were collected for lipidomic analysis and cleaned as described above in the first-step clean-up process (section 4.2.). One sample in each group was available as fresh frozen tissue. We homogenized on average the 30 microsections per sample corresponding to approximately 8 mg of cleaned tissues in 450 μL PBS solutions (pH ~ 7.4). The protein concentrations of this initial tissue lysate were measured via the bicinchoninic acid assay (BCA assay, Thermo Fisher Scientific) in accordance with the manufacturer's instructions. Samples were prepared with the addition of 80 μL of isopropyl alcohol (IPA) to 20 μL of initial tissue lysate (in PBS) followed by vortex[104]. Mixed extracts were centrifuged for 15 min at 21,000 x g at 4 °C and the resulting supernatants were collected and transferred to LC-MS vials for subsequent lipidomic analyses (Supplementary Note 3-4-5).

**Relative comparison of FA levels across tumors.** Extracted samples were first analyzed by reversed phase liquid chromatography coupled to a high-resolution mass spectrometry (RPLC-HRMS) instrument (Agilent 6550 IonFunnel QTOF) in full scan mode in negative ESI. The chromatographic separation was carried out on a Zorbax Eclipse Plus C18 (1.8 μm, 100 mm × 2.1 mm I.D. column) (Agilent technologies, USA). Mobile phase was composed of A = 60:40 (v/v) ACN:water with 10 mM ammonium acetate and 0.1% acetic acid and B = 88:10:2 isopropanol:ACN:water with 10 mM ammonium acetate and 0.1% acetic acid. The linear gradient elution from 15% to 30% B was applied for 2 min, then from 30% to 48% B for 0.5 min, from 48% to 72% B for 6 min and from 72% to 99% B for 3 min, followed by 0.5 min isocratic conditions and 3 min re-equilibration to the initial chromatographic conditions. The flow rate was 600 μL/min, column temperature 60 °C and sample injection volume 2 μL. ESI source conditions were set as follows: dry gas temperature 200 °C, nebulizer 35 psi and flow 14 L/min, sheath gas temperature 300 °C and flow 11 L/min, nozzle voltage 1000 V, and capillary voltage − 3500 V. Full scan acquisition mode in the mas range of 100–1700 $m/z$ was applied for data acquisition in negative ESI.

Pooled QC samples (representative of the entire sample set) and blank solvents were analyzed periodically throughout the overall analytical run to assess the quality of the data, correct the signal intensity drift (if any, this drift is inherent to LC-MS technique and MS detector due to sample interaction with the instrument over time) and remove the peaks with poor reproducibility (CV > 30%).

Raw LC-HRMS data were verified for quality using MassHunter Agilent Technologies software (version10.0) and processed using Agilent Profinder 8.0 software. For FA screening and relative comparison across tumors, we performed targeted data mining by matching the accurate mass and retention time (AMRT) against an in-house database recorded on 25 pure AS analyzed in the same analytical conditions. The standards were obtained from Sigma-Aldrich (Darmstadt, Germany) (Supplementary Note 3).

We annotated peak areas corresponding to 14 FA that were measured consistently across samples. We normalized each sample run based on the average ion counts of the total ion current (TIC) of annotated FA peak areas in that sample run. To normalize for varying amounts of starting material, we reported results as normalized peak areas in respect to protein concentration measured in extracts prior to lipidomic analysis. Peak areas in ion count for detected FAs and total protein content for each analyzed sample are reported in (Supplementary Data 8).

**Absolute quantification of FA of interest by using LC-HRMS Orbitrap™ IQ-X™ Tribrid™ mass spectrometer.** The FA species that showed significant changes between the conditions of interest in the relative comparison (i.e. palmitoleic, oleic, and eicosapentaenoic acid) and palmitic acid, previously highlighted by proteomic data (Fig. 6d,e), were further quantified. For their absolute quantification, we used the calibration curves (prepared with pure AS non-labeled) and the isotopically labeled internal standards spiked with a known concentration. The pure AS standards were obtained from Sigma-Aldrich

(Darmstadt, Germany) and deuterated internal standards (palmitoleic acid-d13, Oleic acid-d9, eicosapentaenoic acid-d5 and hexadecenoic acid-d3) from Larodan AG (Solna, Sweden). LC-MS grade water, iso-propanol, ethanol, acetic acid and ammonium acetate were purchased from Biosolve Chimie (Dieuze, France), Sigma-Aldrich (Darmstadt, Germany) and Merck (Darmstadt, Germany).

The linearity of the calibration curves was evaluated for each fatty acid using a 9-point range (Cal0-Cal9). In addition, peak area integration was manually curated and corrected when necessary (Supplementary Note 4).

Preparation of Calibration Solutions (AS) and Internal Standard (IS) Mixtures. The initial AS mixture of FAs was prepared in ethanol. The concentrations of AS in the highest-level calibrator spanned from 50 μM to 250 μM. The subsequent 10 points of the calibration curve were prepared by serial dilutions of this highest calibrator using ethanol. The stock deuterated IS mixture was prepared in ethanol with concentrations ranging from 50 μM to 250 μM depending on the molecular species. Stock IS mixture is diluted 1/500 times with iso-propanol prior to the sample spike.

Sample and Calibration Curve Preparation. For absolute quantification of FAs, 20 μL of biological samples were prepared by adding 80 μL of iso-propanol spiked with internal standards (using stock IS mixture). Samples were then vortexed and centrifuged for 15 min at 15 °C and 21000 g. The resulting supernatant was transferred to LC-MS vials and injected into the ultra-high performance liquid chromatography (UHPLC)-HRMS system. Ten-point calibration curves were acquired following the same procedure as for the samples.

LC-MS sample acquisition. A Vanquish Horizon (Thermo Fisher Scientific) UHPLC system coupled to Orbitrap IQ-X Tribrid MS interfaced with the Heated Electrospray Ionisation (HESI) source was used for the quantification of palmitoleic, oleic, eicosapentaenoic and hexadecenoic acid. The chromatographic separation was carried out on a Zorbax Eclipse Plus C18 (1.8 μm, 100 mm × 2.1 mm I.D. column) (Agilent technologies, USA). Mobile phase was composed of A = 60:40 (v/v) ACN:water with 10 mM ammonium acetate and 0.1% acetic acid and B = 88:10:2 isopropanol:ACN:water with 10 mM ammonium acetate and 0.1% acetic acid. The linear gradient elution from 15% to 30% B was applied for 2 min, then from 30% to 48% B for 0.5 min, from 48% to 72% B for 6 minand from 72% to 99% B for 3 min, followed by 0.5 min isocratic conditions and 3 min re-equilibration to the initial chromatographic conditions. The flow rate was 600 μL/min, column temperature 60 °C and sample injection volume 2 μl. HESI source conditions operating in negative mode were set as follows: sheath gas flow at 60, aux gas flow rate at 20, sweep gas flow rate at 1, spray voltage at −3kV, capillary temperature at 300 °C, s-lens RF level at 35% and aux gas heater temperature at 300 °C. Full scan HRMS acquisition mode (m/z 180 − 500) was optimized to improve sensitivity with the following MS acquisition parameters: mass resolving power at 60,000 FWHM, 1 μscan, 1e$^5$ AGC and 118 ms as maximum inject time. Limit of detection (LOD) or minimum detected concentrations were reported in the Supplementary Data (Supplementary data 8), and undetected missing values for each FA accounted for half of the minimum detected value. Lipidomic raw data related to relative and absolute FA measurements are available via Zenodo under accession number zenodo.14841692.

## Metallomics of LUAD samples

Trace element concentrations were measured in 14 available LUAD tumor extracts adjusted to a concentration of 1 μg/μL of total protein by inductively coupled plasma system coupled to MS (ICP-MS; 7800 Series; Agilent, Palo Alto, Santa Clara, CA, USA)[72]. The selection of nine metal ions, and elementary quantification of Cadmium (Cd), Chromium (Cr), Lead (Pb), Nickel (Ni), Selenium (Se), Mercury (Hg), Arsenic (As), Cobalt (Co) and Aluminium (Al) was based on an established list of the chemicals and chemical compounds identified by the FDA

agency as Harmful and Potentially Harmful Constituents in Tobacco Products and Tobacco Smoke (https://www.fda.gov/tobacco-products/rules-regulations-and-guidance/harmful-and-potentially-harmful-constituents-tobacco-products-and-tobacco-smoke-established-list). 100 μL of tissue sample was diluted with 1.2 mL of Nitric acid 0.1% solution containing 10 ng/mL Rhodium and 10 ng/mL Indium as internal standards. The samples were processed with laboratory controls, including method blanks and standard reference materials to continuously monitor method performance[72]. Original values of respective metals in supplementary information's (Conc. [ng/ml], Conc. RSD and intensity (CPS); Supplementary Data 9).

## Cancer genomics analysis

Cancer genomics analysis is based on the data generated by the TCGA Research Network: http://cancergenome.nih.gov/. We obtained whole exome somatic mutation datasets deposited in the TCGA cancer genomics repository available from September 2015. We retrieved somatic mutations sequencing level 2 files from the TCGA web service and proceeded with the analyses of those files that passed the Broad Institute quality filters and were listed on their MAF Dashboard site. This included the following TCGA tumor types: acute myeloid leukemia (LAML), adrenocortical carcinoma (ACC), bladder urothelial carcinoma (BLCA), brain lower grade glioma (LGG), breast invasive carcinoma (BRCA), cervical squamous cell carcinoma and endocervical adenocarcinoma (CESC), cholangiocarcinoma (CHOL), colon adenocarcinoma (COAD), glioblastoma multiforme (GBM), head and neck squamous cell carcinoma (HNSC), kidney chromophobe (KICH), kidney renal clear cell carcinoma (KIRC), kidney renal papillary cell carcinoma (KIRP), liver hepatocellular carcinoma (LIHC), lung adenocarcinoma (LUAD), lung squamous cell carcinoma (LUSC), lymphoid neoplasm diffuse large B-cell lymphoma (DLBC), ovarian serous cystadenocarcinoma (OV), pancreatic adenocarcinoma (PAAD), pheochromocytoma and paraganglioma (PCPG), prostate adenocarcinoma (PRAD), rectum adenocarcinoma (READ), sarcoma (SARC), skin cutaneous melanoma (SKCM), stomach adenocarcinoma (STAD), testicular germ cell tumors (TGCT), thyroid carcinoma (THCA), uterine carcinosarcoma (UCS), uterine corpus endometrial carcinoma (UCEC), and uveal melanoma (UVM).

For each SH gene, we assessed the number of patients with at least one non-synonymous mutation within a gene. To account for the differences in gene lengths, the number of patients with a mutated gene was scaled by the total number of amino acids in the encoded protein. This provided an insight into differences of gene mutation rates across cancer patients. We performed the analysis separately for (i) LUAD samples, (ii) studied adenocarcinoma samples (COAD, LUAD, PAAD, PRAD, READ and STAD) and (iii) jointly for all analyzed cancer types. Next, we assessed if a mutation in any of the hydrolase genes significantly often co-occurred together with a mutation in the *KRAS* gene. For this, we used a Fisher's exact test and assessed the number of patients with mutations in both, a SH and *KRAS* gene, compared to patients with only a SH mutated, while considering the background information on the overall fraction of patients with a *KRAS* mutation. The obtained p-values were adjusted with Benjamini-Hochberg correction for multiple testing.

## Validation of SHs with selected activity assays

**ActivX® serine hydrolase probe labelling and detection of SHs.** For the detection of active SHs in the soluble proteome isolated from OCT samples, we used the ActivX® Serine Hydrolase Probe with TAMRA fluorescent reporter (Thermo Scientific, #88318). The probe enables selective labelling of active serine hydrolases and their sensitive detection using fluorescent gel imaging[105]. The labelling was performed immediately after extract preparation according to manufacturer's protocol. Briefly, the lysates were standardized before the labelling at a concentration of 1 mg/mL and a final volume of 100 μL.

For the labelling step, the probe was added to a final concentration of 2 µM and the sample was mixed on a shaker for 30 s at 20 x g. The labelling was allowed to proceed for 30 min at RT. Afterwards, 4xLaemmli reducing sample buffer was added to stop the reaction and the samples were incubated at 95 °C for 5 min.

For the NaDS-PAGE, 20 µg of each labelled sample was separated on a NuPAGE™ 4–12% Bis-Tris Protein Gel with 15-wells (Invitrogen). The gel was scanned with a Typhoon 9000 flat-bed fluorescent scanner (GE Healthcare) using the Green laser (Ex/Em:552/575 nm). The PMT value was set to 500 and pixel size was set to 50 µm to ensure a high-resolution image. The system was operated using Typhoon FLA 9500 control software version 1.1 (GE Healthcare). The fluorescent image analysis was performed in Image Quant TL 1D software package version 8.2.0 (GE healthcare). For the 1D gel analysis, the image was loaded in gel format and the sample lanes were annotated automatically with the build-in annotation function. The lanes were manually curated, and the background was subtracted using the image rectangle method. The detected fluorescence intensities were integrated using build-in functions for integration and summarized in a band-wise and sample-wise report table.

To confirm the loading of samples for fluorescence quantification analysis, we performed a loading control Western blot for GAPDH (MAB374, Sigma Aldrich) and mouse monoclonal anti β-Actin (AC-15, Sigma Aldrich, A5441) at 1/2000 dilution. The images were recorded on a FUSION FX Imager (Vilber).

**Detection of proteases in the gel-separated soluble proteome.** To detect where the SHs were located in the gel-separated proteome and to better annotate the fluorescent images, 30 µg of each tissue sample was loaded on a NuPAGE™ 4–12% Bis-Tris Protein Gel with 15-wells (Invitrogen). After Coomassie brilliant blue staining, 2 representative samples were selected and the respective protein lanes were cut into 10 slices following a standard in-gel protocol[106]. Briefly, the de-stained gel pieces were subjected to reduction and alkylation with subsequent in-gel trypsinization, and the peptides were extracted for analysis and desalted on a C18 spin column (The Nest group).

For the LC-MS/MS analysis an Orbitrap Elite (Thermo Fischer Scientific) coupled to Easy 1000 nano-LC unit (Thermo Fischer Scientific) was used and operated via Xcalibur software (Thermo Fischer Scientific). For each LC-MS/MS run, 1 µg of sample was loaded directly on the analytical column (Acclaim PepMapTM RSLC, 75 µm x 15 cm, nanoViper C18, 2 µm, 100 A, Thermo Fischer Scientific) and a 5–35% 60 min linear acetonitrile/water gradient was used at 300 nL/min flow rate. The MS spectra were acquired in the mass range of 350 to 1600 $m/z$ in the Orbitrap in the profile mode at a resolution of 120.000 at 400 $m/z$. For peptide fragmentation, CID method was used on 15 most intense precursor ions from full spectra and recorded in the ion trap analyzer. Precursors were dynamically excluded with 30 second and repeat count of 1 considering charges +2, +3 and +4.

All database searches were performed with MaxQuant (version 1.5.2.8) and human UniprotKB database (*Homo sapiens*, Uniprot release 2018_10, 20382 entries). For the searches, we used standard tryptic settings with 2 missed cleavages and considered cysteine carbamidomethylation (+57.0215 Da) as fixed modification and methionine oxidation (+15.9949 Da) and N-terminal acetylation (+42.0106 Da) as variable modifications. For the precursor mass tolerance and fragment ion mass tolerance the default values of 4.5 ppm and 0.5 Da were used, respectively. For the identification of hydrolases, a focused database containing 294 known hydrolase sequences was created and used for the searches. For the peptide identification, a general PEP score threshold of 0.05 was applied.

**Activity assays of selected proteases.** To evaluate the activity of prolyl endopeptidases we used activity assay from Sigma Aldrich (MAK088), For each measurement, 20 µg of sample was diluted in

the assay buffer to a final volume of 50 µL. Immediately before the assay, 50 µL of the substrate premix was added to a final concentration of 10 µM and the fluorescence was monitored with a Tecan infinite 2000 Pro (Tecan, Switzerland) at excitation wavelength of 370 nm and the emission wavelength of 460 nm continuously for 60 min. The relative activity of prolyl endopeptidases was compared from the linear parts of the plots. To confirm the specificity of the detected signals, in a repeat of the experiment the samples were pre-incubated for 30 min with PMSF at a 200 µM final concentration. To evaluate the activity of DPP4, we used sitagliptin inhibitor supplemented with the fluorescence assay kit. For the measurement, 20 µg of sample was diluted in assay buffer to a final volume of 50 µL and sitagliptin, a specific inhibitor of DPP4, was added to the assay at a 1/100 dilution. The fluorescence kinetics was measured on the plate reader and the relative activity of DPP4 was calculated as the difference between sitagliptin-inhibited and uninhibited samples. The residual signal after sitagliptin inhibition could be classified as belonging to other prolyl endopeptidases. All measurements were performed in triplicates.

To evaluate the activity of ELANE, we used the commercial fluorescence kit from Sigma Aldrich (MAK246). For the measurement, 20 µg of sample was diluted to a final volume of 50 µL with the assay buffer and. 50 µL of the substrate premix and fluorescence was monitored with the plate reader at an excitation wavelength of 380 nm and the emission wavelength of 500 nm continuously for 60 min. The relative activity of ELANE in the samples was compared from the linear parts of the plots as fluorescence increased over time. All measurements were performed in triplicates.

**Western blot.** 12 µg of proteins from patient lung extracts in PBS were separated by one-dimensional polyacrylamide gel electrophoresis (1D-PAGE) by using precast NuPAGE Novex 4–12% Bis–Tris gels (Invitrogen, Switzerland) and transferred onto nitrocellulose membranes. Then, the membranes were incubated with the respective primary antibodies Anti-SCD1 antibody [CD.E10] (ab19862) from Abcam at 1/1000 dilution at 4 °C overnight. We used enhanced chemiluminescence to reveal the signals by using ECL Select Western Blotting Detection Reagent (Cytiva, Amersham, United Kingdom). Mild stripping was used on each membrane for restaining with GAPDH antibody that was loading control. The stripping buffer was 15% glycine, 1% Sodium Dodecyl Sulfate (SDS) and 10% Tween 20 in distilled water. pH was adjusted to 2.2. Images were processed using ImageJ software. For quantitative comparison (Supplementary Note 1), gel images and uncropped scans are used.

**Statistical analysis**

For the selection of "stable" reference proteins, we used the F-test of equality of variances in order to compare the variances of proteins between depleted and total extract. Pairwise covariance analysis was performed between depleted– and total protein extracts via the cor.test function in R to calculate the Pearson's correlation coefficient and inspect for correlating features. For quality checking of the dd-ABPP method, we used one-sided paired $t$-test for matched depleted and total extract. To measure the dispersion of data for depletion ratios, we used the goodness of fit test, i.e., two sample KS-distance test. To increase statistical power for SHs, we assessed the significance of depletion with a one-sided hypothesis due to chemical-probe affinity for serine nucleophile in the SHs' active site. Non-parametric Spearman's rho correlation was calculated for the RDDAi data associations with total enzyme or protein abundance and metal ion concentration.

Differential analysis was performed depending on data type, data distribution and study design. For statistical analysis, the data distribution was evaluated using the Shapiro Wilk normality test and generating the QQ plots to observe the data residuals in R. Differential

analysis of total protein expression was computed by fitting samples into GLM for each subtype and tissue. The GLM analysis accounted for patient confounders such as age, sex, smoking status, or cell type. The gender and smoking status are treated as categorical variables that have two possible outcomes, 0 or 1 (e.g., male vs female). We defined the family function for a specification of the model to be fitted. Gaussian distribution was used for GLM of protein expression data. If the comparison exceeds two conditions, all pairwise comparisons are performed and post hoc analysis by "multcomp" package used to address a multiple testing correction. The percentage of enzyme activity (RADDi) datasets are inflated with zeros (i.e., inactive enzymes). For this datasets we performed GLM with quasibinomial distribution (link=probit) that is adapted when variable is a proportion or percentage. For enzyme activity comparisons, we also reported the original results with no adjustment for common confounders. Poisson distribution was used for on-bead quantified data as count data (i.e., spectral counts). P-values were corrected for multiple testing by the BH FDR method when possible. Original P-values or BH FDR $p$-adjusted values $\leq 0.05$ were considered as significant. Log-Rank test and Kaplan-Meier plot were used to visualize survival curves by using R packages: 1. "survival" for computing survival analyses and 2. "survminer" for summarizing and visualizing the results. Heat maps were made using R package pheatmap version 1.0.8. The quantitative data for RADDi were directly imputed without scaling. For protein log2abundance, data were imputed with or without scaling. The distance measures used for data clustering were either "Euclidean" or "Manhattan distance."

## Reporting summary

Further information on research design is available in the Nature Portfolio Reporting Summary linked to this article.

## Data availability

Proteomics Raw data of MS measurements are available in the PRIDE archive under the accession code PXD019357. Lipidomic raw data complementing the primary proteomic data and related to relative and absolute FA measurements are available via Zenodo [https://doi.org/10.5281/zenodo.14841692]. Lipidomics Minimal Reporting Checklist (https://lipidomicstandards.org/) for the two types of FA measurements are available in the Supplementary Material. Cancer genomics analysis is based on data generated by the TCGA Research Network [http://cancergenome.nih.gov/]. Metallomics and biochemistry data are available in the main text or in Supplementary Material. Raw gel images are available in the Supplementary Material. The individual morphological tissue for each patient is provided in the Source Data. TAMRA fluorescence output files are available in Figshare [https://doi.org/10.6084/m9.figshare.28755566]. Source data are provided with this paper.

## Code availability

All the above statistical analyses were performed using the straightforward application of statistical libraries in R from Bioconductor (https://bioconductor.org/). Custom R code script related to analysis of paired extracts available on https://doi.org/10.6084/m9.figshare.28755566.

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

## Acknowledgements

We thank all members of IMSB at ETH Zürich for help and critical discussions. In particular, we would like to thank Ludovic Gillet and George Rosenberger for fruitful discussion related to data normalization. We are also grateful to the technical team of the Metabolomics Platform at the University of Lausanne for their excellent support. We gratefully acknowledge contributions from the TCGA Research Network. The work carried out in this study was supported by the European Research Council (AdvG grant 670821) to R.A., by the Swiss National Science Foundation (grant # SNSF 31003A_166435), and by a grant from the Stiftung für angewandte Krebsforschung for the project "Biomarkers with enzymatic activities for improved risk stratification of lung cancer patients". The human body illustration in Fig. 2b and Supplementary Fig. 3 is designed by Freepik.

## Author contributions

Author contributions: Conceptualization: T.S., R.A., S.H., S.A., W.W.; Methodology: T.S., M.V., S.A., S.L., M.M., J.I., H.G., A.T.; Investigation: T.S., M.V., R.C., M.B., S.A.; Visualization: T.S., M.V., M.B., R.C.; Funding acquisition: R.A., S.H., A.T.; Project administration: R.A., T.S., S.H.; Supervision: R.A., S.H.; Writing – original draft: T.S., R.A., R.C., M.V., M.B.; Writing – review & editing: all authors.

## Competing interests

The authors declare no competing interests.

## Additional information

**Supplementary information** The online version contains
supplementary material available at

Tatjana Sajic, Sven Hillinger or Ruedi Aebersold.

**Peer review information** *Nature Communications* thanks Paula Diez,
Simion Kreimer, Jun-Seok Lee, Hiroshi Tsugawa and the other,
anonymous, reviewer(s) for their contribution to the peer review of this
work. A peer review file is available.

**Publisher's note** Springer Nature remains neutral with regard to
jurisdictional claims in published maps and institutional affiliations.

Tatjana Sajic [1,2,9] ✉, Matej Vizovišek[1,9], Stephan Arni [3,9], Rodolfo Ciuffa[1], Martin Mehnert[1], Sébastien Lenglet[4],
Walter Weder[3], Hector Gallart-Ayala [5], Julijana Ivanisevic [5], Marija Buljan[6,7], Aurelien Thomas [2,4], Sven Hillinger[3,10] ✉
& Ruedi Aebersold [1,8,10] ✉

[1]Department of Biology, Institute of Molecular Systems Biology, ETH, Zurich, Switzerland. [2]Faculty Unit of Toxicology, CURML, Faculty of Biology and
Medicine, University of Lausanne, Lausanne, Switzerland. [3]Division of Thoracic Surgery, University Hospital Zurich (UHZ), Zürich, Switzerland. [4]Unit of
Forensic Toxicology and Chemistry, CURML, Lausanne and Geneva University Hospitals, Lausanne, Geneva, Switzerland. [5]Metabolomics and Lipidomics
Platform, Faculty of Biology and Medicine, University of Lausanne, Quartier UNIL-CHUV, Rue du Bugnon 19, CH-1005 Lausanne, Switzerland. [6]Empa, Swiss
Federal Laboratories for Materials Science and Technology, 9014 St Gallen Dübendorf, Switzerland. [7]Swiss Institute of Bioinformatics (SIB),
Lausanne, Switzerland. [8]Faculty of Science, University of Zurich, Zurich, Switzerland. [9]These authors contributed equally: Tatjana Sajic, Matej Vizovišek,
Stephan Arni. [10]These authors jointly supervised this work: Sven Hillinger, Ruedi Aebersold. ✉e-mail: tatjana.sajic@chuv.ch; sven.hillinger@usz.ch;
aebersold@imsb.biol.ethz.ch

