## [Transparent Peer Review file · Nature Communications]

Depletion-dependent activity-based protein profiling using SWATH/DIA-MS detects serine hydrolase lipid remodeling in lung adenocarcinoma progression

Corresponding Author: Dr Tatjana Sajic

Version 0:

Reviewer comments:

Reviewer #1

(Remarks to the Author)

This manuscript describes the development of "depletion-dependent activity-based protein profiling (dd-ABPP)" methods and their applications in investigating lipid remodeling during lung adenocarcinoma progression. Traditional ABPP depends on chemically modifying a specific set of enzymes using a synthetic warhead, and these modified enzymes are enriched through streptavidin-biotin interactions. This method offers several apparent advantages. For example, ABPP assesses the activity status of enzymes by producing covalent bonds through catalytic activity. Unlike many conventional proteomic methods, which measure abundance, ABPP can reveal distinct dimensional information. In this study, the authors propose a slightly altered method, using parallel datasets of normal (or whole proteome) versus ABPP. Interestingly, they remove proteins conjugated with the warhead using streptavidin beads, instead of enriching them, an approach that is novel compared to other ABPP studies. This significantly improves MS analysis by reducing contaminant spectra from streptavidin peptides generated during the tryptic digestion process. It should also be noted that the foundation of dd-ABPP relies on high-coverage SWATH/DIA proteomic methods. Since the target enzymes are not effectively detected without the enrichment step, it is impossible to conduct this type of negative detection method. Throughout the manuscript, the authors thoroughly investigate controls and present convincing results. Therefore, this reviewer agrees with the acceptance of this manuscript for publication in Nature communications.

Here are some minor comments:

- In ABPP methods, enzyme activity is crucial. It would be beneficial to add discussions about the nature of patient samples and the enzyme activity status. For frozen or OCT-embedded tissues, how are enzymes kept in an active state?
- Although comprehensively described in the main text, Figure 1 is not reader-friendly. In Figure 1-b, the purple color representing "bead" and "beads digest" needs to be more distinctly differentiated. In Figure 1-c, the retention time of peptide1 compared to the "total extract" and "depleted extract" should be clarified. The activity should represent the difference between the total and depleted extracts, but the dark blue color area does not consistently reflect this.
- Regarding Figure 5d, the authors highlight the fourth most significant element in their Gene Ontology (GO) analysis. It would be beneficial to add the rationale for not focusing primarily on the top three aspects. Additionally, please clarify the selection criteria used, particularly concerning protein depalmitoylation.

Reviewer #2

(Remarks to the Author)

Sajic et al. present an interesting clinical study on lung adenocarcinoma progression. The presented workflow is based on an automated DIA-MS workflow combining depletion-dependent activity-based proteome profiling. The protein profiles associated with active enzymes could classify an aggressive advanced-stage lung adenocarcinoma using tissue microsections. The manuscript is well written and a multiomics approach was shown, including information layers collected from proteomics, targeted lipidomics, metallomics, and genomics analyses. One significant outcome presented was the

increase in monounsaturated lipids in aggressive tumors. Although the value of the clinical cohort on LUAG and the analytical novelty on the proteomics side are apparent, and the metallomics data on smoking patients are interesting, several pieces of information need to be added on the lipidomics side. Unfortunately, the presented lipidomics data does not follow the minimal requirements to confirm the hypothesis of an increase in monounsaturated lipids in aggressive tumors. Therefore, the quality of the presented work is only suitable for Nature communication with additional experiments and major manuscript changes.

Major comments:

Lipid sample preparation:

- 1. Extraction: What was the sample preparation for the lipidomics analysis? It seems like the starting point was 20 µL.. but of which samples?
- 2. Standards: Were any internal standards added for fatty acid analysis (e.g. commercially available standards from Lipotype or Avanti) at least at the point of sample extraction?
- 3. Sample storage: It is not clear how long the samples were stored at -80°C and if there any oxidation-preventing reagents (e.g. BHT) were added for lipid analysis.

As fatty acids/eicosanoids were analyzed, suitable sample preparation, including the addition of internal standards, is crucial to ensure high-quality lipid results.

Lipid analysis: The presented workflow is based on a bulk lipidomics profiling approach by RP-HRMS using an isopropanol gradient to elute lipids followed by recording a mass range of 100-1700 m/z with the QTOF. If the authors really intended only to acquire fatty acid profiles via targeted lipidomics as presented, the analysis method, e.g., the mass range, is suboptimal (all C8-C22 fatty acids table presented in Fig6 were within the mass range from 140-350 m/z => this would be ideal for a GC-MS FAMES analysis m/z < 500). It rather seems like a standard (non-targeted) lipidomics analysis was performed in the beginning and only the fatty acids profiles were included for data analysis. More information on the rationale of the method has to be provided:

- The authors presented a targeted lipidomics approach based on MS1 high-resolution analysis using an Agilent 6550 QTOF. Targeted lipid analysis is usually done with an SRM/MRM approach, including known retention time windows, specific lipid fragments (to avoid false annotations), and standards (to confirm the annotation). At least from the methods parts, it seems no MS2 information, standards, blanks (for blank subtraction), or known retention time information was used. If this is the case, the workflow is insufficient to prove the identity of the fatty acids analysed, as in-source fragments or lipid contamination (fatty acids are background contaminants) could be the reason for the recorded signals. If known retention time windows were used (e.g. from a previously acquired database) at least the equivalent carbon number model (ECN) can be used to confirm the experimentally determined lipids. The authors must provide more information on the targeted method part and prove the reliability of the presented lipid annotations!

- What about the other bulk lipids such as PC, TG, etc?

- How did the authors ensure that the analyzed fatty acids were not only in-source fragments from bigger lipids, e.g., PC?

- Additionally, the correct normalization for the targeted lipid analysis is important: How was the data normalized? TIC and protein normalization are mentioned, but it is not clear at which point of the analysis the protein determination was performed

Lipid data analysis: From the presented combination of C8-C22, including up to 6 DB positions, around 60 lipids can be produced. One significant outcome presented in the whole manuscript was the increase in monounsaturated lipids in aggressive tumors (Fig 6a+b).

- Although palmitoleic acid 16:1 and oleic acid 18:1 are usually the most prominent fatty acids, the authors cannot conclude that monounsaturated lipids are upregulated in aggressive tumors in general. The presented sum parameter in Fig 6b is misleading as it shows unsaturated fatty acids, which include both mono- and polyunsaturated fatty acids. We agree that an upregulation trend of unsaturated fatty acids is observed using the presented data. However, a selection of such a small subset of fatty acids is not sufficient to draw a general conclusion in aggressive tumors.

- Fig 6a: Which were the bases for selecting and ranking the 14 presented fatty acids/eicosanoids from source data Fig 6? Why is the plot separated in two? Is the separation of the plots based on the normalized areas? Why, after C8:0, C10:0, C12:0, a C20:5 is listed, followed by C18:3? An alphabetic order or order by fatty acids lengths and number of DB would make more sense. In the methods part, CV < 30% was mentioned; was this the selection parameter?

- Fig 6b: The authors conclude that the levels of saturated versus monosaturated and polyunsaturated were lower in aggressive tumors. Was this information derived on the limited sample set of 14 fatty acids (CVs less than 30%?)? For (relative) quantification comparisons of lipids in different groups internal standards are essential.

- What does the text description "Arbitrary Units normalized to the mean of the observations in the long-term survival subtype" in the Figure (6) caption mean?

- Please follow the FAIR principles: (1) No lipidomics raw data is available: The raw data and annotated metabolite and lipid lists must be made accessible to the community (e.g., GNPS, MetaboWorkbench, MetaboLights or in your case, you could also add it to PRIDE with more information on the lipid sample preparation) as it was done for the proteomics data; (2) the minimal reporting checklist for the lipid part should be included and also presents a way of checking the quality of the presented lipid annotation for the authors, the referees and the community: https://lipidomicstandards.org/reporting_checklist/

Other comments

- Page 3: The argumentation line on why DIA was used instead of DDA does not make sense from an analytical point of view. Please clarify and use more recent references

- For a non-proteomics expert, the novel generation of dd-ABPP is not entirely clear. Please introduce your workflow and its novelty in a more comprehensive way, maybe by using an additional (Supp.) Figure? Figure 7 is a bit confusing in its current form. As you also present multi-omic layers, a workflow figure for all performed analyses would enhance the reader's understanding of your workflow and its novelty

- The study design and the applied number of used tissue microsections are very confusing in the method sections. Why did you start with 24 samples and only show 16 in source data Table 1? The tumor and non-tumor samples seemed derived from the same 16 persons. Please provide a more comprehensive list or visualization in the supplementary to explain these numbers and clinical cohort and cohort 2.
- ABP and ABPP: It is a bit misleading to have such close abbreviations used throughout the manuscript, if possible please clarify

Reviewer #3

(Remarks to the Author)

I co-reviewed this manuscript with Evelyn Rampler, who already provided our report. This is part of the Nature Communications initiative to facilitate training in peer review and to provide appropriate recognition for Early Career Researchers who co-review manuscripts.

Reviewer #4

(Remarks to the Author)

Sajic and collaborators describe an approach that combines already existing technologies for the comprehensive characterization of enzyme activities. Despite they do not introduce any new techniques, the description of their SWATH/DIA-MS workflow combined with an ABPP method is interesting and has been well conducted. The authors have focused their investigations on the involvement of serine hydrolases in advanced-stage lung adenocarcinoma, as a proof-of-concept. It is also worth mentioning the extensive data analysis performed throughout the study, including ICP-based metallomics screening, enzyme-activity tests and orthogonal validations. Overall, the topics addressed in this manuscript are timely and of relevance to be published in Nature Communications. Nevertheless, the paper needs some modifications.

1. Fig.4b: I would like to see the quantification of the gel area of the control proteins (actin and GADPH).
2. Section 2.3. (page 8). The authors state that "protein patterns extracted from tumor and surrounding tissue displayed approx two-fold higher fluorescence intensities for tumor samples", but this is not the case for all samples analysed. The authors should further comment on this.
3. Section 2.3. (page 8). The 6 out of 13 SH that classified LUAD tumors should be mentioned. Also, when referring to the corresponding Fig.3g, the authors should explain the criteria for selecting those 6 SH.
4. I was interested in looking at the stable internal proteome list, but I could not find any table in the additional files. Could the authors be so kind as to provide such list?
5. Did the authors note any correlations/associations between gender and SH activities?
6. Section 4.3., subsection "Generation of phylogenetic tree of SH family members". The authors state: "We catalogued a total of 335 proteins", but the indicated proteins sum up 350. I assume the authors have not included in this calculation the 15 inactive enzymes. However, in Extended Data Fig.1C, these 15 proteins are pointed out. Please, double-check all these numbers.

Minor comments:

1. Section 2.2., subsection "Classifying LUAD conditions...". The authors report "...smaller differences (i.e. 130/471, Fig.2d)", should not be 130/438?
2. Section 2.2. (page 7), for data regarding LYPLA1 and FAAH2, the correct figure is Extended Data Fig.5c.
3. In Section 4.2., the word "urea" is in capital letters.
4. Section 4.3. when referring to the phylogenetic tree, the authors cite Extended Data Fig.2. However, such a tree is shown in Extended Data Fig.1C.
5. Section 4.3., "Development of SH semi-specific assay library", the referenced Extended Data Fig.2 should be Extended Data Fig.1C.
6. Section 4.3., subsection "Selection of stable internal proteome...". The authors mention they followed a five-fold selection criteria, however, only 4 criteria are detailed.
7. In Section 4.8., change room temperature by RT
8. Section "4.8. Statistical analysis" should be numbered 4.9.
9. Indicate all centrifugation speeds in g, not in rpm.
10. Some acronyms are listed more than once throughout the text (e.g., FA, ACN).
11. Some units are differentially mentioned in the text, e.g., minutes and min, ml and mL. Please, correct them accordingly.

Reviewer #5

(Remarks to the Author)

The manuscript describes, dd-ABPP, a methodology that measures the activity of a whole class of enzymes by data independent acquisition (DIA) mass spectrometry-based proteomics. Essentially a biotin labelled substrate homologue and streptavidin coated beads fish out the active enzymes that convert the bait substrate. Quantitative LC-MS analysis of the captured fraction is then combined with quantitative LC-MS analysis of the total protein content and the ratio of the captured enzyme to total enzyme is a measure of activity in the sample. This ratio is then compared between cohorts, and additional biological insight can be gleaned about the targeted class of enzymes and their role in the investigated condition. Overall, the manuscript presents compelling evidence from multiple angles that this methodology is sound and presents novel findings that would be missed by typical LC-MS proteomics that quantified protein quantity regardless of activity.

Although the title of the manuscript narrows the scope of the investigation to lung cancer, major question about the universality of dd-ABPP arise: Does dd-ABPP work with other classes of enzymes? Is this discrepancy between active and total enzyme a symptom of cancer, or does this happen in other diseases? Is proteomics missing most of the picture by focusing on only total protein quantity? Given the obvious labor that went into this manuscript it would not be reasonable to ask for further investigation, but inclusion of data from other applications and enzyme classes, if available, in the discussion would help with answering these questions and would enrich the manuscript.

Major issues

It is not very clear how the tumor sections and patients were associated. Were the tumor sections combined for each patient? Were the sections from different tumors treated as different data points? In section 2.2 page 5 it says that 6 tumors and 6 non tumors were analyzed from long term and short term survivor patients (so 12 tumor and 12 nontumor), in table 1 there are 10 male and 6 female patients listed, then Fig 2 b is back to 24 biosamples, but there are 3 intra-tumor replicates and >50 paired tissue extracts. Clarification is needed.

Minor issues

Fig.1a middle panel and Fig1c second panel seem redundant.

Page 8 "five kay proteins" should be "five key proteins."

Version 1:

Reviewer comments:

Reviewer #1

(Remarks to the Author)

Though I didn't fully understand how the frozen or OCT embedded tissue still retain the enzyme activity status, if those findings are consistent, there must be something going on during the reconstitution or removal of OCT. What is happening during this process might be worth to investigate more detail in the separate study.

Thus, this revised version addressed all previous my concerns properly. Anyway, this reviewer support for the publication.

Reviewer #4

(Remarks to the Author)

After reviewing the revised version of the manuscript, I confirm that the authors have adequately addressed the concerns raised during my previous review. The manuscript now meets the high standards and quality expected for publication in Nature Communications.

Reviewer #5

(Remarks to the Author)

The authors have adequately addressed my concerns, and I believe the comments of the other reviewers in this revision of the manuscript. The new Figure 1 is much easier to read. In my opinion the manuscript is acceptable for publication.

Reviewer #6

(Remarks to the Author)

I have reviewed the information in the revised manuscript, focusing on the lipidomics methods. The authors have addressed several comments, although I have some comments relating to the FFA identification and quantification. Some of these also were asked by Reviewer2. Overall, it's not possible to fully verify if the lipids were properly identified without further information, and the raw data doesn't seem to be available at the zenodo link.

The newly included text states that OCT embedded tissues were used for lipidomics analysis. This is very unusual, and introduces a significant issue since OCT itself is a complex medium containing a mixture of glycols and synthetic resins (Qiagen factsheet), and it can't be excluded that it contains also significant amounts of FFA. Normally for lipidomics, unprocessed, snap frozen, fresh tissue is used. There is no mention of any OCT blanks being also extracted for comparison with tissues, and even if there were, how could the amount of OCT in these blanks be matched to tissues, so as to calculate an appropriate background level to subtract? In my view, contamination of samples with OCT could render the FFA analysis completely unreliable unless this was properly addressed experimentally. Even if OCT was not used, FFA contamination is a significant problem in lipidomics analysis of samples, and blanks are always needed to ensure that contamination is accounted for. Reviewer2 asked about background contamination, but this has not been clearly answered.

For the untargeted analysis, more information needs to be provided to evidence the proper identification of the FFA using this method. Can the authors provide sample chromatograms showing each FFA, and demonstrate that they match the RT of the standards used. A RT list for lipids measured using this assay should be provided. They say 25 isotopically labelled standards were used for this method, but there is no list of these standards or information about their m/z values. This contrasts with the use of only 4 labelled standards for the targeted assay.

For quantitation in a target assay, it is usual to have a primary authentic standard for each compound being quantified, and an internal standard (for which they used 4), to control for extraction efficiency and relative ionization differences between the IS and the analyte. There isn't any mention in the methods about use of primary standards, so can more information be provided relating to how exactly the quantification was performed using the internal standards. A representative chromatogram for each quantified FFA would also be helpful to evidence the quality of the data.

Reviewer 2 asked how long the samples were stored at -80. This question wasn't clearly answered in the rebuttal. Information based on when samples were collected, then stored should be provided to answer this question. Oxidation is a major problem in cohort studies that go back several years. Although eicosanoids were not measured, several oxidizable FFA were measured and oxidation will result in their loss. So, it would be important to know that samples being compared had been stored for similar lengths of time at the very least, so that oxidation issues are not impacted by different groups being stored for different times.

The lipidomics data doesn't seem to be accessible at the Zenodo link provided. This means the quality of the data can't be checked using the raw data. Provision of a full set of representative chromatograms for both methods would be important at the very least.

Version 2:

Reviewer comments:

Reviewer #7

(Remarks to the Author)

I have limited my comments specifically to the authors' responses to Reviewer #6. Overall, the authors have provided logical and satisfactory answers to Reviewer #6's concerns. However, I have several questions regarding the figures that were submitted, and it would be beneficial for the authors to address these clearly.

Regarding For-review Figure 3:

The EIC for the oleic acid standard in Figure 3A appears to show two peaks. The authors should discuss this clearly. Additionally, there seems to be significant contaminant peaks present for oleic acid and palmitic acid. I strongly recommend the authors provide three separate chromatograms: (1) solvent-only analysis, (2) procedure blank (samples processed identically but containing only OCT without biological samples), and (3) biological samples. Demonstrating the absence (or presence) of these peaks, especially in the procedure blank, is critical. Even if contaminants are observed in procedure blank, submitting these chromatograms will clarify how background subtraction was performed.

Regarding Figure 4, the very large background observed for palmitic acid again raises doubts about the reliability of the data. From an analytical chemistry perspective, the severe peak tailing observed compromises quantitative accuracy—even if standard curves are employed. Peak tailing should ideally be avoided to ensure quantitative accuracy. The authors should explain clearly how they validated the peak area ranges of palmitic acid. Similar to the previous point, providing extracted ion chromatograms (EIC) from solvent-only and procedure blank analyses would clarify how background peaks were accounted for.

Point-by-point response to the reviewers' comments

REVIEWER COMMENTS

Reviewer #1

This manuscript describes the development of "depletion-dependent activity-based protein profiling (dd-ABPP)" methods and their applications in investigating lipid remodeling during lung adenocarcinoma progression. Traditional ABPP depends on chemically modifying a specific set of enzymes using a synthetic warhead, and these modified enzymes are enriched through streptavidin-biotin interactions. This method offers several apparent advantages. For example, ABPP assesses the activity status of enzymes by producing covalent bonds through catalytic activity. Unlike many conventional proteomic methods, which measure abundance, ABPP can reveal distinct dimensional information. In this study, the authors propose a slightly altered method, using parallel datasets of normal (or whole proteome) versus ABPP. Interestingly, they remove proteins conjugated with the warhead using streptavidin beads, instead of enriching them, an approach that is novel compared to other ABPP studies. This significantly improves MS analysis by reducing contaminant spectra from streptavidin peptides generated during the tryptic digestion process. It should also be noted that the foundation of dd-ABPP relies on high-coverage SWATH/DIA proteomic methods. Since the target enzymes are not effectively detected without the enrichment step, it is impossible to conduct this type of negative detection method. Throughout the manuscript, the authors thoroughly investigate controls and present convincing results. Therefore, this reviewer agrees with the acceptance of this manuscript for publication in Nature communications.

Here are some minor comments:

- In ABPP methods, enzyme activity is crucial. It would be beneficial to add discussions about the nature of patient samples and the enzyme activity status. For frozen or OCT-embedded tissues, how are enzymes kept in an active state?
- Although comprehensively described in the main text, Figure 1 is not reader-friendly. In Figure 1-b, the purple color representing "bead" and "beads digest" needs to be more distinctly differentiated. In Figure 1-c, the retention time of peptide1 compared to the "total extract" and "depleted extract" should be clarified. The activity should represent the difference between the total and depleted extracts, but the dark blue color area does not consistently reflect this.
- Regarding Figure 5d, the authors highlight the fourth most significant element in their Gene Ontology (GO) analysis. It would be beneficial to add the rationale for not focusing primarily on the top three aspects. Additionally, please clarify the selection criteria used, particularly concerning protein depalmitoylation.

Reviewer #2

Sajic et al. present an interesting clinical study on lung adenocarcinoma progression. The presented workflow is based on an automated DIA-MS workflow combining depletion-dependent activity-based proteome profiling. The protein profiles associated with active enzymes could classify an aggressive advanced-stage lung adenocarcinoma using tissue microsections. The manuscript is well written and a multiomics approach was shown, including information layers collected from proteomics, targeted lipidomics, metallomics, and genomics analyses. One significant outcome presented was the increase in monounsaturated lipids in aggressive tumors. Although the value of the clinical cohort on LUAG and the analytical novelty on the proteomics side are apparent, and the metallomics data on smoking patients are interesting, several pieces of information need to be added on the lipidomics side. Unfortunately, the presented lipidomics data does not follow the minimal requirements to confirm the hypothesis of an increase in monounsaturated lipids in aggressive tumors. Therefore, the quality of the presented work is only suitable for Nature communication with additional experiments and major manuscript changes.

Major comments:

Lipid sample preparation:

- 1. Extraction: What was the sample preparation for the lipidomics analysis? It seems like the starting point was 20 μ L. but of which samples?
- 2. Standards: Were any internal standards added for fatty acid analysis (e.g. commercially available standards from Lipotype or Avanti) at least at the point of sample extraction?
- 3. Sample storage: It is not clear how long the samples were stored at -80°C and if there any oxidation-preventing reagents (e.g. BHT) were added for lipid analysis.

As fatty acids/eicosanoids were analyzed, suitable sample preparation, including the addition of internal standards, is crucial to ensure high-quality lipid results.

Lipid analysis: The presented workflow is based on a bulk lipidomics profiling approach by RP-HRMS using an isoproponal gradient to elute lipids followed by recording a mass range of 100-1700 m/z with the QTOF. If the authors really intended only to acquire fatty acid profiles via targeted lipidomics as presented, the analysis method, e.g., the mass range, is suboptimal (all C8-C22 fatty acids table presented in Fig6 were within the mass range from 140-350 m/z => this would be ideal for a GC-MS FAMES analysis $m/z < 500$). It rather seems like a standard (non-targeted) lipidomics analysis was performed in the beginning and only the fatty acids profiles were included for data analysis. More information on the rationale of the method has to be provided:

- The authors presented a targeted lipidomics approach based on MS1 high-resolution analysis using an Agilent 6550 QTOF. Targeted lipid analysis is usually done with an SRM/MRM approach, including known retention time windows, specific lipid fragments (to avoid false annotations), and standards (to confirm the annotation). At least from the methods parts, it seems no MS2 information, standards, blanks (for blank subtraction), or known retention time information was used. If this is the case, the workflow is insufficient to prove the identity of the fatty acids analysed, as in-source fragments or lipid contamination (fatty acids are background contaminants) could be the reason for the recorded signals. If known retention time windows were used (e.g. from a previously acquired database) at least the equivalent carbon number model (ECN) can be used to confirm the experimentally determined lipids. The authors must provide more information on the targeted method part and prove the reliability of the presented lipid annotations!
- What about the other bulk lipids such as PC, TG, etc?
- How did the authors ensure that the analyzed fatty acids were not only in-source fragments from bigger lipids, e.g., PC?
- Additionally, the correct normalization for the targeted lipid analysis is important: How was the data normalized? TIC and protein normalization are mentioned, but it is not clear at which point of the analysis the protein determination was performed

Lipid data analysis: From the presented combination of C8-C22, including up to 6 DB positions, around 60 lipids can be produced. One significant outcome presented in the whole manuscript was the increase in monounsaturated lipids in aggressive tumors (Fig 6a+b).

- Although palmitoleic acid 16:1 and oleic acid 18:1 are usually the most prominent fatty acids, the authors cannot conclude that monounsaturated lipids are upregulated in aggressive tumors in general. The presented sum parameter in Fig 6b is misleading as it shows unsaturated fatty acids, which include both mono- and polyunsaturated fatty acids. We agree that an upregulation trend of unsaturated fatty acids is observed using the presented data. However, a selection of such a small subset of fatty acids is not sufficient to draw a general conclusion in aggressive tumors.

- Fig 6a: Which were the bases for selecting and ranking the 14 presented fatty acids/eicosanoids from source data Fig 6? Why is the plot separated in two? Is the separation of the plots based on the normalized areas? Why, after C8:0, C10:0, C12:0, a C20:5 is listed, followed by C18:3? An alphabetic order or order by fatty acids lengths and number of DB would make more sense. In the methods part, CV < 30% was mentioned; was this the selection parameter?

- Fig 6b: The authors conclude that the levels of saturated versus monosaturated and polyunsaturated were lower in aggressive tumors. Was this information derived on the limited sample set of 14 fatty acids (CVs less than 30%)? For (relative) quantification comparisons of lipids in different groups internal standards are essential.

- What does the text description "Arbitrary Units normalized to the mean of the observations in the long-term survival subtype" in the Figure (6) caption mean?

- Please follow the FAIR principles: (1) No lipidomics raw data is available: The raw data and annotated metabolite and lipid lists must be made accessible to the community (e.g., GNPS, MetaboWorkbench, MetaboLights or in your case, you could also add it to PRIDE with more information on the lipid sample preparation) as it was done for the proteomics data; (2) the minimal reporting checklist for the lipid part should be included and also presents a way of checking the quality of the presented lipid annotation for the authors, the referees and the community: https://lipidomicstandards.org/reporting_checklist/

Other comments

- Page 3: The argumentation line on why DIA was used instead of DDA does not make sense from an analytical point of view. Please clarify and use more recent references

- For a non-proteomics expert, the novel generation of dd-ABPP is not entirely clear. Please introduce your workflow and its novelty in a more comprehensive way, maybe by using an additional (Supp.) Figure? Figure 7 is a bit confusing in its current form. As you also present multi-omic layers, a workflow figure for all performed analyses would enhance the reader's understanding of your workflow and its novelty

- The study design and the applied number of used tissue microsections are very confusing in the method sections. Why did you start with 24 samples and only show 16 in source data Table 1? The tumor and non-tumor samples seemed derived from the same 16 persons. Please provide a more comprehensive list or visualization in the supplementary to explain these numbers and clinical cohort and cohort 2.

- ABP and ABPP: It is a bit misleading to have such close abbreviations used throughout the manuscript, if possible please clarify

Reviewer #3

Reviewer #4

Sajic and collaborators describe an approach that combines already existing technologies for the comprehensive characterization of enzyme activities. Despite they do not introduce any new techniques, the description of their SWATH/DIA-MS workflow combined with an ABPP method is interesting and has been well conducted. The authors have focused their investigations on the involvement of serine hydrolases in advanced-stage lung adenocarcinoma, as a proof-of-concept. It is also worth mentioning the extensive data analysis performed throughout the study, including ICP-based metallomics screening, enzyme-activity tests

and orthogonal validations. Overall, the topics addressed in this manuscript are timely and of relevance to be published in Nature Communications. Nevertheless, the paper needs some modifications.

1. Fig.4b: I would like to see the quantification of the gel area of the control proteins (actin and GADPH).
2. Section 2.3. (page 8). The authors state that “protein patterns extracted from tumor and surrounding tissue displayed approx two-fold higher fluorescence intensities for tumor samples”, but this is not the case for all samples analysed. The authors should further comment on this.
3. Section 2.3. (page 8). The 6 out of 13 SH that classified LUAD tumors should be mentioned. Also, when referring to the corresponding Fig.3g, the authors should explain the criteria for selecting those 6 SH.
4. I was interested in looking at the stable internal proteome list, but I could not find any table in the additional files. Could the authors be so kind as to provide such list?
5. Did the authors note any correlations/associations between gender and SH activities?
6. Section 4.3., subsection “Generation of phylogenetic tree of SH family members”. The authors state: “We catalogued a total of 335 proteins”, but the indicated proteins sum up 350. I assume the authors have not included in this calculation the 15 inactive enzymes. However, in Extended Data Fig.1C, these 15 proteins are pointed out. Please, double-check all these numbers.

Minor comments:

1. Section 2.2., subsection “Classifying LUAD conditions...”. The authors report “...smaller differences (i.e. 130/471, Fig.2d)”, should not be 130/438?
2. Section 2.2. (page 7), for data regarding LYPLA1 and FAAH2, the correct figure is Extended Data Fig.5c.
3. In Section 4.2., the word “urea” is in capital letters.
4. Section 4.3. when referring to the phylogenetic tree, the authors cite Extended Data Fig.2. However, such a tree is shown in Extended Data Fig.1C.
5. Section 4.3., “Development of SH semi-specific assay library”, the referenced Extended Data Fig.2 should be Extended Data Fig.1C.
6. Section 4.3., subsection “Selection of stable internal proteome...”. The authors mention they followed a five-fold selection criteria, however, only 4 criteria are detailed.
7. In Section 4.8., change room temperature by RT
8. Section “4.8. Statistical analysis” should be numbered 4.9.
9. Indicate all centrifugation speeds in g, not in rpm.
10. Some acronyms are listed more than once throughout the text (e.g., FA, ACN).
11. Some units are differentially mentioned in the text, e.g., minutes and min, ml and mL. Please, correct them accordingly.

Reviewer #5

The manuscript describes, dd-ABPP, a methodology that measures the activity of a whole class of enzymes by data independent acquisition (DIA) mass spectrometry-based proteomics. Essentially a biotin labelled substrate homologue and streptavidin coated beads fish out the active enzymes that convert the bait substrate. Quantitative LC-MS analysis of the captured fraction is then combined with quantitative LC-MS analysis of the total protein content and the ratio of the captured enzyme to total enzyme is a measure of activity in the sample. This ratio is then compared between cohorts, and additional biological insight can be

gleamed about the targeted class of enzymes and their role in the investigated condition. Overall, the manuscript presents compelling evidence from multiple angles that this methodology is sound and presents novel findings that would be missed by typical LC-MS proteomics that quantified protein quantity regardless of activity.

Although the title of the manuscript narrows the scope of the investigation to lung cancer, major question about the universality of dd-ABPP arise: Does dd-ABPP work with other classes of enzymes? Is this discrepancy between active and total enzyme a symptom of cancer, or does this happen in other diseases? Is proteomics missing most of the picture by focusing on only total protein quantity? Given the obvious labor that went into this manuscript it would not be reasonable to ask for further investigation, but inclusion of data from other applications and enzyme classes, if available, in the discussion would help with answering these questions and would enrich the manuscript.

Major issues

It is not very clear how the tumor sections and patients were associated. Were the tumor sections combined for each patient? Were the sections from different tumors treated as different data points? In section 2.2 page 5 it says that 6 tumors and 6 non tumors were analyzed from long term and short term survivor patients (so 12 tumor and 12 nontumor), in table 1 there are 10 male and 6 female patients listed, then Fig 2 b is back to 24 biosamples, but there are 3 intra-tumor replicates and >50 paired tissue extracts. Clarification is needed.

Minor issues

Fig.1a middle panel and Fig1c second panel seem redundant.

Page 8 “five kay proteins” should be “five key proteins.”

We thank the reviewers for their comprehensive and helpful reviews. We found the arguments constructive and believe that our additional work has appropriately addressed their concerns. Below we summarize the changes made in our revised manuscript to address the reviewers' concerns. Changes are marked in the text of the revised manuscript file R1, which is dedicated to the peer review process.

Reviewer #1 (Remarks to the Author):

This manuscript describes the development of "depletion-dependent activity-based protein profiling (dd-ABPP)" methods and their applications in investigating lipid remodelling during lung adenocarcinoma progression. Traditional ABPP depends on chemically modifying a specific set of enzymes using a synthetic warhead, and these modified enzymes are enriched through streptavidin-biotin interactions. This method offers several apparent advantages. For example, ABPP assesses the activity status of enzymes by producing covalent bonds through catalytic activity. Unlike many conventional proteomic methods, which measure abundance, ABPP can reveal distinct dimensional information. In this study, the authors propose a slightly altered method, using parallel datasets of normal (or whole proteome) versus ABPP. Interestingly, they remove proteins conjugated with the warhead using streptavidin beads, instead of enriching them, an approach that is novel compared to other ABPP studies. This significantly improves MS analysis by reducing contaminant spectra from streptavidin peptides generated during the tryptic digestion process. It should also be noted that the foundation of dd-ABPP relies on high-coverage SWATH/DIA proteomic methods. Since the target enzymes are not effectively detected without the enrichment step, it is impossible to conduct this type of negative detection method. Throughout the manuscript, the authors thoroughly investigate controls and present convincing results. Therefore, this reviewer agrees with the acceptance of this manuscript for publication in Nature communications.

Response: We would like to thank the reviewer for the positive feedback and for highlighting the novelty of our approach.

Here are some minor comments:

1. In ABPP methods, enzyme activity is crucial. It would be beneficial to add discussions about the nature of patient samples and the enzyme activity status. For frozen or OCT-embedded tissues, how are enzymes kept in an active state?

Response: We appreciate the reviewer's comment and have clarified this point in the revised manuscript R1. Each tissue sample is rapidly frozen after surgical removal, quickly embedded in optimal cutting temperature (OCT) compound and stored at -80°C until analysis. Prior to analysis, the OCT-embedded tissue is used for frozen sectioning at optimal cutting temperature (-20°C) using a cryostat to minimise tissue degradation and preserve the protein structure and function. Indeed, we found no significant difference between fresh-frozen (FF) and OCT-embedded tissue material in protein and peptide numbers detected in the standard proteomics (**R1 Supplementary Fig1. a**). Revised manuscript also contains additional figure showing no substantial difference in capitation of active enzymes between FF and OCT-embedded tissue (**see below R1-Supplementary Fig. 1 b, Beads digest, ABPP workflow**). Of note fresh-frozen human tissue is already considered as optimal material compatible with standard ABPP workflow (Jessani et al., 2005) - Please refer to the reference list in the point-to-point file.

R1 Supplementary Fig1. a-b, Venn diagram shows overlap of protein identification between sample types, frozen (FF) and OCT-embedded lung tissue. Barplots with corresponding standard error (n=2-4) represent the number of proteins and peptides identified from frozen FF and OCT-embedded tissue in total digests (**a**), or number of captured enzymes in beads digest (**b**), respectively.

Although enzyme activity is well preserved in OCT-embedded tissue, the downstream analytical step of "removal" of the OCT material from tissue is the critical point for preservation of enzyme activity status. For example, prior to active enzyme labeling at physiological pH, organic solvents and strong detergents should be avoided in protein/enzyme extraction. To ensure that the extracted enzymes remain active, we have optimized the procedure for extracting proteins from OCT-embedded patient samples. The importance of this procedure is highlighted in the main text and in the Methods section. Please see:

- **Page 2-3, line 94 (Introduction):** " *Chemical removal of embedding material using detergents or organic solvents may result in loss of enzyme catalytic properties.*"
- **Page 4, line 157 (Result section 2.1):** " *First, in order to maintain the enzyme activity status, the dd-ABPP extraction protocol for OCT-embedded tissue sections was optimized. Proteins were extracted from each tissue sample by cleaning the OCT embedding medium in a two-step process (Method).*"
- **Page 14, line 714 (Method section 4.2.):** " *To remove the OCT compound prior to protein digestion and LC-MS analysis and to preserve enzyme physiological activity for probe labelling, we used two-step OCT cleanup process...*"

2. Although comprehensively described in the main text, Figure 1 is not reader-friendly.

Response: We agree with the reviewer on this specific point. We have modified Figure 1 (**R1-Figure 1**) by combining the content of two panels into a single panel and summarizing our proteomics approach in a more concise manner. To gain a better overview of the whole study, we have also included the other omics analyses used in this study in panel b (see below **R1-Figure 1b**, Multi-omics data analysis).

3. In Figure 1-b, the purple color representing "bead" and "beads digest" needs to be more distinctly differentiated.

Response: To avoid confusion with the standard protocol and bead-based protein digestion, we have changed "bead digest" to "bead pellet" within the novel illustration (**R1-Figure 1a**). The purple circle represents the bead (bead pellet) remaining after supernatant collection, supernatant which is used for in-solution protein digestion in novel dd-ABPP-SWATH/DIA-MS analysis.

4. In Figure 1-c, the retention time of peptide1 compared to the "total extract" and "depleted extract" should be clarified. The activity should represent the difference between the total and depleted extracts, but the dark blue colour area does not consistently reflect this.

Response: We agree that the retention time (RT) label was confusing, specifically because this RT should hypothetically be the same for the same peptide in two different MS injections of the paired patient samples. We have changed this. Please see below the new figure in **R1 - Figure 1a - Bioinformatic Analysis**.

a dd-ABPP DIA-MS

Preparation of paired extracts

Bioinformatic analysis

SWATH-MS data processing

dd-ABPP data output (Patient 1)

ID	Peptide	Intensity
Enzyme A	ABCDDE	X
Enzyme A (active)	ABCDDE	Y
Protein B	BBCDEEF	Z

b Multi-omics data analysis

Data integration

R1-Fig. 1. Overview of the dd-ABPP-SWATH/DIA-MS workflow. a, Schematic of clinical sample analysis in dd-

ABPP DIA-MS. Sample preparation (upper panel): Protocol starting with mechanical OCT removal from OCT-embedded tissue, PBS washes, and tissue homogenization. Individual tissue extracts were incubated with reaction solvent DMSO (total extract) or biotinylated chemical probe FP-biotin (depleted extract), and active enzymes removed on streptavidin beads. The soluble proteomes were recovered in the remaining supernatants of the paired extracts and analysed using an automated SWATH/DIA-MS platform that included software tools for targeted spectral library search, MS raw data alignment and quantification. **Bioinformatic analysis (lower panel):** Paired sample records were normalized by consistently detected fragment ion chromatograms of a set of endogenous ISP peptides selected from housekeeping and cytoskeletal proteins. The enzyme activity status was assessed from differences in complementary total and depleted sample records based on SWATH-MS peptide intensities (ΔInt) in a logarithmic scale (i.e., $\log_2 \Delta \text{Int}$).

b, Multi-omics data analysis. The integration of multiple levels of omics data, background total proteome, depleted proteome, and enzyme activity status across the same set of samples. Examples of cancer-related regulation of enzyme activity status detectable by dd-ABPP-MS pattern. $[\text{E}]_{\text{act}}$, $[\text{E}]_{\text{inact}}$, and $[\text{E}]_{\text{tot}}$ correspond to active, inactive, and total enzyme forms. FP-biotin ABP corresponds to biotinylated fluorophosphonate.

- Regarding Figure 5d, the authors highlight the fourth most significant element in their Gene Ontology (GO) analysis. It would be beneficial to add the rationale for not focusing primarily on the top three aspects.

Additionally, please clarify the selection criteria used, particularly concerning protein depalmitoylation. GO analysis, highlighted 5 top following GO terms: lipid catabolic process, macromolecule depalmitoylation, macromolecule deacylation, protein depalmitoylation and protein deacylation.

Response: We appreciate the reviewer’s comment. Protein de-palmitoylation (GO:0002084) also constitutes the top five GO biological processes listed in Fig. 5d (i.e. macromolecule de-palmitoylation, macromolecule deacylation, protein de-palmitoylation and protein de-acylation), which are all lipid catabolic processes in a broader sense (see below the anchor GO:term diagram). Depending on the chemical terminology, the removal of palmitoyl groups from a macromolecule (i.e. protein) is known as protein depalmitoylation. Protein depalmitoylation is often referred to as protein de-acylation or the removal of an acyl group (palmitoyl group) from a macromolecule (F et al., 2024) To avoid confusion, we have now highlighted all top GO terms in the text. **Page 10, line 479 (Result section 2.4):** “The Gene Ontology (GO) enrichment analysis integrating enzymes and co-depleted proteins in the discriminant model suggested significantly accelerated lipid catabolic process, macromolecule de-palmitoylation, macromolecule de-acylation and, protein de-palmitoylation (FDR <0.001) via enrichment of several palmitoyl hydrolases.”

Reviewer #2 (Remarks to the Author):

Sajic et al. present an interesting clinical study on lung adenocarcinoma progression. The presented workflow is based on an automated DIA-MS workflow combining depletion-dependent activity-based proteome profiling. The protein profiles associated with active enzymes could classify an aggressive advanced-stage lung adenocarcinoma using tissue microsections. The manuscript is well written and a multiomics approach was shown, including information layers collected from proteomics, targeted lipidomics, metallomics, and genomics analyses.

Response: Thanks to the reviewer for the overall positive feedback.

One significant outcome presented was the increase in monounsaturated lipids in aggressive tumours. Although the value of the clinical cohort on LUAG and the analytical novelty on the proteomics side are apparent, and the metallomics data on smoking patients are interesting, several pieces of information need to be added on the lipidomics side. Unfortunately, the presented lipidomics data does not follow the minimal requirements to confirm the hypothesis of an increase in monosaturated lipids in aggressive tumors. Therefore, the quality of the presented work is only suitable for Nature communication with additional experiments and major manuscript changes.

Response: We completely agree with the reviewer that this part was incomplete. To address the reviewer's concerns in the revised manuscript, we have:

1. corrected and clarified the initial lipidomic analysis, which we called free fatty acid (FFA) analysis to avoid confusion and which represents a complementary validation part of the proteomic study.

2. performed additional high-quality experiments based on authentic stable isotope-labelled internal standards on an independent platform equipped with Orbitrap™ IQ-X™ Tribrid™ mass spectrometer for the final identification and quantification of four fatty acids of interest (palmitic acid, palmitoleic acid, oleic acid and eicosapentaenoic acid).

3. removed all panels related to monounsaturated fatty acid (FA) levels from the current **R1-Figure 6**. In particular, and as mentioned by the reviewer, we recognise that our selection of a small subset of fatty acids is not sufficient to draw a general conclusion about overall levels of monounsaturated lipids in aggressive tumours. In the **R1- Figure 6**, the old panels are replaced with the results of new experiments related to the absolute quantification of the FA of interest (**R1-Figure 6. c-d-e-f**). We have changed the manuscript and the corresponding Figure 6 accordingly, as we have clarified below in specific comments from reviewer 2.

Part of the R1 Figure 6 (a-f) FFA analysis reveals the excess of palmitate and its monounsaturated metabolites in the tumors of the aggressive LUAD phenotype. **a**, Relative comparison of selected FFA levels obtained through a full scan untargeted LC-HRMS profiling. FFA annotation was done by the accurate mass and retention time (AMRT) matching against standard database. No MS/MS spectra are available due to well-known low fragmentation efficiency of fatty acids in negative ionization mode. Barplots with data points represent levels of annotated FA in LUAD tumors of long- and short-survival patients. FAs compared using a two-sided Student's t-test (n=5 samples per LUAD subtype). **b**, Protein expressions of Stearoyl-CoA Desaturase (SCD1), (n=7/group). GAPDH is used as loading control. Boxplot represents mean \pm S.E.M expressed as arbitrary units normalized to the mean of long-term survival subtype. P-value from two-sided t-test. **c-d-e-f**, Bar plots with data points represent the absolute tumour concentration of the FAs of interest, hexadecenoic/palmitic, palmitoleic, oleic and eicosapentaenoic acids, respectively, determined through the internal standard spike using authentic isotopically labeled standards and multipoint calibration curves. p-value corresponds to two-sided Wilcoxon signed rank test. (see also Figure 6 in the manuscript)

Major comments:

Lipid sample preparation:

1. Extraction: What was the sample preparation for the lipidomics analysis? It seems like the starting point was 20 μ L.. but of which samples?

Response: This analysis represents a complementary validation part of the proteomic study and aimed to address the FFA status due to the altered catalytic activity of enzymes involved in palmitic fatty acid metabolism in aggressive tumors (see **R1-Figures 3h** and **5d-e**). For this purpose, we collected tumor tissue samples (N=5 samples per LUAD tumor subtype from 2nd validation cohort). We homogenized on the average 30 microsections per each individual sample, corresponding to approximately 8 mg of lung tissue in 450 μ L phosphate-buffered saline (PBS) solution at physiologic pH. An aliquot (20 μ L) of these homogeneous solutions was used for subsequent lipid extraction and analysis. Lipid extraction was performed by the addition of 80 μ L isopropyl alcohol (IPA) as described by Medina et al. Please refer to the reference list in the point-to-point file (Medina et al., 2023; Medina et al., 2020). Following centrifugation, the resulting supernatant was collected and transferred to LC-MS vials for injection. Please refer to manuscript **Page 23, line 1089 (Method section 4.5; Measurements of FFA- Sample Extraction)**

As suggested by the reviewer we add more details and rewrite the FFA analysis from scratch. For clarification, and because additional experiments were performed within revised R1 manuscript., the method **section 4.5. Measurements of FFA** contains three parts: **i)** Sample Extraction, **ii)** Full scan untargeted LC-HRMS profiling of FFA, **iii)** Absolute quantification of FFA using authentic internal standard spike

2. Standards: Were any internal standards added for fatty acid analysis (e.g. commercially available standards from Lipotype or Avanti) at least at the point of sample extraction?

Response: Yes, we used four corresponding (or authentic) stable isotope labelled internal standards (palmitoleic acid-d13, Oleic acid-d9, eicosapentaenoic acid-d5 and hexadecenoic acid-d3) purchased from Larodan AG (Solna, Sweden) for the quantification of selected, endogenous fatty acids (palmitoleic, oleic, eicosapentaenoic and palmitic (hexadecenoic) acid). This information has been added to the materials and method section (**Page 24, line 1151**).

3. Sample storage: It is not clear how long the samples were stored at -80°C and if there any oxidation-preventing reagents (e.g. BHT) were added for lipid analysis.

Response: We used common long-term storage of samples at -80°C in clinical biobank (i.e. several years), and no anti-oxidative agents were added to the samples. This is necessary for the analysis of oxidized lipids, such as oxylipins (i.e., eicosanoids), which are derived from polyunsaturated fatty acids (subject to enzymatic and spontaneous chemical oxidation). We have not analyzed oxylipins. Tissue samples were embedded in OCT and immediately stored at -80°C. Tissue was dissected at -20°C and all preparations were kept on ice. All sample preparation procedures were identical and performed in parallel.

4. As fatty acids/eicosanoids were analysed, suitable sample preparation, including the addition of internal standards, is crucial to ensure high-quality lipid results.

Response: We analyzed free fatty acids but not eicosanoids (derived from PUFAs). In the first instance, we have performed the untargeted profiling in a full scan mode to identify the potential changes across a wide panel of free fatty acids. The results of this discovery approach were then validated through the targeted quantification of specific FFA species of interest, using authentic isotopically labelled standards (Larodan AG, Solna, Sweden) and calibration curves. Material and method description was revised accordingly.

Please refer to our previous response and see manuscript page 23, line 1086, Method section 4.5.

5. Lipid analysis: The presented workflow is based on a bulk lipidomics profiling approach by RP-HRMS using an isopropanol gradient to elute lipids followed by recording a mass range of 100-1700 m/z with the QTOF. If the authors really intended only to acquire fatty acid profiles via targeted lipidomics as presented, the analysis method, e.g., the mass range, is suboptimal (all C8-C22 fatty acids table presented in Fig6 were within the mass range from 140-350 m/z => this would be ideal for a GC-MS FAMES analysis m/z < 500). It rather seems like a standard (non-targeted) lipidomics analysis was performed in the beginning and only the fatty acids profiles were included for data analysis. More information on the rationale of the method has to be provided:

Response: Thanks to the reviewer for his constructive comments. We have effectively, first, performed a full scan screen in a wide mass range 100-1700 m/z. The data was mined in a targeted way due to specific interest in fatty acids, to corroborate the results of proteomics experiments. Selected candidate FFAs were further quantified in a new experiment (revised manuscript R1) using a stable isotope dilution approach and calibration curves. As specified above, the manuscript, material and method description were modified accordingly.

In our initial analysis, samples were analysed by Liquid Chromatography High-Resolution Mass Spectrometry (LC-HRMS) operating in full scan mode in negative ESI on Agilent QTOF. The free fatty acid annotation was performed by matching the accurate mass and retention time (AMRT) against an in-house database recorded on pure standards of 25 different FFA species analysed in the same analytical conditions. As a result, among 14 detected and matched FFAs, the relative abundance of three species were significantly altered between tumors.

These three fatty acids and palmitic acid (highlighted through proteomic data), were recently quantified using the authentic internal standard (IS) spike on a newly acquired UHPLC-UHRMS Orbitrap IQ-X Tribrid system. For this targeted validation we narrowed the mass range to 180–500 m/z to maximize the sensitivity. The MS acquisition parameters are described in **Method section 4.5 (Page 24, line 1142)**. We performed all necessary background corrections and deposited raw data files, on samples, standard solutions and extracted blanks, within Zenodo data platform (<https://doi.org/10.5281/zenodo.13895748>.) Indeed, the analysis of FFA can be performed by GC-MS using a FAMES method (methyl ester derivatization). However, due to the instrumentation available and the limited amount of tumor biopsies, we measured the selected FFA by liquid chromatography coupled to high resolution mass spectrometry, as in previous LC-MS works (Park et al., 2021) (Koch et al., 2021; Li & Franke, 2011; Medina et al., 2023). Please refer to the reference list in the point-to-point file.

6. The authors presented a targeted lipidomics approach based on MS1 high-resolution analysis using an Agilent 6550 QTOF. Targeted lipid analysis is usually done with an SRM/MRM approach, including known retention time windows, specific lipid fragments (to avoid false annotations), and standards (to confirm the annotation). At least from the methods parts, it seems no MS2 information, standards, blanks (for blank subtraction), or known retention time information was used. If this is the case, the workflow is insufficient to prove the identity of the fatty acids analysed, as in-source fragments or lipid contamination (fatty acids are background contaminants) could be the reason for the recorded signals. If known retention time windows were used (e.g. from a previously acquired database) at least the equivalent

carbon number model (ECN) can be used to confirm the experimentally determined lipids. The authors must provide more information on the targeted method part and prove the reliability of the presented lipid annotations!

Response: We thank the reviewer for bringing up this important point. As described above, in the first untargeted experiment, the free fatty acid annotation was performed by matching the accurate mass and retention time (AMRT) against an in-house database recorded on pure standards of 25 different FFA species analysed in the same analytical conditions (see lipid annotations in **Supplementary data 7**). The results of untargeted analysis were recently validated by the subsequent targeted quantification using an authentic internal standard (IS) spike (**Page 24, line 1151**).

While we fully agree with the reviewer that the lipid quantification should/is usually done in the SRM/MRM mode on triple quadrupole instruments, for fatty acids this would have been less selective compared to high resolution MS due to **low fragmentation efficiency of fatty acids** (oleic and palmitic acid, specifically) and therefore, low signal intensity of MS/MS fragments for quantification with isotopically labelled standards. As shown in the Figure from m/z Cloud below, when operating in ESI negative mode (needed to achieve the optimal sensitivity for FFA analysis), FFAs are not fragmented. Due to the low fragmentation efficiency of FFA in LC-MS analysis, this family of metabolites are generally measured in a full scan mode (mass range 180–500 m/z) on high resolution instruments (in our case novel Orbitrap™ IQ-X™ Tribrid™ MS platform) or a pseudo-SRM/MRM mode based on precursor-precursor transitions (Koch et al., 2021), with authentic stable isotope labelled standards. Using the HRMS, the high mass resolving power and chromatographic separation allow for the measurement with optimal selectivity. Otherwise, we perform targeted lipid analysis daily in the context of a multitude of research projects, as recently published by Medina et al. 2023 (Medina et al., 2023).

As specified above, we performed all necessary background corrections and deposited raw data files, on samples, standard solutions and extracted blanks, within Zenodo data platform (<https://doi.org/10.5281/zenodo.13895748>).

Figure of Oleic acid fragmentation

7. What about the other bulk lipids such as PC, TG, etc? - How did the authors ensure that the analyzed fatty acids were not only in-source fragments from bigger lipids, e.g., PC?

Response: The chromatographic separation method used allows the separation between FFA and other complex lipids like PC and TG avoiding the contribution of FFA coming from the in-source fragmentation of other complex lipids.

8. Additionally, the correct normalization for the targeted lipid analysis is important: How was the data normalized? TIC and protein normalization are mentioned, but it is not clear at which point of the analysis the protein determination was performed.

Response: For the first untargeted experiment, we used the 20 μ L of tissue extracts in PBS solution for lipid extraction in FFA measurements. To account for analytical variability (e.g., "MS injection variability"), we normalized the annotated FA peak areas (ion counts) of each sample MS run based on the total ion current (TIC) of fatty acids measured in that sample run. We have also performed the pre-acquisition normalization to total protein content - to normalize for varying amounts of starting material. We reported original and normalized data in the **Supplementary data 7**.

9. Lipid data analysis: From the presented combination of C8-C22, including up to 6 DB positions, around 60 lipids can be produced. One significant outcome presented in the whole manuscript was the increase in monounsaturated lipids in aggressive tumors (Fig 6a+b).

Although palmitoleic acid 16:1 and oleic acid 18:1 are usually the most prominent fatty acids, the authors cannot conclude that monounsaturated lipids are upregulated in aggressive tumors in general. The presented sum parameter in Fig 6b is misleading as it shows unsaturated fatty acids, which include both mono- and polyunsaturated fatty acids.

Response: We agree with the reviewer that theoretically other mono-unsaturated fatty acids could be present in a given biological sample (14:1; 16:1; 18:1; 20:1; 22:1.). Indeed, a selection of a small subset of fatty acids is not sufficient to draw a general conclusion about the overall levels of mono-unsaturated lipids in aggressive tumors. Therefore, we have removed all graphs related to this overstatement from our revised manuscript.

10. Fig 6a: Which were the bases for selecting and ranking the 14 presented fatty acids/eicosanoids from source data Fig 6?

Response: The analysis/selection of fatty acids (but NOT eicosanoids which are derived from PUFAs) were complementary validation part to proteomics results. Due to altered serine lipase activities involved FAs processing and enrichment of protein de-palmitoylation in aggressive tumor subtypes (**R1-Figure 5d**, GO analysis, $FDR \leq 0.01$), we mainly focused on measuring palmitic acid (PA) levels and other FAs related to PA metabolism in tumor subtypes. For the initial profiling we extracted the data in a targeted way using a list of 25 FFA for which the data were available within *our in-house standard database*. Of these 25 we detected 14 fatty acids in each individual patient tissue, including palmitic acid and their most common metabolites, palmitoleic and oleic, acid. To clarify this, we modified the language in the text of **Result section 2.5., page 11, line 502:** " *To experimentally investigate our assumption that aggressive tumors display lipid remodeling in LUAD via increased rate of lipolysis and protein de-palmitoylation (Fig. 5d–e), we analyzed the sections of the ten tumors (i.e., 5 per each LUAD subtype, validation cohort) for a panel of 25 saturated, monoenoic and unsaturated FA levels (C8–C22) with a primary focus on palmitic (hexadecenoic acid, C16:0) acid metabolism. We performed an initial profiling of FAs species using liquid chromatography high-resolution mass spectrometry (LC-HRMS) and detected 14 FAs species.*"

11. Why is the plot separated in two? Is the separation of the plots based on the normalized areas? Why, after C8:0, C10:0, C12:0, a C20:5 is listed, followed by C18:3? An alphabetic order or order by fatty acids lengths and number of DB would make more sense. In the methods part, CV < 30% was mentioned; was this the selection parameter?

Response: The initial separation of the plot on two was done to improve the readability of different ranges of peak areas. We have now modified this plot based on i) two ranges on the X-axis and ii) we have ordered the detected FA features according to the number of carbons present in putative lipid species (i.e. C8:0 → C22:0). See above panel a, for **R1-Figure 6** related to FFA measurements.

Fig 6b: The authors conclude that the levels of saturated versus monosaturated and polyunsaturated were lower in aggressive tumors. Was this information derived on the limited sample set of 14 fatty acids (CVs less than 30%)? For (relative) quantification comparisons of lipids in different groups internal standards are essential. - What does the text description "Arbitrary Units normalized to the mean of the observations in the long-term survival subtype" in the Figure (6) caption mean?

Response: We completely understand the reviewer's concern and have now corrected this conclusion, modified the text, and removed the corresponding plots (in arbitrary units) related to monosaturated and polyunsaturated fatty acids. (See new **R1-Figure 6 c-d-e-f** does not contain these data and panels are replaced with absolute quantification results of FA of interest)

12. Please follow the FAIR principles: (1) No lipidomics raw data is available: The raw data and annotated metabolite and lipid lists must be made accessible to the community (e.g., GNPS, MetaboWorkbench, MetaboLights or in your case, you could also add it to PRIDE with more information on the lipid sample preparation) as it was done for the proteomics data; (2) the minimal reporting checklist for the lipid part should be included and also presents a way of checking the quality of the presented lipid annotation for the authors, the referees and the community: https://lipidomicstandards.org/reporting_checklist/

Response: As suggested by the reviewer and according to the FAIR data principles, we have now made available raw data files corresponding to fatty acid measurements performed on two independent analytical platforms (two experiments). Raw data files on samples, standards and extracted blanks related to the FFA analysis as a complementary validation part to the proteomic data have been made publicly available via the Zenodo platform <https://zenodo.org/> (DOI 10.5281/zenodo.13895747). Reporting checklists for the two independent experiments are now generated on <https://lipidomicstandards.org/report/> and are available within the Supplementary Information file (please refer to **Supplementary Note 3**).

Other comments

13. Page 3: The argumentation line on why DIA was used instead of DDA does not make sense from an analytical point of view. Please clarify and use more recent references.

Response: We agree with reviewer constructive remark that we need to add more recent references related to DIA-MS operating mode in sample analysis. At the request of the reviewer, we add the most recent references on different software and variants of DIA-MS analysis (Meier et al., 2020; Sinitcyn et al., 2021; Yu et al., 2023) in line with the work done with SWTAH/DIA-MS (Rost et al., 2016; Rost et al., 2014) and we add DIA-MS operating mode examples for clinical use (i.e. (Niu et al., 2022),(Guo et al., 2015)) . For more information please refer to Reference list of this file.

Although both MS acquisition modes (DIA and DDA) are used to detect molecules of interest, the concept of DIA acquisition has dramatically improved the quality of MS spectra. This is, of course, consistent with improvements in MS instrumentation. Unlike DDA, in DIA mode there is no stochastic selection of precursor ions for MS2 sequencing, and entire MS1 maps are acquired. The DIA spectra are more complex to analyze, and recent developments in bioinformatics tools for spectral deconvolution and especially spectral alignment for identical peptides detected across multiple samples have significantly improved the quality of the DIA analysis (Rost et al., 2016; Sinitcyn et al., 2021; Yu et al., 2023).

Notably, the integration of the TRIC algorithm (Rost et al., 2016) into the SWATH/DIA-MS workflow allows a more accurate calculation of the depletion ratio between peptides of two complementary sample sets (total and depleted) in dd-ABPP. The use of SWATH/DIA-MS is important for the consistency of peptide detection across extracts and their alignment across MS runs for accurate estimation of the depletion effect in the dd-ABPP workflow.

Language clarified in the text with additional references, manuscript Page 3, Line 100 (Introduction): “Third, recent advances in peptide quantification based on data-independent MS acquisition (DIA)^{29-31 32-34} have not been sufficiently exploited in activity based profiling³⁵. The DIA-MS mode ensures consistency of sample analysis by generating a complete sample data record of all ionized peptides without instrument-driven criteria. Bioinformatics tools have been developed to improve the analysis of complex DIA data records, including reproducible signal mining, and quantification accuracy based on MS2 intensity trace analysis. Previous studies have used fluorescence-based in-gel protein quantification or on-bead peptide quantification based on data-dependent MS acquisition (DDA-MS)²³, which leads to sparse matrices when sample cohorts are analyzed, and inconsistent MS1 peak-area quantification across samples³⁶”.

14. For a non-proteomics expert, the novel generation of dd-ABPP is not entirely clear. Please introduce your workflow and its novelty in a more comprehensive way, maybe by using an additional (Supp.) Figure? Figure 7 is a bit confusing in its current form. As you also present multi-omic layers, a workflow figure for all performed analyses would enhance the reader’s understanding of your workflow and its novelty

Response: Thanks, reviewer, for this constructive remark. We have included **Supplementary Note 2**, which summarizes the characteristics of each method for studying enzyme families, the new generation of dd-ABPP and the standard ABPP approach, as suggested by the reviewer. Please also see our new R1-Figure 1b, which integrates multi-omics levels of data analysis within the workflow. This modification of Figure 1 introduces our work from biggening and clarifies the data presented in **R1-Figure 7**.

Supplementary Notes 2

Method	Description of procedure	Quantitative data reports	Method advantages
dd-ABPP SWATH/DIA-MS 	Data acquisition  DIA-MS with spectral library Protein digestion  In-solution digestion Analysed samples  Total tissue extract Depleted tissue extract for desactivated enzymes 	 Fraction of active enzymes Abundance of enzymes Abundance of proteins (contextual proteome) Protein interactors co-depleted with active enzymes 	 Streamlined in-solution protein digestion No streptavidin contamination No time-consuming beads washing Total and active form of enzymes of interest available Contextual sample proteome and protein interactors with enzymes available
ABPP-MS 	Data acquisition  DDA-MS Protein digestion  On-bead digestion Analysed samples  Streptavidin beads pull-down enriched for desactivated enzymes Streptavidin beads pull-down with contaminants 	 Active enzymes Accessory non-enzyme proteins (potential interactors) 	 Detection of probe-inactivated enzymes that allows specific discovery of new members

15. The study design and the applied number of used tissue microsections are very confusing in the method sections. Why did you start with 24 samples and only show 16 in source data Table 1? The tumor and non-tumor samples seemed derived from the same 16 persons. Please provide a more comprehensive list or visualization in the supplementary to explain these numbers and clinical cohort and cohort 2.

Response: We appreciate the reviewer’s comment. Our revised manuscript contains now **Supplementary Figure 3** that summarise number of patients, biospecimens and microsections in each cohort. The type of analysis performed on each set of samples was noted. In summary, our entire study was performed on 16 patients, corresponding to 32 different biospecimens, as we collected tumor and adjacent lung tissue for each patient.

Supplementary Figure 3

16. ABP and ABPP: It is a bit misleading to have such close abbreviations used throughout the manuscript, if possible please clarify

Response: We corrected this, and we now only used ABPP abbreviations corresponding to “ Activity-based protein profiling “ and reflects to the proteomics approach.

Reviewer #3 (Remarks to the Author):

Response: We thank the reviewer for his work on this manuscript.

Reviewer #4 (Remarks to the Author):

Sajic and collaborators describe an approach that combines already existing technologies for the comprehensive characterization of enzyme activities. Despite they do not introduce any new techniques, the description of their SWATH/DIA-MS workflow combined with an ABPP method is interesting and has been well

conducted. The authors have focused their investigations on the involvement of serine hydrolases in advanced-stage lung adenocarcinoma, as a proof-of-concept. It is also worth mentioning the extensive data analysis performed throughout the study, including ICP-based metallomics screening, enzyme-activity tests and orthogonal validations. Overall, the topics addressed in this manuscript are timely and of relevance to be published in Nature Communications. Nevertheless, the paper needs some modifications.

Response: We appreciate the reviewer's positive feedback on our study and hope that we have addressed the constructive comments made by the reviewer.

1. Fig.4b: I would like to see the quantification of the gel area of the control proteins (actin and GAPDH).

Response: We thank the reviewer for these constructive comments. We have now normalized the fluorescence area in **R1-Fig. 4a** to the housekeeping protein areas (GADPH + actin area). Please see below. New plots correspond to TAMRA-FP fluorescence intensities quantified on gel and normalized to housekeeping protein areas. Individual values for actin area and GAPDH area gel quantification are provided in **the respective source data tables** (Source data figure 4a).

2. Section 2.3. (page 8). The authors state that “protein patterns extracted from tumor and surrounding tissue displayed approx. two-fold higher fluorescence intensities for tumor samples”, but this is not the case for all samples analysed. The authors should further comment on this.

Response: Indeed, this was not a precise formulation and specifically for patients number 4, 7 and 10 after normalization with control proteins as suggested by the reviewer (see Figure, above). We rephrase this sentence (**Page 8, line 369; Results section 2.3**) to: “Protein patterns extracted from tumor and surrounding tissues showed higher fluorescence intensities for tumor samples compared to their respective adjacent tissues (**Figure R1-Figure. 4b**)”.

3. Section 2.3. (page 8). The 6 out of 13 SH that classified LUAD tumors should be mentioned. Also, when referring to the corresponding Figure.3g, the authors should explain the criteria for selecting those 6 SH.

Response: We thank the reviewer for bringing up this important point. The dd-ABPP-SWATH/DIA-MS data showed that as many as 17 different serine hydrolases increased their activities in LUAD tumors compared to adjacent tissues. For example, based on their overall high activities in LUAD tumors, 13 SHs (but not all 17) almost accurately classified tumors and adjacent tissues in both LUAD subtypes (**R1-Figure 3g, source data Figure 3g**). In conventional ABPP (validation part), we detected only 6 out of 13 highly active tumor SH enzymes with ability to classify tumorigenesis. Notably, these 6 enzymes confirmed an increase in activity in tumors with orthogonal method and even 5 out of 6 showed statistically significant changes. Their names are given in the text **Page 8, line 384** (SIAE, PREP, FASN, DPP4, PRCP, ELANE; **R1-Figure 4b**). The data for 13 highly active tumor enzymes are in **source data Figure 3**.

4. I was interested in looking at the stable internal proteome list, but I could not find any table in the additional files. Could the authors be so kind as to provide such list?

Response: Please note, we now included 96 ISP peptides (originating from their corresponding 21 proteins) that are used as stable internal standards in the dd-ABPP experiment. The list of these peptides is in **Supplementary data 2**.

5. Did the authors note any correlations/associations between gender and SH activities?

Response: The lipase ABHD10 show the significant positive association between their enzyme activities and the female sex (β coefficient =0.859652, $p=0.046$; mean (male)= 0.09697, mean (female) = 0.2032;

LYPLAL1 show positive associations with female sex (β coefficient =0.60, $p= 0.0221$); mean (male)=0.3734, mean (female) = 0.6248; and age (β coefficient =0.038, $p=0.0385$). However, most of the measured enzymatic activities do not show associations with gender or age in the LUAD and Lung-adjacent tissues. We believe that application of the method to other human tissues (e.g. hormone-sensitive tissues) for SH measurements would most likely reveal these differences.

6. Section 4.3., subsection "Generation of phylogenetic tree of SH family members". The authors state: "We catalogued a total of 335 proteins", but the indicated proteins sum up 350. I assume the authors have not included in this calculation the 15 inactive enzymes. However, in Extended Data Fig.1C, these 15 proteins are pointed out. Please, double-check all these numbers.

Response: We thank reviewer for this comment. We corrected this section.

Please see sentence highlighted in yellow, page 18, line 869, section 4.3., revised manuscript. "**Generation of phylogenetic tree of SH family members**": "*We catalogued a total of 335 proteins (including 15 inactive enzymes): 175 serine proteases, 134 metabolic serine hydrolases and 26 non-annotated cases. Specifically, 294 proteins were documented as SHs, including SHs with putative activity or SHs for which catalytic activity was not directly demonstrated. (e.g., human Haptoglobin (HP) classified in Merops, Serotransferrin (TF), Merops S60.972)*".

Minor comments:

7. Section 2.2., subsection "Classifying LUAD conditions...". The authors report "...smaller differences (i.e. 130/471, Fig.2d)", should not be 130/438?

Response: Our formulation was correct since 471 is total number of differentially changed proteins in four comparisons of interest, while 438 is number of proteins changed between tumors and controls. However, we have rephrased this to (i.e. **130 vs. 438, Figure. 2d**) because it seems more intuitive to read it this way. (Results section 2.2., Page 6, line 252)

8. Section 2.2. (page 7), for data regarding LYPLA1 and FAAH2, the correct figure is Extended Data Fig.5c.

Response: Thanks. We changed this part.

9. In Section 4.2., the word "urea" is in capital letters.

Response Thanks. We corrected term "urea" in capital letters.

10. Section 4.3. when referring to the phylogenetic tree, the authors cite Extended Data Fig.2. However, such a tree is shown in Extended Data Fig.1C.

Response. Thanks to the reviewer for pointing this out. We have corrected this.

11. Section 4.3., "Development of SH semi-specific assay library", the referenced Extended Data Figure 2 should be Extended Data Figure 1C.

Response. Thanks to the reviewer for pointing this out. We have corrected this. (now is Supplementary Fig 1C)

12. Section 4.3., subsection "Selection of stable internal proteome...". The authors mention they followed a five-fold selection criteria, however, only 4 criteria are detailed.

Response Indeed, this selection five-fold step process was incomplete. We have corrected this.

13. In Section 4.8., change room temperature by RT

Response. We changed this.

14. Section “4.8. Statistical analysis” should be numbered 4.9.

Response. We have corrected this error.

15. Indicate all centrifugation speeds in g, not in rpm.

Response. We applied centrifugation speeds in g.

16. Some acronyms are listed more than once throughout the text (e.g., FA, ACN).

Response. We check and correct this.

17. Some units are differentially mentioned in the text, e.g., minutes and min, ml and mL. Please, correct them accordingly.

Response. We thank reviewer on these points that we corrected in revised R1 manuscript.

Reviewer #5 (Remarks to the Author):

The manuscript describes, dd-ABPP, a methodology that measures the activity of a whole class of enzymes by data independent acquisition (DIA) mass spectrometry-based proteomics. Essentially a biotin labelled substrate homologue and streptavidin coated beads fish out the active enzymes that convert the bait substrate. Quantitative LC-MS analysis of the captured fraction is then combined with quantitative LC-MS analysis of the total protein content and the ratio of the captured enzyme to total enzyme is a measure of activity in the sample. This ratio is then compared between cohorts, and additional biological insight can be gleaned about the targeted class of enzymes and their role in the investigated condition. Overall, the manuscript presents compelling evidence from multiple angles that this methodology is sound and presents novel findings that would be missed by typical LC-MS proteomics that quantified protein quantity regardless of activity.

Response: We thank the reviewer for highlighting the novelty and advantage of our approach compared to standard proteomics approaches. We would also like to thank the reviewer for his/her professional and valuable comments.

1. Although the title of the manuscript narrows the scope of the investigation to lung cancer, major question about the universality of dd-ABPP arise: Does dd-ABPP work with other classes of enzymes? Is this discrepancy between active and total enzyme a symptom of cancer, or does this happen in other diseases? Is proteomics missing most of the picture by focusing on only total protein quantity? Given the obvious labor that went into this manuscript it would not be reasonable to ask for further investigation, but inclusion of data from other applications and enzyme classes, if available, in the discussion would help with answering these questions and would enrich the manuscript.

Response: We appreciate these comments from the reviewer. While we did not test the other class of enzymes with our dd-ABPP SWATH/DIA-MS, we established a workflow that could be readily adopted for large-scale screening of other enzyme families for which activity-based probes are available (Cravatt et al., 2008)- including, for example, matrix metalloproteases (Sieber et al., 2006) and cysteine proteases (Joyce et al., 2004). The activity of these enzymes are prominent targets in cancer, but also in other human pathologies related to inflammation (Ravindra et al., 2018) and host-pathogen interactions in infectious diseases (Lentz et

al., 2018; Puri et al., 2012). We clarify this in the discussion part of revised manuscript. (Please refer to the reference list in the point-to-point file).

- Is proteomics missing most of the picture by focusing on only total protein quantity?

Response: Standard quantitative proteomics lacks the ability to directly identify the molecular forms of the proteins being quantified (including the active and inactive forms of enzymes). Even if the analysis reveals differential changes in protein abundance under different conditions, focusing on total protein levels alone, without knowing which form of the protein is being changed, may not provide full insight into downstream biological effects. We believe that workflows based on chemical affinity enrichment of protein forms of interest (e.g. catalytic enzymes, PTM-modified proteins) would further benefit from the simultaneous standardization of their measurements to the biological levels of the contextual proteins in time and space to obtain a more complete picture of biological condition. Notably, the readouts from our approach (i.e., levels of enzymatic activity) do not overlap with the results of current proteomic technologies but complement these other technologies to better understand disease mechanisms. We comment this in **Discussion** (See emphasized text on page 14, line 644).

Major issues

2. It is not very clear how the tumor sections and patients were associated. Were the tumor sections combined for each patient? Were the sections from different tumors treated as different data points? In section 2.2 page 5 it says that 6 tumors and 6 non tumors were analyzed from long term and short term survivor patients (so 12 tumor and 12 nontumor), in table 1 there are 10 male and 6 female patients listed, then Fig 2 b is back to 24 biosamples, but there are 3 intra-tumor replicates and >50 paired tissue extracts. Clarification is needed.

Response: We thank the reviewer for these constructive comments. The part about clinical tissue samples was indeed unclear. Our revised manuscript now includes **R1 Supplementary Figure 3** (Please see above, comment 16, Reviewer #2) which summarizes the number of patients, biospecimens, and microsections in each cohort. In summary, our total study was performed on 16 patients corresponding to 32 different biospecimens, as we collected tumor and adjacent lung tissue for each patient. In our initial dd-ABPP DIA-MS proteomics, we analyzed 24 biospecimens (12 patients with paired tumor and adjacent tissue).

3. Minor issues

Fig.1a middle panel and Fig1c second panel seem redundant.

Response: We appreciate this valuable comment and, as reviewer suggested, we have redesigned Figure 1 in a simpler and clearer way, combining the contents of two panels into a single Figure 1a. Please refer to the new **R1-Figure 1** in the manuscript. (Please also see our response to reviewer #1's comment 2-3-4 above.)

Page 8 “five kay proteins” should be “five key proteins.”

Thanks. We have corrected this error.

Reference list (point-to-point file)

- Cravatt, B. F., Wright, A. T., & Kozarich, J. W. (2008). Activity-based protein profiling: from enzyme chemistry to proteomic chemistry. *Annu Rev Biochem*, *77*, 383-414. <https://doi.org/10.1146/annurev.biochem.75.101304.124125>
- F, S. M., Abrami, L., Linder, M. E., Bamji, S. X., Dickinson, B. C., & van der Goot, F. G. (2024). Mechanisms and functions of protein S-acylation. *Nat Rev Mol Cell Biol*, *25*(6), 488-509. <https://doi.org/10.1038/s41580-024-00700-8>
- Guo, T., Kouvonen, P., Koh, C. C., Gillet, L. C., Wolski, W. E., Rost, H. L., Rosenberger, G., Collins, B. C., Blum, L. C., Gillissen, S., Joerger, M., Jochum, W., & Aebersold, R. (2015). Rapid mass spectrometric conversion of tissue biopsy samples into permanent quantitative digital proteome maps. *Nat Med*, *21*(4), 407-413. <https://doi.org/10.1038/nm.3807>
- Jessani, N., Niessen, S., Wei, B. Q., Nicolau, M., Humphrey, M., Ji, Y., Han, W., Noh, D. Y., Yates, J. R., 3rd, Jeffrey, S. S., & Cravatt, B. F. (2005). A streamlined platform for high-content functional proteomics of primary human specimens. *Nat Methods*, *2*(9), 691-697. <https://doi.org/10.1038/nmeth778>
- Joyce, J. A., Baruch, A., Chehade, K., Meyer-Morse, N., Giraudo, E., Tsai, F. Y., Greenbaum, D. C., Hager, J. H., Bogyo, M., & Hanahan, D. (2004). Cathepsin cysteine proteases are effectors of invasive growth and angiogenesis during multistage tumorigenesis. *Cancer Cell*, *5*(5), 443-453. [https://doi.org/10.1016/s1535-6108\(04\)00111-4](https://doi.org/10.1016/s1535-6108(04)00111-4)
- Koch, E., Wiebel, M., Hopmann, C., Kampschulte, N., & Schebb, N. H. (2021). Rapid quantification of fatty acids in plant oils and biological samples by LC-MS. *Anal Bioanal Chem*, *413*(21), 5439-5451. <https://doi.org/10.1007/s00216-021-03525-y>
- Lentz, C. S., Sheldon, J. R., Crawford, L. A., Cooper, R., Garland, M., Amieva, M. R., Weerapana, E., Skaar, E. P., & Bogyo, M. (2018). Identification of a *S. aureus* virulence factor by activity-based protein profiling (ABPP). *Nat Chem Biol*, *14*(6), 609-617. <https://doi.org/10.1038/s41589-018-0060-1>
- Li, X., & Franke, A. A. (2011). Improved LC-MS method for the determination of fatty acids in red blood cells by LC-orbitrap MS. *Anal Chem*, *83*(8), 3192-3198. <https://doi.org/10.1021/ac103093w>
- Medina, J., Borreggine, R., Teav, T., Gao, L., Ji, S., Carrard, J., Jones, C., Blomberg, N., Jech, M., Atkins, A., Martins, C., Schmidt-Trucksass, A., Giera, M., Cazenave-Gassiot, A., Gallart-Ayala, H., & Ivanisevic, J. (2023). Omic-Scale High-Throughput Quantitative LC-MS/MS Approach for Circulatory Lipid Phenotyping in Clinical Research. *Anal Chem*, *95*(6), 3168-3179. <https://doi.org/10.1021/acs.analchem.2c02598>
- Medina, J., van der Velpen, V., Teav, T., Guitton, Y., Gallart-Ayala, H., & Ivanisevic, J. (2020). Single-Step Extraction Coupled with Targeted HILIC-MS/MS Approach for Comprehensive Analysis of Human Plasma Lipidome and Polar Metabolome. *Metabolites*, *10*(12). <https://doi.org/10.3390/metabo10120495>
- Meier, F., Brunner, A. D., Frank, M., Ha, A., Bludau, I., Voytik, E., Kaspar-Schoenefeld, S., Lubeck, M., Raether, O., Bache, N., Aebersold, R., Collins, B. C., Rost, H. L., & Mann, M. (2020). diaPASEF: parallel accumulation-serial fragmentation combined with data-independent acquisition. *Nat Methods*, *17*(12), 1229-1236. <https://doi.org/10.1038/s41592-020-00998-0>
- Niu, L., Thiele, M., Geyer, P. E., Rasmussen, D. N., Webel, H. E., Santos, A., Gupta, R., Meier, F., Strauss, M., Kjaergaard, M., Lindvig, K., Jacobsen, S., Rasmussen, S., Hansen, T., Krag, A., & Mann, M. (2022). Noninvasive proteomic biomarkers for alcohol-related liver disease. *Nat Med*, *28*(6), 1277-1287. <https://doi.org/10.1038/s41591-022-01850-y>
- Park, H., Song, W. Y., Cha, H., & Kim, T. Y. (2021). Development of an optimized sample preparation method for quantification of free fatty acids in food using liquid chromatography-mass spectrometry. *Sci Rep*, *11*(1), 5947. <https://doi.org/10.1038/s41598-021-85288-1>

- Puri, A. W., Broz, P., Shen, A., Monack, D. M., & Bogyo, M. (2012). Caspase-1 activity is required to bypass macrophage apoptosis upon Salmonella infection. *Nat Chem Biol*, 8(9), 745-747. <https://doi.org/10.1038/nchembio.1023>
- Ravindra, K. C., Ahrens, C. C., Wang, Y., Ramseier, J. Y., Wishnok, J. S., Griffith, L. G., Grodzinsky, A. J., & Tannenbaum, S. R. (2018). Chemoproteomics of matrix metalloproteases in a model of cartilage degeneration suggests functional biomarkers associated with posttraumatic osteoarthritis. *J Biol Chem*, 293(29), 11459-11469. <https://doi.org/10.1074/jbc.M117.818542>
- Rost, H. L., Liu, Y., D'Agostino, G., Zanella, M., Navarro, P., Rosenberger, G., Collins, B. C., Gillet, L., Testa, G., Malmstrom, L., & Aebersold, R. (2016). TRIC: an automated alignment strategy for reproducible protein quantification in targeted proteomics. *Nat Methods*, 13(9), 777-783. <https://doi.org/10.1038/nmeth.3954>
- Rost, H. L., Rosenberger, G., Navarro, P., Gillet, L., Miladinovic, S. M., Schubert, O. T., Wolski, W., Collins, B. C., Malmstrom, J., Malmstrom, L., & Aebersold, R. (2014). OpenSWATH enables automated, targeted analysis of data-independent acquisition MS data. *Nat Biotechnol*, 32(3), 219-223. <https://doi.org/10.1038/nbt.2841>
- Sieber, S. A., Niessen, S., Hoover, H. S., & Cravatt, B. F. (2006). Proteomic profiling of metalloprotease activities with cocktails of active-site probes. *Nat Chem Biol*, 2(5), 274-281. <https://doi.org/10.1038/nchembio781>
- Sinitcyn, P., Hamzeiy, H., Salinas Soto, F., Itzhak, D., McCarthy, F., Wichmann, C., Steger, M., Ohmayer, U., Distler, U., Kaspar-Schoenefeld, S., Prianichnikov, N., Yilmaz, S., Rudolph, J. D., Tenzer, S., Perez-Riverol, Y., Nagaraj, N., Humphrey, S. J., & Cox, J. (2021). MaxDIA enables library-based and library-free data-independent acquisition proteomics. *Nat Biotechnol*, 39(12), 1563-1573. <https://doi.org/10.1038/s41587-021-00968-7>
- Yu, F., Teo, G. C., Kong, A. T., Frohlich, K., Li, G. X., Demichev, V., & Nesvizhskii, A. I. (2023). Analysis of DIA proteomics data using MSFragger-DIA and FragPipe computational platform. *Nat Commun*, 14(1), 4154. <https://doi.org/10.1038/s41467-023-39869-5>

Point-by-point Response to Reviewers' Comments

REVIEWER COMMENTS

Reviewer #1 (Remarks to the Author):

Though I didn't fully understand how the frozen or OCT embedded tissue still retain the enzyme activity status, if those findings are consistent, there must be something going on during the reconstitution or removal of OCT. What is happening during this process might be worth to investigate more detail in the separate study.

Thus, this revised version addressed all previous my concerns properly. Anyway, this reviewer support for the publication.

Reviewer #4 (Remarks to the Author):

After reviewing the revised version of the manuscript, I confirm that the authors have adequately addressed the concerns raised during my previous review. The manuscript now meets the high standards and quality expected for publication in Nature Communications.

Reviewer #5 (Remarks to the Author):

The authors have adequately addressed my concerns, and I believe the comments of the other reviewers in this revision of the manuscript. The new Figure 1 is much easier to read. In my opinion the manuscript is acceptable for publication.

Reviewer #6 (Remarks to the Author):

I have reviewed the information in the revised manuscript, focusing on the lipidomics methods. The authors have addressed several comments, although I have some comments relating to the FFA identification and quantification. Some of these also were asked by Reviewer2. Overall, it's not possible to fully verify if the lipids were properly identified without further information, and the raw data doesn't seem to be available at the zenodo link.

The newly included text states that OCT embedded tissues were used for lipidomics analysis. This is very unusual, and introduces a significant issue since OCT itself is a complex medium containing a mixture of glycols and synthetic resins (Qiagen factsheet), and it can't be excluded that it contains also significant amounts of FFA. Normally for lipidomics, unprocessed, snap frozen, fresh tissue is used. There is no mention of any OCT blanks being also extracted for comparison with tissues, and even if there were, how could the amount of OCT in these blanks be matched to tissues, so

as to calculate an appropriate background level to subtract? In my view, contamination of samples with OCT could render the FFA analysis completely unreliable unless this was properly addressed experimentally. Even if OCT was not used, FFA contamination is a significant problem in lipidomics analysis of samples, and blanks are always needed to ensure that contamination is accounted for. Reviewer2 asked about background contamination, but this has not been clearly answered.

For the untargeted analysis, more information needs to be provided to evidence the proper identification of the FFA using this method. Can the authors provide sample chromatograms showing each FFA, and demonstrate that they match the RT of the standards used. A RT list for lipids measured using this assay should be provided. They say 25 isotopically labelled standards were used for this method, but there is no list of these standards or information about their m/z values. This contrasts with the use of only 4 labelled standards for the targeted assay.

For quantitation in a target assay, it is usual to have a primary authentic standard for each compound being quantified, and an internal standard (for which they used 4), to control for extraction efficiency and relative ionization differences between the IS and the analyte. There isn't any mention in the methods about use of primary standards, so can more information be provided relating to how exactly the quantification was performed using the internal standards. A representative chromatogram for each quantified FFA would also be helpful to evidence the quality of the data.

Reviewer 2 asked how long the samples were stored at -80. This question wasn't clearly answered in the rebuttal. Information based on when samples were collected, then stored should be provided to answer this question. Oxidation is a major problem in cohort studies that go back several years. Although eicosanoids were not measured, several oxidizable FFA were measured and oxidation will result in their loss. So, it would be important to know that samples being compared had been stored for similar lengths of time at the very least, so that oxidation issues are not impacted by different groups being stored for different times.

The lipidomics data doesn't seem to be accessible at the Zenodo link provided. This means the quality of the data can't be checked using the raw data. Provision of a full set of representative chromatograms for both methods would be important at the very least.

Point-by-point Response to Reviewers' Comments:

We sincerely appreciate the detailed feedback provided by the reviewers and editorial team. We thank all reviewers for their thoughtful and constructive suggestion that significantly improved this study. Below, we have addressed the concerns and suggestions raised by Reviewer #6 to ensure that the revised version meets the journal's guidelines.

Reviewer #1 (Remarks to the Author):

Though I didn't fully understand how the frozen or OCT embedded tissue still retain the enzyme activity status, if those findings are consistent, there must be something going on during the reconstitution or removal of OCT. What is happening during this process might be worth to investigate more detail in the separate study.

Thus, this revised version addressed all previous my concerns properly. Anyway, this reviewer support for the publication.

We sincerely thank the reviewer for his constructive suggestion regarding OCT-embedded tissues, which we will certainly consider in the near future. We also thank reviewer for supporting publication of our revised manuscript.

Reviewer #4 (Remarks to the Author):

After reviewing the revised version of the manuscript, I confirm that the authors have adequately addressed the concerns raised during my previous review. The manuscript now meets the high standards and quality expected for publication in Nature Communications.

Thanks to the reviewer for his positive feedback.

Reviewer #5 (Remarks to the Author):

The authors have adequately addressed my concerns, and I believe the comments of the other reviewers in this revision of the manuscript. The new Figure 1 is much easier to read. In my opinion the manuscript is acceptable for publication.

Thanks to the reviewer for his positive feedback and appreciation of our new Figure 1.

Reviewer #6 (Remarks to the Author):

I have reviewed the information in the revised manuscript, focusing on the lipidomics methods. The authors have addressed several comments, although I have some comments relating to the FFA identification and quantification. Some of these also were asked for by Reviewer2. Overall, it's not possible to fully verify if the lipids were properly

identified without further information, and the raw data doesn't seem to be available at the zenodo link.

Response 1: We would like to thank reviewer #6 for his constructive and valuable comments and for recognizing our previous work in revising the manuscript. We appreciate the reviewer's comments, which have helped us greatly to improve the current manuscript.

We apologize if the reviewer reported some problems in downloading the raw data. We provide a link to the revised raw data on *the Zenodo platform* (<https://zenodo.org/>) with Identifier (zenodo.14841692 or <https://doi.org/10.5281/zenodo.14841692>) or link <https://zenodo.org/records/14841692>

Specifically, our raw data are uploaded as two .zip files (folder) corresponding to **lipidomic experiment 1** (zip file “*FFA_FullScan_AMRT_Exp1.zip*” 13.7 GB; raw data of each sample is folder.d readable by MassHunter Agilent Technologies software) and **lipidomic experiment 2** (zip file: “*FFA_Exp2.zip*” 12.4 GB; raw data of each sample is file.raw Thermo Scientific). We suggest that reviewers download these two .zip files containing the respective raw data files. The filenames and descriptions of the raw files are included in sample annotation.xls files.

We have ensured that access to these raw files is available and open to the public:

Published February 2025 | Version v2

Dataset  Open 
Depletion-dependent Activity-Based Protein Profiling coupled to SWATH/DIA Mass Spectrometry detects serine hydrolase lipid remodeling in lung adenocarcinoma progression

Sajic, Tatjana (Contact person)¹ 
Show affiliations

These raw data correspond to a lipidomic analysis of free fatty acids in tumour biopsies to validate the results of the chemical proteomic method.

Main point of the study

Below we supply additional information related to lipidomic analysis and asked for by the reviewer.

1. Comment: The newly included text states that OCT embedded tissues were used for lipidomics analysis. This is very unusual and introduces a significant issue since OCT itself is a complex medium containing a mixture of glycols and synthetic resins (Qiagen factsheet), and it can't be excluded that it contains also significant amounts of FFA. Normally for lipidomics, unprocessed, snap frozen, fresh tissue is used.

Response 2: We thank the reviewer for his constructive feedback. We understand that fresh frozen tissue is the best option for the lipidomic part. However, most of these unique clinical tissue specimens were only available as tissues embedded in OCT.

Our **proteomics** and **lipidomics** LC-MS sample injections were carefully verified and do not display OCT contamination. Indeed, we did detailed research on OCT composition and cleanup. According to our OCT vendors specification (i.e. <https://www.cellpath.com/oct-embedding-matrix-125ml.html>, but also Thermofisher or Qiagen) OCT compound has two main characteristics: i) it is clear, water-soluble compound that contain glycol polymers *soluble in water without residue*; and ii) it is fast-freezing compound for optimal section quality that minimizes tissue degradation preserving the protein structure (e.g. suitable for protein immunochemistry). *OCT could not contain fatty acids that are not miscible with water, specifically those made up of ten or more carbon atoms.*

In our two-step clean-up process, the OCT is washed away during sample preparation by extensive PBS washing steps and the piece of clean tissue is used for proteomics (**a**) and lipidomic (**b**) extraction of interest. (Please see below we provide Total Ion Chromatogram (TIC) of samples in each respective omics).

- a) Proteomics. **For-review Figure 1-Left:** TIC plot showing overlay MS injections from OCT cleaned tissue with standard profiles of high-quality peptide digests. The three independent protein extractions were performed from the same OCT tissue block and each of three extract is tested with 2 independent sample preparation and peptide's digestions based on proteomics method used. **-Right:** Person correlations matrix between the MS injections display high reproducibility of measurements (P=0.96-0.98) between detected peptide features.

For-review Figure 1

b) Lipidomics: **For-review Figure 2.** Lipidome extraction performed from the OCT-cleaned tissue. TIC profiles of 10 different tumor tissues analyzed for FFA content. Overlay of tissue sample chromatograms (N=10) showing similar TIC profiles of MS injected patient samples into two different analytical platforms: **A)** TICs on Agilent QTOF system and **B)** TICs on Orbitrap IQX Tribid MS. NO polymer contamination was detected, and the background signals (based on blank extraction as described below) were subtracted. This is a part of lipidomics data processing and quality control workflow, by default. Importantly, no difference was detected between the TIC profiles acquired from frozen tissue lysates vs. OCT cleaned tissue lysates (See below) **C)** The sample chromatograms - UHPLC-HRMS injections on Orbitrap IQX Tribid MS - in "superimposed mode" show that fresh frozen tissue (middle panel, Tu6 was available as fresh frozen) has the same TIC profiles as OCT cleaned tissue.

For-review Figure 2

- We have included **Supplementary Note 4** in the manuscript and Figure S. Note4.1 as TIC sample chromatograms showing no differences between TIC profiles of fresh-frozen and OCT-cleaned tissue (tissue recorded on LC-HRMS Orbitrap IQX Tribrid MS)

- In the manuscript, we clarify OCT characteristics. Please refer to: the page 4 line 158: “The OCT medium freezes rapidly to optimize the preservation of biological specimens and dissolves in water to facilitate removal from the tissue” and the page 15 method section 4.2. line 721: “To remove the water-soluble OCT medium (www.cellpath.com/oct-embedding-721 matrix) and obtain high quality peptide digests for LC-MS analysis, “

-See also **Supplementary Figure 1a-b** of the manuscript, which shows no differences in the number of proteins, peptides and functional SHs between fresh-frozen and OCT tissues in LC-MS proteomic protocols

-We thank the reviewer for raising this comment, which prompted us to slightly modify Figure (1b) and clearly illustrate the importance of the type of tissue samples and health data record in this study.

We stress that we used routinely collected OCT-embedded lung adenocarcinoma (LUAD) tissues, the same specimens used for TNM classification of patients. We analyzed these specimens (the same tissue blocks) with detailed clinical follow-up available retrospectively and distinguished LUAD patients with low survival rates who were not recognized by pathologists at the time of diagnosis. *Please refer to: Figure 1b.*

When accurate clean-up of OCT-embedded samples was performed, as in our case or in the case of previous lipidomic studies⁽¹⁾⁽²⁾ no measurements contaminations are detected. Of-course If OCT is not removed from tissue pieces, and the OCT traces are directly homogenized with the tissue, the contamination is visible in LC-MS profiles of samples (i.e. TIC variability, typical polymer chains) and this could interfere with analytical MS measurement, but this was not our case.

2. Comment: There is no mention of any OCT blanks being also extracted for comparison with tissues, and even if there were, how could the amount of OCT in these blanks be matched to tissues, to calculate an appropriate background level to subtract? In my view, contamination of samples with OCT could render the FFA analysis completely unreliable unless this was properly addressed experimentally. Even if OCT was not used, FFA contamination is a significant problem in lipidomic analysis of samples, and blanks are always needed to ensure that contamination is accounted for. Reviewer2 asked about background contamination, but this has not been clearly answered.

Response3: We agree with reviewer that use of OCT-embedded tissue should be properly addressed experimentally. In our case, OCT is washed away during sample preparation by the consecutive PBS washing steps. OCT is designed as a neutral water-soluble matrix to facilitate easy removal of tissue from the block holder and does

not penetrate the tissue. Since we only use clean tissues for further lipid extraction, there was no need to perform the OCT extraction. However, we have, of course, performed the blank extraction, using Cal0, which was treated and processed in the same way as the tissue lysates. The background subtraction from this blank extract is always performed, by default, in lipidomics, to avoid introducing any potential contamination bias or bias due to potential carry-over. Our data were corrected accordingly, and we don't have any doubt about the actual FA signals.

3. Comment: For the untargeted analysis, more information needs to be provided to evidence the proper identification of the FFA using this method. Can the authors provide sample chromatograms showing each FFA, and demonstrate that they match the RT of the standards used. A RT list for lipids measured using this assay should be provided. They say 25 isotopically labelled standards were used for this method, but there is no list of these standards or information about their m/z values. This contrasts with the use of only 4 labelled standards for the targeted assay.

Response 4: We apologize for the confusion. In the initial lipidomic screen based on relative comparison of peak areas of FA of interest (experiment 1), we created the RT database by injecting 25 authentic, non-labeled FA standards purchased from Sigma-Aldrich (Darmstadt, Germany). (See below **For-review Table 1** with authentic primary standards information: m/z, RT, individual Sigma-Aldrich reference). These data were used for FA annotation based on **accurate mass and retention time (AMRT)** matching. Please note: These standards were not isotopically labelled.

- We have corrected this mistake in the Methods section 4.5 and in the Supplementary Material (Supplementary Notes 3-4-5), and we apologize for this oversight.

-Please see **page 23, Method section 4.5. Measurements of FFA - Relative comparison of FA levels, line 1142:** “..,we performed targeted data mining by comparing the AMRT of the detected peak areas with an in-house database containing information for 25 FA obtained on pure authentic standards obtained from Sigma-Aldrich (Darmstadt, Germany)”.

In brief, we characterized and analyzed on UHPLC-HRMS Agilent QTOF system mixtures of i) Saturated and Monounsaturated FA and ii) Unsaturated FA (raw files: *TS_FFA_102022_Unsaturated.d* and *TS_FFA_102022_Saturated_Monounsaturated.d*. See below *For-review Figure 3*). The patient samples were analyzed under the same analytical conditions and peak areas were monitored and matched based on AMRT of these panel of fatty acid standards (**For-review Table 1**).

Please note that the list of these primary FA standards and annotated AMRT peak areas in patient tumor lysates are available in the revised manuscript in **Supplementary Note 3** and **Supplementary Data 8**.

For-review Table 1

Profiling of FA levels obtained through a full scan untargeted LC-HRMS in negative (ESI-)mode on 6550 iFunnel Q-TOF MS					
Saturated Fatty acids					
Synonym	Name	Carbon/Ins	Sigma-Aldrich Ref	[M-H] ⁻ (m/z)	RT (min)
Hexanoic	Caproic	C6:0	21529-5ML	115.0765	0.51
Octanoic	Caprylic	C8:0	C2875	143.1078	0.68
Decanoic	Capric	C10:0	C1875	171.1391	1.03
Dodecanoic	Lauric	C12:0	W261408	199.1704	1.67
Tetradecanoic	Myristic	C14:0	70082	227.2017	2.65
Hexadecanoic	Palmitic	C16:0	P0500	255.2330	3.48
Heptadecanoic	Margaric	C17:0	H3500	269.2486	3.71
Octadenoic	Stearic	C18:0	S4751	283.2643	4.08
Eicosanoic	Arachidic	C20:0	A3631	311.2956	4.9
Docosanoic	Behenic	C22:0	216941	339.3269	5.8
Tetracosanoic	Lignoceric	C24:0	L6641	367.3582	6.8
Hexacosanoic acid	Cerotic	C26:0	H0388	395.3895	7.8
Monoenoic fatty acids					
Synonym	Name	Carbon/Ins	Sigma-Aldrich Ref	[M-H] ⁻	RT (min)
cis-9-hexadecenoic	Palmitoleic	C16:1 (n-7)	P9417	253.2173	2.87
cis-6-octadecenoic	Petroselinic	C18:1 (n-6)	P8750	281.2486	3.58
cis-9-octadecenoic	Oleic	C18:1 (n-9)	O1008	281.2486	3.61
trans-9-octadecenoic	Elaidic	C18:1 (z-9)	E4637	281.2486	3.71
cis-11-octadecenoic	cis-vaccenic	C18:1 (n-7)	V0384	281.2486	3.49
cis-13-docodecenoic	Erucic	C22:1 (n-9)	E3385	337.3112	4.99
cis-15-tetracosenoic	Nervonic	C24:1 (n-9)	N1514	365.3425	5.89
Polyunsaturated fatty acids					
Synonym	Name	Carbon/Ins	Sigma-Aldrich Ref	[M-H] ⁻	RT (min)
9,12-octadecadienoic	Linoleic	18:2 (n-6)	L1012	279.2330	3.16
6,9,12-octadecatrienoic	γ-linolenic	18:3 (n-6)	62174-100MG-F	277.2173	2.67
9,12,15-octadecatrienoic	α-linolenic	18:3 (n-3)	L2376	277.2173	2.59
5,8,11,14-eicosatetraenoic	arachidonic	20:4 (n-6)	23401-50MG	303.2330	3.01
5,8,11,14-17-eicosapentaenoic	Eicosapentaenoic	20:5 (n-3)	E2011	301.2173	2.61
4,7,10,13,16,19-docosahexaenoic	Docosahexaenoic	22:6 (n-3)	D2534	327.2330	2.87

FA or FFA= free fatty acids.

As per reviewer request, below we supply extracted ion chromatograms (EICs) for specific FA standards and AMRT matched peaks during the analyses of tissue lysates in **experiment 1**. Please note, we determined their absolute concentrations in **experiment 2** using calibration curves and isotopically labelled internal standards (IS).

For-review Figure 3: Based on the accurate mass and retention time (AMRT) matching to pure standards (mixtures recorded on UHPLC-HRMS Agilent QTOF system), the following chromatograms illustrate the saturated, monounsaturated (MUFA) and polyunsaturated (PUFA) FAs detected in the participants tissue extracts: A). Saturated & monounsaturated FA (e.g. palmitic acid (C16:0) and oleic acid (C16:1 (n-7))) and B). Polyunsaturated FA (e.g. eicosapentaenoic acid (20:5 (n-3)), linoleic acid (18:2 (n-6))).

4. Comment: For quantitation in a target assay, it is usual to have a primary authentic standard for each compound being quantified, and an internal standard (for which they used 4), to control extraction efficiency and relative ionization differences between the IS and the analyte. There isn't any mention in the methods about the use of primary standards, so can more information be provided relating to how exactly the quantification was performed using the internal standards. A representative

chromatogram for each quantified FFA would also be helpful to evidence the quality of the data.

Response 5: We appreciate this important point raised by the reviewer. We have now clearly specified in the method description that we used *the calibration curves* (prepared with authentic non-labeled pure standards) *and isotopically labelled or deuterated internal standards (IS)* (palmitoleic acid-d13, oleic acid-d9, eicosapentaenoic acid-d5 and hexadecenoic acid-d3, Larodan AG, Solna, Sweden) for the absolute quantification of four fatty acids of interest.

To clarify:

In **our experiment 1** (FA screening and relative comparison across tumors), samples were analyzed by LC-HRMS operating in full scan mode in negative ESI on Agilent QTOF. The FA annotation was performed by matching the accurate mass and retention time (AMRT) against an in-house database recorded on pure authentic standards analyzed in the same analytical conditions (**For-review Figure 3**). No isotopically labelled standards were used in this first qualitative, untargeted screen designed as a discovery (screening) approach. (See our response to comment 3).

Our **experiment 2** (i.e., targeted quantification) was performed on a newly acquired UHPLC-UHRMS Orbitrap IQ-X Tribrid system. The FA species that showed significant changes in experiment 1 between the conditions of interest in the relative comparison (e.g., eicopentaeoic acid, oleic acid, palmitoleic acid) and palmitic acid, previously highlighted by proteomic data, were further quantified. *For their absolute quantification, we used the calibration curves (prepared with pure authentic standards (AS) non-labeled) and the isotopically labeled internal standards (IS) spiked with a known concentration.* The linearity of the calibration curves was evaluated for each fatty acid using a 9-point range (Cal0-Cal9). In addition, peak area integration was manually curated and corrected when necessary. Therefore, as described, the corresponding IS were used to correct for the differential ionization and fragmentation efficiencies, depending on the fatty acid chain length and the degree of unsaturation.

*-Please see **page 24, Method section 4.5. Measurements of FFA - “Absolute quantification of FA of interest”** and **line 1175 “Preparation of Calibration Solutions (with AS) and Internal Standard Mixtures (IS - or deuterated IS).***

As requested, we are providing below the chromatograms of quantified FAs and IS extracted from randomly selected sample. (For-review Figure 4)

For-review Figure 4: Extracted ion chromatograms (EICs) for the measured fatty acids in one (randomly selected) tissue lysate (left panel), EIC of the corresponding IS (middle panel) and the associated calibration curves (right panel). The integration of the respective peak areas was always considered in the same way across the samples.

-Please refer to **Supplementary Note 4** of the revised manuscript. We included the list of commercial isotopically labelled internal standards (Table S.Note4) and EICs for the measured fatty acids, and their respective IS.

5. Comment: Reviewer 2 asked how long the samples were stored at -80. This question wasn't clearly answered in the rebuttal. Information based on when samples were collected, then stored should be provided to answer this question. Oxidation is a major problem in cohort studies that go back several years. Although eicosanoids were not measured, several oxidizable FFA were measured and oxidation will result in their loss. So, it would be important to know that samples being compared had been stored for similar lengths of time at the very least, so that oxidation issues are not impacted by different groups being stored for different times.

Response 6: We thank the reviewer for his constructive suggestion. Indeed, for both compared groups the samples were collected simultaneously in the period from 2005-2013, handled and stored in the same way, during approximately the same amount of time.

Importantly, we applied the same routine procedure to all samples from the day of sample collection. To minimize variability for omics analysis, cryosectioning was performed on the same day by the same person (for samples selected in the biobank). This means that the samples of the compared groups were exposed to the same storage conditions and the same experimental procedure, and if oxidation occurred, it occurred in the same way and to the same extent in the samples of two compared groups.

For each individual sample, the date of collection is unique (i.e., the date of the patient's surgery - OP date, see *For-review Table 2*) and is available now in **Supplementary Data 3** of the revised manuscript (*Please refer to **Supplementary Data 3***). In this context, the samples of both compared groups were collected simultaneously between 2005-2013 and used in the proteomic experiments in 2017, implying a storage time at -80°C of about 6-12 years. The lipidomic analysis was performed in 2022, so the samples of both groups will be stored for 11-17 years. This long-term storage related to detailed clinical follow up (i.e. the tracking of patient survival over the years) required an ultra-low temperature of -80°C to preserve the samples for molecular analysis and clinical use.

For-review Table 2

N	Sex	Birth date	OP-Date	Lung cancer type / Stage	Operation ty
1	male	03/02/1934	29/08/2007	AdenoCA middle lobe / Stadium IIIA	middle lobe-resection
2	male	03/05/1929	04/09/2008	AdenoCA middle lobe / Stadium IIIA	extended mid lobe-resectio
3	female	27/09/1942	01/12/2009	AdenoCA left upper lobe / Stadium IIIA	extended left lobe resection
4	female	08/10/1958	21/05/2010	AdenoCA left upper lobe / Stadium IIIA	left upper lob resection

- We also clarify the sample collection/storage timeline in the text of the manuscript. Please refer to **4.1. Study design** (Methods, **Page 14-15**): **line 690**: “Clinical cohort comprised patients who had undergone surgical lung resection at the University Hospital Zurich (UHZ) between 2005 and 2013...” **line 694**: “The tissues were collected routinely on the date of the patient's surgery (OP date) as either tumor or surrounding, non-neoplastic lung parenchyma..” **line 701**: “Samples were handled in the same manner, frozen, and stored at -80°C until used (Table 1).”

6. Comment: The lipidomics data doesn't seem to be accessible at the Zenodo link provided. This means the quality of the data can't be checked using the raw data. Provision of a full set of representative chromatograms for both methods would be important at the very least.

Response 7: We are sorry for this. Please note that the samples are now accessible via the revised Zenodo version. Please refer to this link: <https://zenodo.org/records/14841692>.

We hope that the reviewer will be able to access the data as also explained above in our *response 1*. In addition, the corresponding chromatograms of the FFA of our interest have been provided for both experiments within this file and in the Supplementary data. We hope that this addresses the reviewer's concerns.

References

1. Vaswani A, Alcazar Magana A, Zimmermann E, Hasan W, Raman J, Maier CS. Comparative Liquid Chromatography/Tandem Mass Spectrometry Lipidomics Analysis of Macaque Heart Tissue Flash-Frozen or Embedded in Optimal Cutting Temperature Polymer (Oct): Practical Considerations. *Rapid Commun Mass Spectrom* (2021) 35(18):e9155. doi: 10.1002/rcm.9155.
2. Rohrbach TD, Boyd AE, Grizzard PJ, Spiegel S, Allegood J, Lima S. A Simple Method for Sphingolipid Analysis of Tissues Embedded in Optimal Cutting Temperature Compound. *J Lipid Res* (2020) 61(6):953-67. Epub 20200427. doi: 10.1194/jlr.D120000809.

Point-by-point Response to Reviewers' Comments

REVIEWERS' COMMENTS

Reviewer #7 (Remarks to the Author):

I have limited my comments specifically to the authors' responses to Reviewer #6. Overall, the authors have provided logical and satisfactory answers to Reviewer #6's concerns. However, I have several questions regarding the figures that were submitted, and it would be beneficial for the authors to address these clearly.

Thanks to the reviewer for his positive feedback.

Comment 1 Regarding For-review Figure 3:

The EIC for the oleic acid standard in Figure 3A appears to show two peaks. The authors should discuss this clearly. Additionally, there seems to be significant contaminant peaks present for oleic acid and palmitic acid. I strongly recommend the authors provide three separate chromatograms: (1) solvent-only analysis, (2) procedure blank (samples processed identically but containing only OCT without biological samples), and (3) biological samples. Demonstrating the absence (or presence) of these peaks, especially in the procedure blank, is critical. Even if contaminants are observed in procedure blank, submitting these chromatograms will clarify how background subtraction was performed.

We thank the reviewer for his constructive feedback. As requested by the reviewer, we have provided extracted ion chromatograms (EICs) in the initial screening experiment for measured FAs in the:

- 1) Sample extracts
- 2) Extracted standard (processed in the same way as samples)
- 3) Blank or neat solvent for background subtraction

The EICs of the respective FA are provided in the Supplementary Note 3, which are related to the initial screening of the FA and the relative quantification within the Supplementary Information file (see **Supplementary Note 3 - Figure S. Note 3.2**, page 25/34 of the Supplementary Information file). As previously described, we used either fresh-frozen or cleaned human tissues for the lipid extraction in the organic solvent. We did not perform 'OCT compound extraction' in the respective organic solvent. The extracted blank was used for background correction.

Comment 2: Regarding Figure 4, the very large background observed for palmitic acid again raises doubts about the reliability of the data. From an analytical chemistry perspective, the severe peak tailing observed compromises quantitative accuracy—even if

standard curves are employed. Peak tailing should ideally be avoided to ensure quantitative accuracy. The authors should explain clearly how they validated the peak area ranges of palmitic acid. Similar to the previous point, providing extracted ion chromatograms (EIC) from solvent-only and procedure blank analyses would clarify how background peaks were accounted for.

As requested by the reviewer, we have provided extracted ion chromatograms (EICs) in the absolute quantification experiment for FAs of interest in the:

- 1) Blank or neat solvent
- 2) Extracted blank or calibrator 0 (processed in the same way as samples)
- 3) Pooled sample extract (representative of all samples)

Please see below (For-review Figure 1) the EICs provided for respective FA with internal standards (ISTD). Quantitative accuracy is ensured by the linearity of calibration curve, following the manual peak integration and background subtraction. Despite the peak shape (e.g., tailing), *when integrated manually* respective peak areas nicely align and show linear MS signal response. In addition, the internal standard (IS or ISTD) behaves in the same way in the same analytical conditions which allowed for the matrix effect correction for improved quantification accuracy. Please also refer to Supplementary Note 4 to check entire EICs of the four FA of interest in the absolute quantification experiment, internal standards and extracted blanks (see **Supplementary Note 4 - Figure S. Note4.2**, page 27/34 of the Supplementary Information file).

For-review Figure 1

Plots the EICs reported by Thermo TraceFinder software, optimized for clinical research.